# Klf5 acetylation regulates luminal differentiation of basal progenitors in prostate development and regeneration

Baotong Zhang [1,2], Xinpei Ci[1,2,7], Ran Tao[1,2,8], Jianping Jenny Ni[1,2], Xiaoyan Xuan[1,3,9], Jamie L. King[1,2], Siyuan Xia[1,2], Yixiang Li[1,2], Henry F. Frierson[3], Dong-Kee Lee[4], Jianming Xu [4], Adeboye O. Osunkoya[1,5,6] & Jin-Tang Dong [1,2✉]

Prostate development depends on balanced cell proliferation and differentiation, and acetylated KLF5 is known to alter epithelial proliferation. It remains elusive whether post-translational modifications of transcription factors can differentially determine adult stem/progenitor cell fate. Here we report that, in human and mouse prostates, Klf5 is expressed in both basal and luminal cells, with basal cells preferentially expressing acetylated Klf5. Functionally, Klf5 is indispensable for maintaining basal progenitors, their luminal differentiation, and the proliferation of their basal and luminal progenies. Acetylated Klf5 is also essential for basal progenitors' maintenance and proper luminal differentiation, as deacetylation of Klf5 causes excess basal-to-luminal differentiation; attenuates androgen-mediated organoid organization; and retards postnatal prostate development. In basal progenitor-derived luminal cells, Klf5 deacetylation increases their proliferation and attenuates their survival and regeneration following castration and subsequent androgen restoration. Mechanistically, Klf5 deacetylation activates Notch signaling. Klf5 and its acetylation thus contribute to postnatal prostate development and regeneration by controlling basal progenitor cell fate.

[1] Winship Cancer Institute, Emory University, Atlanta, GA, USA. [2] Department of Hematology and Medical Oncology, Emory University School of Medicine, Atlanta, GA, USA. [3] Department of Pathology, University of Virginia Health System, Charlottesville, VA, USA. [4] Department of Molecular and Cellular Biology, Baylor College of Medicine, Houston, TX, USA. [5] Department of Pathology, Emory University School of Medicine, Atlanta, GA, USA. [6] Department of Urology, Emory University School of Medicine, Atlanta, GA, USA. [7] Present address: Vancouver Prostate Centre, Department of Urologic Sciences, University of British Columbia, Vancouver, BC, Canada. [8] Present address: Department of Essential Surgery, Xiangya Hospital, Central South University, Changsha, Hunan, China. [9] Present address: Department of Immunology, School of Basic Medical Sciences, Zhengzhou University, Zhengzhou, Henan, China. ✉email: j.dong@emory.edu

Prostate epithelia are comprised of columnar secretory luminal cells, cuboidal basal cells, and rare neuroendocrine cells. They require lineage hierarchy to maintain epithelial homeostasis. Adult stem/progenitor cells have been detected in both basal and luminal cells of the prostate. For example, basal cells contain a small population of self-renewing cells[1], when mixed with embryonic urogenital sinus mesenchymal (UGSM) cells and transplanted into the renal capsules of immunodeficient mice, which can reconstitute xenograft tissues that contain the three cell lineages[2,3]. Lineage tracing with basal cell-specific gene promoters of keratin 5 (K5) and p63 revealed that basal cells can indeed give rise to other lineages including the luminal lineage[4–6]. On the other hand, single luminal cells from human or mouse prostates can give rise to organoids that closely resemble the prostate gland, even though luminal cells have a lower organoid forming capacity than basal cells[7–9]. Furthermore, castration-resistant Nkx3.1 expressing cells (CARNs), which constitute a small population of luminal cells, can also generate prostate xenografts with three cell lineages in the renal capsule of nude mice[10], being bipotent and able to self-renew in vivo[10]. Therefore, a small subpopulation of both basal and luminal lineages are adult stem cells/progenitor cells with multiple differentiation potentials.

Postnatal prostate development can be divided into four consecutive stages: initiation or budding, branching and morphogenesis, differentiation, and pubertal maturation[11], with ductal branching in mice normally completed by 60–90 days after birth[12]. Genetic lineage tracing experiments have revealed that both multipotent stem cells and unipotent progenitors from two lineages are responsible for prostate development. One is basal multipotent stem cells that divide and differentiate into basal, luminal and neuroendocrine cells; the other is unipotent basal and luminal progenitors that give rise to basal and luminal cells, respectively[13]. The prostate is androgen dependent, and prostate development originates from the underlying mesenchyme in an androgen-dependent manner. Androgens are also essential for sustaining the growth and differentiation of prostate epithelial cells[14,15]. Androgen deprivation leads to rapid apoptosis of ~90% of luminal cells and a small percentage of basal cells, and the prostate epithelium regenerates completely ~2 weeks after androgen administration[16,17]. Both luminal and basal stem/progenitor cells contribute to prostate regeneration[4,10].

Krüppel-like factor 5 (Klf5), a basic transcription factor, regulates diverse cellular processes including cell proliferation, apoptosis, epithelial mesenchymal transition (EMT), and stemness[18–22]. Klf5 is expressed in prostate epithelial cells[23], and its absence suppresses sphere formation in RWPE-1 human prostate epithelial cells[23]. In prostate tumors induced by Pten deletion[24–26], Klf5 loss alters the constitution of basal and luminal cells[27]. KLF5 exerts opposing functions in cell proliferation and tumor growth in prostate epithelial cells, and its acetylation status regulated by TGF-β and likely other signaling molecules[28–31] is the determinant, with deacetylation of KLF5 at lysine 369 (K369) promoting tumor growth[31] by altering the transcriptional complex of KLF5 in the expression of its downstream target genes such as *CDKN2B*, *MYC*, *PDGFA*, and *CDKN1A*[30,32,33].

In mouse prostates, acetylated Klf5 (Ac-Klf5) is expressed in both luminal and basal cells[23], yet whether Klf5 and its acetylation regulate prostate development is unknown. In this study, we apply in vitro and in vivo models to demonstrate that Klf5 and its acetylation control the cell fate of prostatic basal progenitors to contribute to postnatal development and regeneration of prostates in part by regulating the Notch signaling.

## Results

**Expression of KLF5 and its acetylation in prostates.** Costaining total KLF5 and acetylated KLF5 (Ac-KLF5) with basal cell marker p63 and luminal cell marker CK8 demonstrated that, in human prostates, total KLF5 was detected in about 62% and Ac-KLF5 in 38% of basal cells, while total KLF5 and Ac-KLF5 were detected in ~50 and 15% of luminal cells respectively (Fig. 1a, b). In mouse prostates, 68 and 53% of basal cells and 70 and 27% of luminal cells expressed total KLF5 and Ac-KLF5, respectively (Fig. 1c, d). Therefore, although total KLF5 is expressed in both basal and luminal cells, Ac-KLF5 is more commonly expressed in basal cells (one-fold difference).

**KLF5 is essential for progenitor activities of basal cells.** In two immortalized human prostate epithelial cell lines, RWPE-1 and PZ-HPV-7, *KLF5* was deleted via the CRISPR Cas9 system (Supplementary Fig. 1a, b), and the deletion downregulated basal cell marker ΔNp63 (Supplementary Fig. 1c) and suppressed sphere formation (Supplementary Fig. 1e, f), even though on a plastic surface the proliferation rate was not affected (Supplementary Fig. 1d). In isolated KLF5-null single clones of RWPE-1 cells (i.e., K2, K8, and K9) (Supplementary Table 1), the expression of basal markers ΔNp63 and CK5 was apparently lower while the CK18 luminal marker was not obviously affected (Fig. 2a and Supplementary Fig. 1g), and spheres were hardly formed (Fig. 2b, c). The few spheres that formed had irregular shape and deranged cells (Supplementary Fig. 1h).

*PB-Cre4* mediated deletion of *Klf5* in mouse prostate epithelial cells, which was traced with YFP and occurs in both luminal and basal cells, decreased the percentage of basal cells (Supplementary Fig. 1i). Basal cells have a higher capability for organoid formation, an indicator of progenitor activity;[7] and absence of Klf5 reduced organoid formation (Supplementary Fig. 1j, k; Supplementary Movies 1–3) and disrupted luminal organization of organoids (Supplementary Fig. 1l).

*Klf5* was also specifically deleted in basal cells using *p63^CreERT2* mice, in which the tamoxifen-responsive *p63* promoter activates *Cre* expression only in basal cells upon tamoxifen administration (Supplementary Fig. 2a). We traced Cre-expressing and thus Klf5-null basal cells with YFP by crossing *p63^CreERT2* mice with *R26R^YFP* mice. Immediately after 5-day tamoxifen administration, induced *Klf5* knockout, which was confirmed in both prostates and tails of mice at 3 weeks (Supplementary Fig. 2b), decreased basal cells but did not affect the YFP labeling efficiency (Supplementary Fig. 2c). No two or more adjacent p63+ basal cells were labeled by YFP (Supplementary Fig. 2c).

Five weeks later, *Klf5* deletion in basal cells significantly decreased both YFP+ basal cells (Fig. 2d–f) and the population of CD49f+/Sca-1+ basal stem/progenitor cells (Supplementary Fig. 2f), which were accompanied with reduced proliferation rate of YFP+ basal cells (Fig. 2j, k). Klf5 thus plays a role in the maintenance of basal progenitor cells, even though loss of Klf5 did not cause noticeable histological changes in prostates at least at 8 weeks (Supplementary Fig. S2e). Remarkably, loss of Klf5 also decreased the body weight of mice (Supplementary Fig. 2d), suggesting that Klf5 deletion in p63-expressing cells, which exist in multiple organs, compromises postnatal growth of mice.

**Loss of *Klf5* attenuates basal to luminal differentiation.** Induced *Klf5* deletion in basal cells also significantly decreased YFP+ luminal cells (Fig. 2e, g). The decrease in YFP+ luminal cells by *Klf5* deletion in basal progenitors could be attributed to reduced basal progenitor production, interrupted basal to luminal differentiation, and/or compromised luminal cell division[6,13]. We thus analyzed YFP+ luminal units to clarify the role of Klf5 in basal-to-luminal

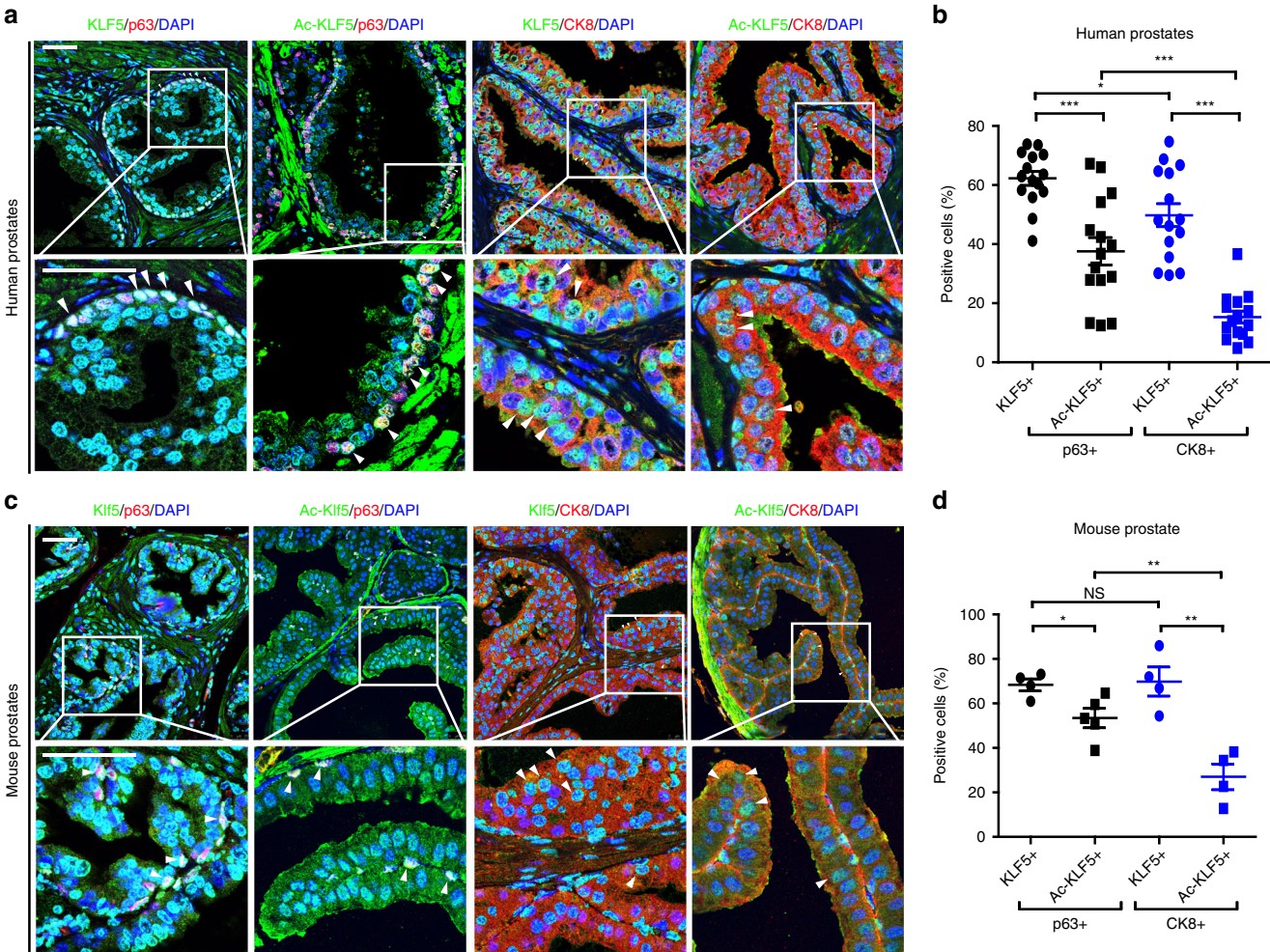

**Fig. 1 Expression of total KLF5 and acetylated KLF5 in the normal prostate gland. a**, **b** Detection of KLF5 and acetylated KLF5 (Ac-KLF5) by IF staining in benign prostate tissues from 15 men. Statistical analysis ($n = 15$ men) was performed in **b**. **c**, **d** Detection of Klf5 and acetylated Klf5 (Ac-Klf5) in an 8-week wild-type mouse prostate. Statistical analysis ($n = 5$ fields for costaining of Ac-KLF5 and p63, $n = 4$ fields for other staining conditions) was performed in **d**. White arrowhead indicates expression of KLF5 or Ac-KLF5 expression in basal (p63+) or luminal (CK8+) cells. Representative images are selected based on the statistical analysis. Scale bar, 50 μm. Data are shown in mean ± S.E.M. NS not significant, *$P < 0.05$; **$P < 0.01$; ***$P < 0.001$ (two-tailed Student's $t$-test). Source data are provided as a Source Data file.

differentiation, where an unit can be either a cluster of at least 2 adjacent YFP+/CK18+ cells or a single such cell. Each cluster should have been primarily derived from a single YFP-labeled p63+ basal progenitor, either by the amplification of a progenitor into a cluster and the cluster's differentiation or by the differentiation of a progenitor into a luminal cell and the luminal cell's division. In either case, basal to luminal differentiation must occur. On the other hand, a single YFP+ luminal cell not in a cluster should have also arisen from the differentiation of a basal progenitor because luminal cells normally do not migrate horizontally due to their tight junctions[13]. Absence of Klf5 significantly decreased the number of YFP+ luminal units (Fig. 2h), supporting a role of Klf5 in the differentiation of basal to luminal cells. In addition, the decrease in YFP+ cells by *Klf5* deletion was more severe in luminal cells than in basal cells (Fig. 2i), indicating that the decrease in YFP+ luminal cells cannot be completely attributed to reduced basal cell proliferation. Consistently, the proliferation of basal progenitor-derived luminal cells (BPDLCs), as indicated by positive staining of Ki67, YFP and luminal markers, was reduced by *Klf5* loss (Fig. 2j, k). Without YFP tracing, the percentage of Ki67+ luminal cells was not significantly affected by *KLF5* deletion (Fig. 2k). Therefore, Klf5 plays a role in both basal to luminal differentiation and the replication of BPDLCs.

**Establishment of a *Klf5^{K358R}* mouse strain.** In *KLF5*-null RWPE-1 human prostate epithelial cells, retroviral expression of wild-type *KLF5* (*KLF5^{WT}*) and acetylation-deficient mutant *KLF5^{K369R}* (KR) restored, but the acetylation-mimicking mutant *KLF5^{K369Q}* (KQ) did not affect, the expression of basal markers CK5, ΔNp63, and CK14 (Supplementary Fig. 3a, b) and efficiencies of colony (Supplementary Fig. 3c) and sphere formation (Supplementary Fig. 3d, e). Expression of luminal marker CK18 was not affected by acetylation status of KLF5 though (Supplementary Fig. 3a). Acetylated and deacetylated KLF5 thus could have different effects on the progenitor capability of prostate epithelial cells (Supplementary Fig. 3a). To more precisely address this question, we generated a transgenic mouse strain in which lysine 358 (K358, homologous site to K369 in human KLF5) can be mutated to arginine upon Cre expression using the same homologous recombination strategy used for targeting *Klf5* in our previous study[23]. The *Klf5^{LSL-K358R}* allele (hereafter referred to as *Klf5^{LSL-KR}*) does not affect wild-type *Klf5* expression (Fig. 3a and Supplementary Fig. 4a–c), while Cre expression converts the *Klf5^{LSL-KR}* allele to the *Klf5^{K358R}* (hereafter referred to as *Klf5^{KR}*) allele while deleting the wild-type allele, which was confirmed at both mRNA and protein levels (Supplementary Fig. 4d, e).

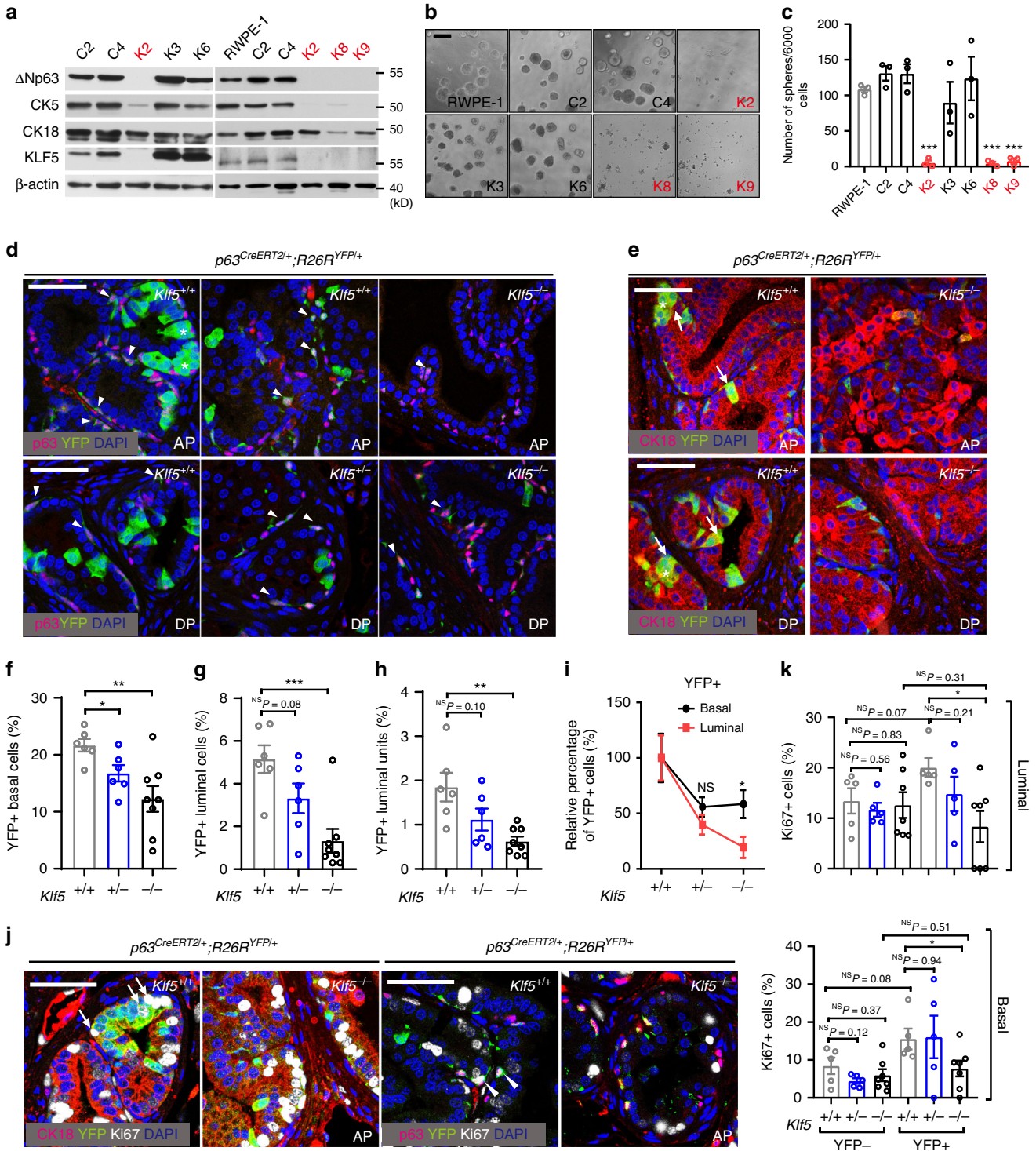

**Constitutive knockin of *Klf5^KR* retards prostate development**.
*Klf5^LSL-KR* mice were crossed to *Ella-Cre* mice, which express Cre in the early embryo and can thus activate the *Klf5^K358R* allele in a wide range of tissues including germ cells that transmit the transgene to progenies. Prostates from 8-week-old mice in with *Ella-Cre* driven *Klf5^KR* knockin were more dense, smaller, and lighter; and overall had less acinar areas and fewer prostate cells (Fig. 3b–e). *Klf5^KR/KR* knockin was more effective than *Klf5^+/KR*. At 12 and 16 weeks, the retarding effect of *Klf5^KR/KR* knockin was still significant (Fig. 3c and Supplementary Fig. 5a); but at 24 weeks, the effect was no longer detectable (Fig. 3c).

Interruption of Klf5 acetylation thus appears to retard postnatal prostate development. While increasing cell proliferation, *Klf5^KR* knockin reduced the percentage of basal cells (p63 or CK5 positive, Supplementary Fig. 5b–e) but increased secretory proteins probasin and Spink3 in prostate fluids (Supplementary Fig. 5f), further suggests that interruption of Klf5 acetylation may cause more basal cells to prematurely differentiate into luminal cells.

In vitro organoid culture provides a convenient model that resembles the in vivo development of prostates[2,7,9], and application of an optimized DHT-containing organoid culture

**Fig. 2 Klf5 is essential for basal progenitors' luminal differentiation and their progenies' proliferation. a–c** Deletion of *KLF5* in RWPE-1 human prostate epithelial cells reduced the expression of basal cell markers CK5 and p63, as measured by Western blotting (**a**), and abolished their sphere forming capability in Matrigel, as indicated by images (**b**) and numbers (**c**) of spheres. $n = 3$ biologically independent samples in **c**. KLF5-null clones are marked in red. **d–i** Deletion of *Klf5*, driven by the p63 promoter and traced by YFP, reduced the numbers of both basal and luminal cells, as indicated by the staining of YFP (**d**, **e**), basal cell marker p63 (**d**), and luminal cell marker CK18 (**e**), percentages of YFP+ cells in total basal cells (**f**) and luminal cells (**g**), percentages of YFP+ luminal units (either single YFP+/CK18+ cell or cluster of 2 or more YFP+/CK18+ cells, **h**), and the percentage of surviving YFP+ basal cells and luminal cells (**i**). Induction of *Klf5* deletion was at postnatal day 18, and prostate tissues were collected at postnatal week 8. In **f–i**, the numbers ($n$) of mice are as follows: $Klf5^{+/+}$, $n = 6$; $Klf5^{+/-}$, $n = 6$; $Klf5^{-/-}$, $n = 8$. Arrowheads and arrow indicate YFP labeled basal (**d**) and luminal cells (**e**) respectively, and stars mark YFP+ luminal clusters (**e**). AP anterior prostate, DP dorsal prostate. **j, k** Knockout of *Klf5* suppressed the proliferation of both luminal and basal cells, as analyzed by costaining the Ki67 proliferation marker, YFP and the CK18 luminal marker or the p63 basal marker (**j**), followed by counting YFP-traced Ki67+ cells (**k**). In **k**, the numbers ($n$) of mice are as follows: $Klf5^{+/+}$, $n = 5$; $Klf5^{+/-}$, $n = 5$; $Klf5^{-/-}$, $n = 7$. Arrows and arrowheads indicate Ki67+/YFP+ luminal and basal cells respectively (**j**). Representative images are selected based on the statistical analysis. Scale bar in **b**, 200 μm; Scale bars in others, 50 μm. Data are shown in mean ± S.E.M. NS, not significant; *$P < 0.05$; **$P < 0.01$; ***$P < 0.001$ (two-tailed Student's *t*-test). Source data are provided as a Source Data file.

medium makes the system particularly relevant to androgen signaling, which is essential for prostate development[7,9]. Basal cells typically have a much higher organoid forming capability than luminal cells[7,9]. Surprisingly, prostate cells with *Klf5*[KR] knockin had fewer basal cells but higher organoid forming ability than *Klf5*[WT] wild-type cells (Fig. 3f, g), but the *Klf5*[KR] knockin significantly reduced DHT-induced typical cystic organoids (from ~90 to ~55%). The *Klf5*[KR] knockin also deranged organoid's typical basal-to-luminal organization with basal cells (CK5+) residing around the outer layers and luminal cells (CK18+) oriented towards the lumen, reducing and deranging basal cells while increasing luminal cells (Fig. 3h). Klf5 acetylation thus plays a role in the proper development of prostates.

**Roles of *Klf5*[KR] in luminal differentiation and proliferation.** To better define the role of Klf5 acetylation in prostate basal progenitors, $p63^{Cre\ ERT2}$ mice were crossed with $Klf5^{LSL-K358R}$ mice, with $Rosa^{YFP}$ mice also introduced to trace Cre-expressing cells, for inducible interruption of Klf5 acetylation. Mice with homozygous $Klf5^{LSL-K358R}$ alleles are not viable, so we crossed $Klf5^{flox/flox}$ mice[23] to $Klf5^{LSL-K358R}$ mice to produce mice with one $Klf5^{LSL-K358R}$ allele and one $Klf5^{flox}$ allele. In these mice, Cre expression deletes *Klf5* in one allele while simultaneously knocking in the $Klf5^{K358R}$ mutation in the other allele, producing $Klf5^{-/K358R}$ mice. $Klf5^{-/+}$ mice were used as controls. Tamoxifen was administered at day 18 after birth to activate Cre expression in p63+ basal cells, and prostates were collected at postnatal week 8, at which point prostates have fully matured.

Immediately after tamoxifen administration, the p63+ basal cells were labeled by YFP at a similar rate (~10%) between different genotypes (Supplementary Fig. 6a). No two or more YFP-labeled p63+ cells were adjacent to each other. Consistent with constitutive *Klf5*[KR] knockin (Fig. 3b–e), conditional *Klf5*[KR] knockin in p63+ cells also retarded prostate development, as indicated by smaller lumens (Fig. 4a and Supplementary Fig. 6b) and decreased prostate weights (Fig. 4b). Interestingly, *Klf5*[KR] knockin increased mitotic figures and resulted in low-grade mouse prostatic intraepithelial neoplasia (mPIN) (Fig. 4a). Conditional *Klf5*[KR] knockin did not cause noticeable gross physical or behavioral abnormalities in mice during the period of investigation, even though the *Klf5*[KR] knockin in p63+ cells should have also occurred in other tissues.

Related to cell fate determination, induced *Klf5*[KR] knockin in basal progenitors dramatically increased the number of YFP+ luminal cells (Fig. 4e–g). Noticeably, the percentage of YFP+ basal cells was not significantly affected (Fig. 4c, d). Furthermore, the percentage of YFP+ cells positive for both the CK5 basal marker and the CK8 luminal marker, which are considered as

intermediate cells during basal to luminal differentiation[34,35], was not affected by *Klf5*[KR] knockin either (Fig. 4j, k).

We then determined the numbers of YFP+ luminal units, which included both YFP+ luminal clusters of 2 or more adjacent cells and single YFP+ luminal cells not in a cluster. In theory, YFP+ luminal clusters should have originated from either the differentiation of a YFP+ basal progenitor cluster, the proliferation of a single YFP+ luminal cell, or the combination of these two processes. A single YFP+ luminal cell not in a cluster should have originated from the differentiation of a single basal progenitor[13]. The number of YFP+ luminal units could thus indicate basal to luminal differentiation while minimizing the effect of cell proliferation on the analyses. *Klf5*[KR] knockin increased YFP+ luminal units by approximately 1.32 folds (Fig. 4i, $p = 0.0098$), indicating that interruption of Klf5 acetylation promotes basal to luminal differentiation.

Consistent with increased YFP+ luminal clusters (Fig. 4h), *Klf5*[KR] knockin promoted luminal cell proliferation, as indicated by IF staining of the Ki67 proliferation marker (Fig. 4l, m). No statistically significant increase was detected in the number of proliferating YFP+ basal cells though (Supplementary Fig. 6d).

Mice were also treated with tamoxifen at postnatal day 7 to induce *Klf5*[KR] knockin, at which time basal cells more actively differentiate into luminal cells for prostate development[13,36]. When prostates were analyzed immediately after tamoxifen treatment, the efficiency of YFP labeling in p63+ basal cells was similar between $Klf5^{-/+}$ and $KLF5^{-/KR}$ (Supplementary Fig. 6e). In prostates collected at postnatal week 6, before sexual maturation, the findings were consistent with those from knockin at postnatal day 18, as *Klf5*[KR] knockin clearly gave rise to more proliferative YFP+ luminal cells while not affecting the number or proliferation rate of basal cells (Supplementary Fig. 6f-n). Therefore, interruption of Klf5 acetylation in p63+ basal progenitors leads to more luminal cells during prostate development by enhancing both the basal to luminal differentiation program and the proliferation of BPDLCs.

**Ac-Klf5 also maintains basal cells in prostate organoids.** Organoid formation assays were performed using YFP-labeled prostate epithelial cells from $p63^{CreERT2/+};R26R^{YFP/+}$ mice immediately after tamoxifen treatment. For wild-type *Klf5* ($Klf5^{-/+}$), DHT treatment significantly increased the number of YFP+ organoids (Fig. 5a) and the ratio of cystic organoids (Fig. 5b), a more typical morphology of normal prostates, and induced nuclear localization of the androgen receptor (Ar) (Fig. 5c), validating the experimental system. For induced *Klf5*[KR] knockin, the number of organoids was significantly increased regardless of DHT treatment (Fig. 5a), but DHT did not increase the ratio of cystic organoids (Fig. 5b).

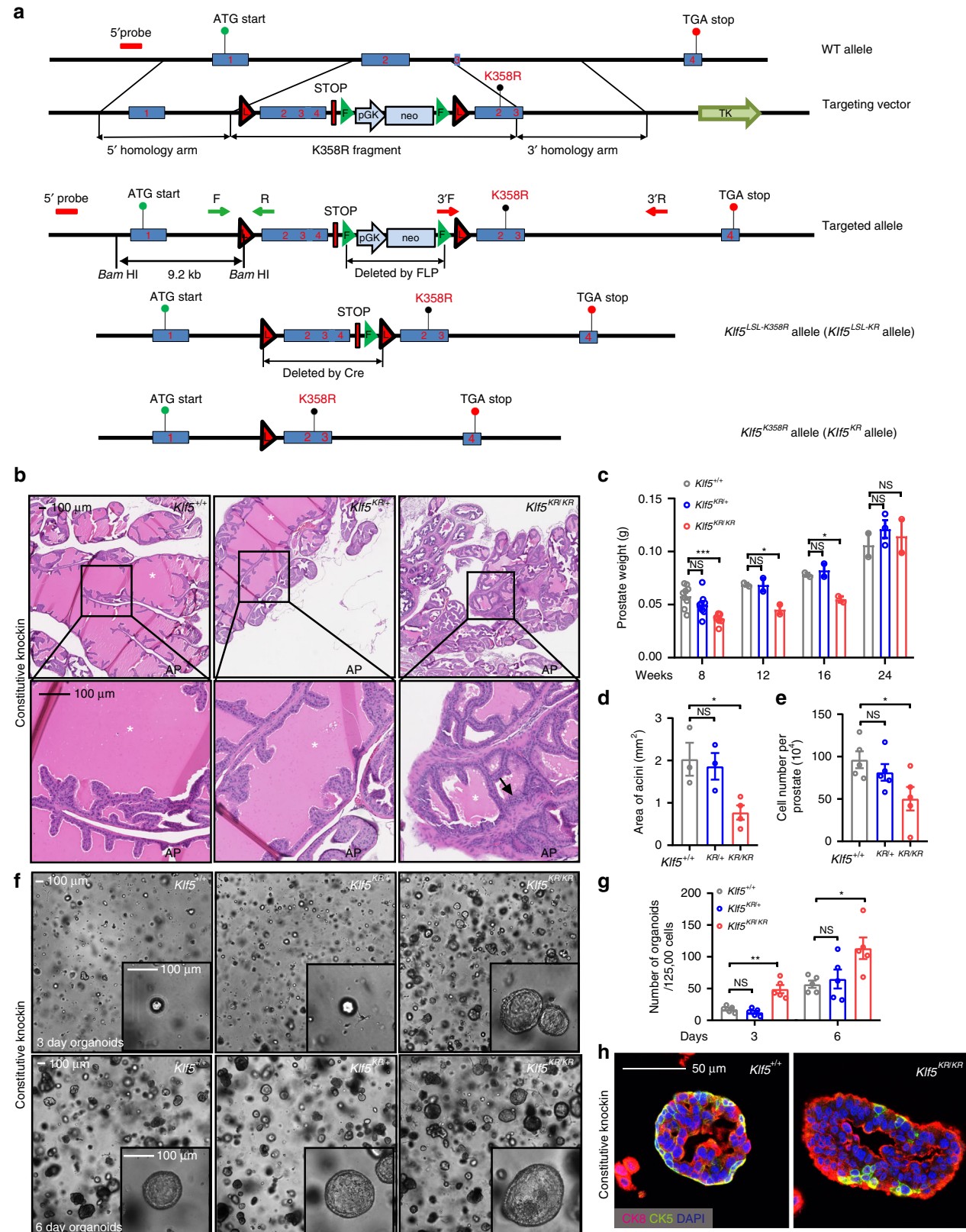

Luminal apoptosis is necessary for the formation of lumens in organoid culture under DHT treatment[37]. Compared to luminal cells without *Klf5KR* knockin (YFP−), those with *Klf5KR* knockin (YFP+) had fewer apoptotic cells (Fig. 5d and Supplementary Fig. 7a).

In addition, while a significant number of basal cells (p63+/CK5+) were observed at the outer layer of wild-type organoids as expected, *Klf5KR* knockin not only significantly reduced the number of such basal cells but also increased the number of luminal cells (CK18+/AR+) in the organoids (Fig. 5c), which

**Fig. 3** $Klf5^{K358R}$ **knockin retards prostate development and promotes abnormal organoid formation. a** Engineering strategy for Cre-mediated knockin of the acetylation-deficient $Klf5^{K358R}$ mutant allele in mice. Shown from top to bottom are the wild-type $Klf5$ locus (WT allele), the targeting vector constructs and their comparison to the wild-type genomic locus, the targeted allele after homologous recombination but before FLP and Cre expression, the $Klf5^{LSL-K358R}$ allele ($Klf5^{LSL-KR}$ allele) after FLP expression, and the $Klf5^{K358R}$ allele ($Klf5^{KR}$ allele) after Cre expression. Primers 3'F and 3'R were used for long-amplifying PCR to confirm the 3' arm; and 5' probe was used for Southern blotting to confirm the 5' arm. Primers F and R were used for genotyping of the $Klf5^{LSL-KR}$ allele. **b–e** Knockin of the $Klf5^{KR}$ allele attenuates normal prostate development in mice, as indicated by abnormal organization of the gland at postnatal week (PW) 8, and revealed by H&E staining (**b**), reduced prostate weights during PW 8-16 (prostate number for **c**: PW 8, $n = 9$ for $Klf5^{+/+}$ and $Klf5^{+/KR}$, $n = 8$ for $Klf5^{KR/KR}$; PW 12 and 16, $n = 2$ for each genotype; PW 24, $n = 2$ for $Klf5^{+/+}$ and $Klf5^{KR/KR}$, $n = 3$ for $Klf5^{+/KR}$), reduced areas of acini in the prostate at PW 8 (d, $n = 3$ for $Klf5^{+/+}$ and $Klf5^{+/KR}$, $n = 4$ for $Klf5^{KR/KR}$ prostates), and reduced number of cells per prostate at PW 8 (e, $n = 5$ prostates). Asterisks indicate prostate lumens, while the black arrow indicates an over-folded epithelium. **f, g** Knockin of the $Klf5^{KR}$ mutant promotes organoid formation in DHT (1 nM) containing medium, as indicated by images (**f**) and numbers (**g**) of organoids at culture day 3 or 6. $n = 5$ biologically independent samples for each genotype in **g**. **h** $Klf5^{KR}$ knockin gives rise to larger organoids with more CK18+ luminal cells but fewer CK5+ basal cells. Data are shown in mean ± S.E.M. NS, not significant; *$P < 0.05$; **$P < 0.01$; ***$P < 0.001$ (two-tailed Student's $t$-test). Representative images are selected based on the statistical analysis. Scale bars in panels **b** and **f**, 100 μm; scale bar in panel **h**, 50 μm. Source data are provided as a Source Data file.

are consistent with the organoids from constitutive $Klf5^{KR}$ knockin.

Wild-type $KLF5$ ($KLF5^{WT}$), acetylation-deficient mutant $KLF5^{K369R}$ ($KLF5^{KR}$), and acetylation-mimicking mutants $KLF5^{K369Q}$ ($KLF5^{KQ}$) and $KLF5^{K369A}$ ($KLF5^{KA}$) were ectopically expressed in YFP+ and $Klf5$-null basal cells isolated from $p63^{CreERT2/+};R26R^{YFP/+};Klf5^{-/-}$ mouse prostates immediately after tamoxifen administration. $KLF5^{WT}$ and $KLF5^{KR}$ gave rise to more YFP+ organoids regardless of DHT conditions, while $KLF5^{KQ}$ and $KLF5^{KA}$ did not (Fig. 5e), suggesting that deacetylation of KLF5 promotes organoid formation of basal cells. Furthermore, $KLF5^{WT}$ expression restored normal differentiation of basal cells into cystic organoids, with basal cells (CK5+) resided around the outer layers and luminal cells (CK8+) orientated towards the lumen (Fig. 5f); and DHT increased luminal cells (CK8+) in these $KLF5^{WT}$ organoids (Fig. 5f, g). For KLF5 mutants, $KLF5^{KR}$ expression led to organoids with many more luminal cells (CK8+); while in contrast, $KLF5^{KQ}$ and $KLF5^{KA}$ mutants led to organoids with more basal cells (CK5+) (Fig. 5f, g). Therefore, KLF5 and its acetylation maintain basal progenitors and their luminal differentiation, and deacetylated KLF5 makes excess basal cells differentiate into luminal cells.

**Klf5 deacetylation overactives Notch signaling.** To explore how the interruption of Klf5 acetylation promotes basal to luminal differentiation, we evaluated molecules from different signaling pathways that have been reported to regulate prostate development by analyzing the expression of these molecules in mouse prostates with both constitutive and induced knockin of the $Klf5^{KR}$ allele (Fig. 6a). Notch1, Ptch1, Fgfr2, and Ar were consistently upregulated in prostates with both types of $Klf5^{KR}$ knockin (Fig. 6a). Among these molecules, increased Notch signaling stimulates the proliferation of luminal cells and promotes the formation of prostate spheres with luminal marker expression[38], similar to the effect of $Klf5^{KR}$ knockin (Figs. 4m, 3h, and 5c, g, and Supplementary Fig. 6m, o). In addition, loss of Notch signaling by disrupting the canonical Notch effector Rbp-j impairs the differentiation of basal progenitors[39], again similar to the effect of $Klf5^{KR}$ knockin. $Klf5^{KR}$ knockin indeed increased the expression of multiple Notch signaling components (Fig. 6b–d), including the Notch1 receptor, Notch1 target genes Hes1 and Myc (Fig. 6b–d), and ligands Jagged1 and Dll1 (Supplementary Fig. 7c). The expression of Notch target gene Hey1 (Supplementary Fig. 7b), processing machinery Adam17, and essential transcription factor Rbp-j were not affected by $Klf5^{KR}$ knockin (Supplementary Fig. 7d).

Organoid culture was performed with prostate cells isolated from $p63^{CreERT2/+};R26R^{YFP/+}$ mice immediately after tamoxifen treatment, which excludes organoids originated from luminal

cells. The numbers of YFP+ organoids were determined in different groups. Treatment with DAPT, a Notch signaling inhibitor, clearly blocked the promoting effect of $Klf5^{KR}$ knockin on organoid formation (Fig. 6f), further implicating overactive Notch signaling in the function of $Klf5^{KR}$ knockin.

Notably, while $Klf5^{KR}$ knockin significantly reduced the number of cystic organoids compared to wild-type Klf5 with DHT treatment (Fig. 6e, g), DAPT treatment eliminated the effect of $Klf5^{KR}$ knockin (Fig. 6g). Notch inhibition by DAPT also rendered smaller, irregular (sometimes hyper budded) and cystless organoids in both the wild-type and $Klf5^{KR}$ groups (Fig. 6e, g), eliminating morphological differences in organoids between the two groups (Fig. 6e, g).

**Ac-Klf5 is required for BPDLCs to survive castration.** Castration induces rapid apoptosis of ~90% of luminal cells and a small percentage of basal cells in the prostate[40], and the homeostasis of prostate epithelia highly depends on androgen signaling. Prostates with constitutive knockin of $Klf5^{KR}$ regressed after castration and regenerated after androgen restoration, as indicated by prostate weight (Fig. 7a). In prostates with constitutive and induced $Klf5^{KR}$ knockin, both luminal and basal cells had a normal pattern of Ar expression (Fig. 7b, c), even though Ar mRNA level appeared to be increased (Fig. 6a).

Some prostate luminal cells can survive castration, and a group of such cells, such as CARNs and CARBs (castration-resistant Bmi1-expressing cells), contain luminal stem cells[10,41]. In prostates with Cre activation in p63+ cells at postnatal (PN) day 18, castration at PN week 8, and tissue collection at PN week 12 (Fig. 7d), ~3.6% of all luminal cells in the wild-type $Klf5$ group were YFP-positive (Fig. 7e), which is comparable to the 3.3% before castration (Fig. 4g), indicating that BPDLCs contain castration-resistant cells (e.g., CARNs, CARBs or others). After $Klf5^{KR}$ knockin, however, the proportion of castration-resistant YFP+ luminal cells dropped to about 0.4% (Fig. 7e), even though the $Klf5^{KR}$ knockin did not affect the survival of basal cells after castration (Fig. 7f). Therefore, acetylation of Klf5 is likely essential for BPDLCs to survive castration.

**Ac-KLF5 is required for androgen-mediated BPDLC regeneration.** Before castration, deletion of $Klf5$ in basal progenitors decreased both YFP+ basal cells and YFP+ luminal cells (i.e., BPDLCs) (Fig. 8a–c and Supplementary Fig. 8a), with $Klf5^{+/+}$, $Klf5^{-/+}$, and $Klf5^{-/-}$ prostates having 5.2, 3.3, and 1.3% BPDLCs, respectively (Fig. 2g and Supplementary Fig. 8h); and these numbers increased ~3-fold to 13.6, 11.5 and 5.2%, respectively, after a castration-regeneration cycle (Fig. 8b and Supplementary Fig. 8h).

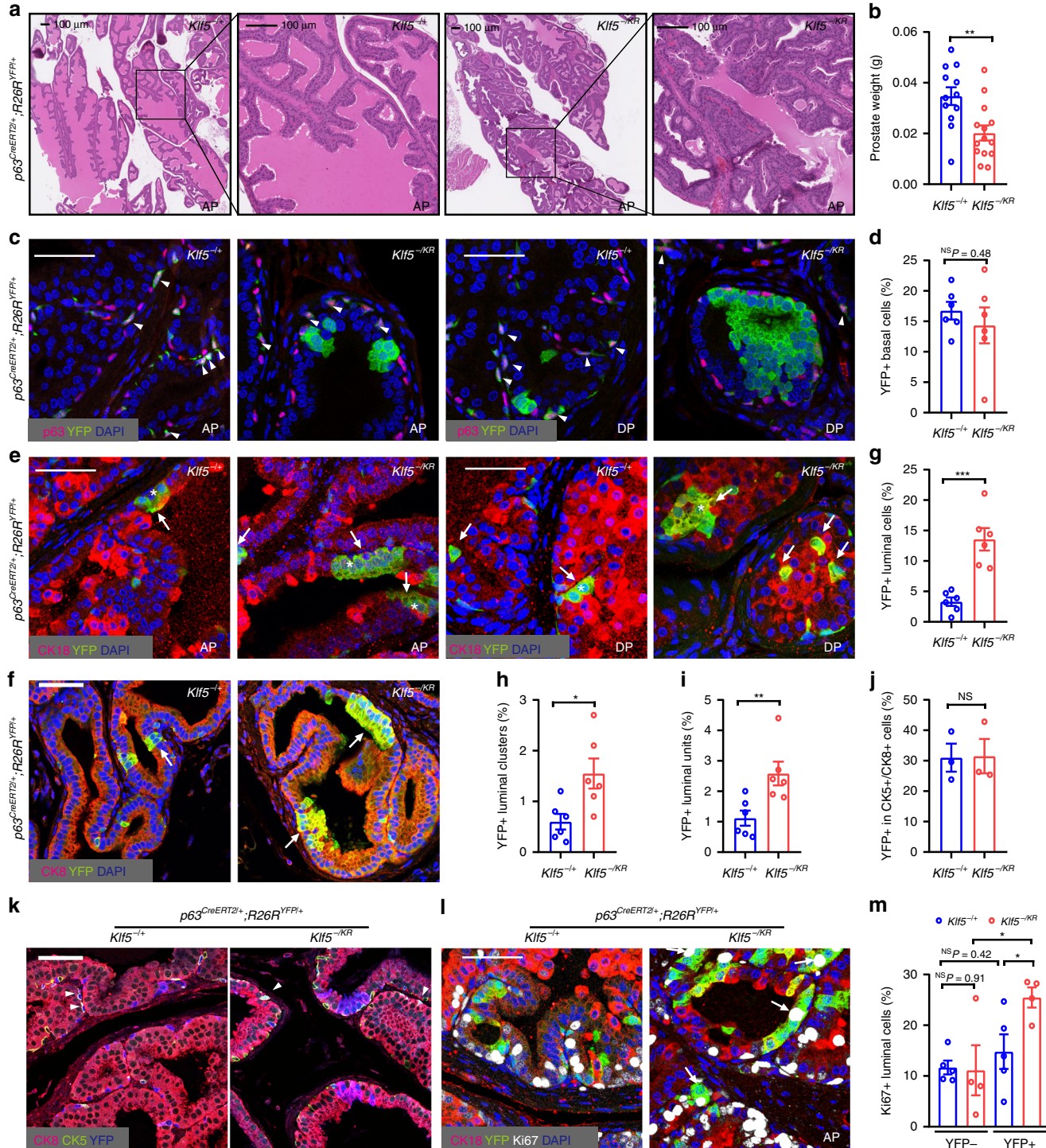

On the other hand, knockin of the $Klf5^{KR}$ allele in basal progenitors resulted in 13.6% of BPDLCs before castration (Fig. 4g and Supplementary Fig. 8h), which is much greater than in those with wild-type $Klf5$ or its deletions (1.3–5.2%, Fig. 2g and Supplementary Fig. 8h). After castration, the percentage of BPDLCs in the $Klf5^{KR}$ knockin group dropped to only 0.4% (Fig. 7e and Supplementary Fig. 8h). After one castration-regeneration cycle, the percentage of BPDLCs increased to only 4.5%, approximately one-third of that before castration (Fig. 8b and Supplementary Fig. 8h). Consistent changes were detected for YFP+ luminal clusters, YFP+ luminal units, and the ratio of YFP+ luminal units to basal cells in the $Klf5^{KR}$ group after a

castration-regeneration cycle (Supplementary Fig. 8b–d), and the luminal cell proliferation index did not increase (Fig. 8d).

In prostates of similar ages without undergoing a castration-regeneration cycle, $Klf5^{KR}$ knockin still increased YFP+ luminal cells, YFP+ luminal clusters, and YFP+ luminal units, while YFP+ basal cells were not affected (Fig. 8e–g and Supplementary Fig. 8e, f), consistent with the results from 8-week-old mice. $Klf5^{KR}$ knockin also increased the ratio of YFP-labeled luminal units to basal cells, although the increase did not reach statistical significance (Supplementary Fig. 8g).

Therefore, while increasing the number of luminal cells before castration, interruption of Klf5 acetylation in basal progenitors

**Fig. 4 Klf5^K358R knockin in basal progenitors increases hyper proliferative luminal cells.** Tamoxifen was applied to mice at postnatal day 18 to induce Cre expression via p63 promoter to knock in the Klf5^KR mutant and activate YFP expression in basal cells, and prostates were collected at postnatal week 8 for analysis. **a, b** Knockin of the Klf5^KR mutant decreased lumen areas (**a**) and reduced prostate weights in mice (**b**). In **b**, Klf5^+/−, n = 12 mouse prostates; Klf5^−/KR, n = 14 mouse prostates. **c, d** Knockin of the Klf5^KR mutant did not significantly affect the number of basal cells, as indicated by costaining YFP and the p63 basal cell marker. **e–i** Knockin of Klf5^KR significantly increased luminal cells, as indicated by costaining YFP and the luminal cell markers CK18 (**e**) or CK8 (**f**) and the percentages of YFP+ luminal cells (**g**), YFP+ luminal clusters (a cluster of 2 or more adjacent YFP+/CK18+ cells, **h**), and YFP+/CK18+ units (either a YFP+/CK18+ single cell or a cluster of YFP+/CK18+ cells, **i**). In **d–i**, the number of mice are as follows: Klf5^+/−, n = 6; Klf5^−/KR, n = 6. Arrowheads indicate YFP expressing p63+ cells (basal), arrows indicate YFP expressing CK18+ cells (luminal), and asterisks indicate YFP+ luminal clusters. **j, k** Knockin of Klf5^KR did not affect the percentage of YFP+ cells in hybrid CK5+/CK8+ cells, as indicated by costaining YFP, CK5 and CK8 (**k**) and statistical analysis (**j**). Three mice were used for each genotype in **j**. **l, m** Knockin of the Klf5^KR mutant increased the proliferation index of luminal cells, as revealed by costaining Ki67, YFP and the CK18 luminal marker. Arrows indicate Ki67+ and YFP+ luminal cells. In **m**, the number of mice are as follows: Klf5^+/−, n = 5; Klf5^−/KR, n = 4. Representative images are selected based on the statistical analysis. Scale bar in **a**, 100 µm; scale bars in others, 50 µm. Data are shown in mean ± S.E.M. NS not significant; *P < 0.05; **P < 0.01; ***P < 0.001 (two-tailed Student's t-test). Source data are provided as a Source Data file.

---

attenuates the regeneration of BPDLCs during a castration-regeneration cycle[42].

## Discussion

Klf5 is essential for the maintenance of basal progenitors, because in mouse prostate organoids, deletion of Klf5 prevented the formation of organoids (Supplementary Fig. 1j–l). In addition, induced knockout of Klf5 in p63-expressing basal cells rapidly decreased the numbers of p63+ cells (Figs. 2f, 8g) and CD49f +/Sca-1+ cells (Supplementary Fig. 2f); the former contain basal stem/progenitor cells[4,5] and the latter define basal progenitor cells. Furthermore, the higher proliferation index in basal progenies was significantly reduced by Klf5 deletion (Fig. 2k lower).

Acetylation of Klf5 is also necessary for basal progenitor maintenance without affecting basal cell proliferation. While prostate basal cells express both acetylated Klf5 and deacetylated Klf5[23], the former was preferentially expressed in basal cells (Fig. 1) and indeed plays a role in the maintenance of basal marker expression, as demonstrated by findings using the acetylation-mimicking mutants KLF5^KQ and KLF5^KA and acetylation-deficient mutant KLF5^KR in organoid formation experiments (Fig. 5g). In addition, while interruption of Klf5 acetylation did not reduce the proliferation rate of basal cells (Supplementary Fig. 6d, i), it decreased the proportion of basal cells in the prostate (Supplementary Figs. 5e and 6c), likely by causing the differentiation of excess basal cells into luminal cells, as discussed below.

Klf5 and its acetylation are indispensable for proper differentiation of basal progenitor cells to luminal cells in the prostate. Differentiation is indispensable for a basal cell to become a luminal cell[13]. Firstly, deletion of Klf5 clearly suppressed basal to luminal differentiation, as indicated by the decrease in YFP+ luminal units (Fig. 2h). Secondly, deacetylation of Klf5 via Klf5^KR knockin dramatically increased the number of BPDLCs in both the induced (Fig. 4e–g) and constitutive (Supplementary Fig. 5b, c, e) knockin systems. In theory, the increase in BPDLCs can be caused by three mechanisms[13]: (1) an increase in basal cells, which leads to basal cell clusters that could subsequently differentiate into luminal cell clusters; (2) an increase in the proliferation of BPDLCs via cell division, which leads to clusters of BPDLCs but not to YFP+ single luminal cells because tight junctions between luminal cells prevent them from migrating horizontally[36]; (3) uncontrolled or undesired differentiation of basal cells to luminal cells, which could exhaust the pool of basal cells while increasing luminal cells. The first scenario is less likely because Klf5^KR knockin did not increase the number of YFP+ basal cells (Fig. 4d and Supplementary Fig. 6l). The second scenario clearly occurs, because Klf5^KR knockin increased not only

the number of but also the proliferation index of BPDLCs (Fig. 4m and Supplementary Fig. 6m). The third scenario also occurs, because the Klf5^KR knockin increases YFP+ luminal units by more than one-fold (Fig. 4i and Supplementary Fig. 6k) but did not increase basal cells, and each YFP+ luminal unit should have arisen from a single YFP+ basal progenitor because luminal cells normally do not migrate due to the tight junctions between them[36].

A role of Klf5 acetylation in basal to luminal differentiation is further supported by findings from the organoid formation assay[2,7,9], where interruption of Klf5 acetylation disrupted the typical basal to luminal organization of organoids, led to cyst-less organoids (Fig. 5b) with fewer and deranged basal cells and more luminal cells (Figs. 3h and 5c), and interfered with the differentiation-inducing function of DHT (Fig. 5b). Furthermore, in Klf5-null organoids, ectopic expression of acetylation-mimicking mutants KLF5^KQ and KLF5^KA kept many more cells as basal but the acetylation-deficient mutant KLF5^KR did not (Fig. 5f, g), suggesting that acetylation of Klf5 restrains basal cells from excessively entering the differentiation process.

Interruption of Klf5 acetylation retards the overall development of prostates, resulting in smaller lumens (Fig. 4a and Supplementary Fig. 6b) and lighter prostate weights (Fig. 4b); less luminal apoptosis in androgen-induced organoids (Fig. 5d), which is necessary for lumen formation;[37] and even a low-grade mPIN (Fig. 4a). Interestingly, the decrease in prostate weight was observed at 8, 12 and 16 weeks but not at 24 weeks (Fig. 3c), suggesting that the effect of Klf5 acetylation could be compensated by 24 weeks of age. We thus propose that acetylation of Klf5 is required for a prostate to maintain a pool of basal progenitors and a balanced proliferation and differentiation of such cells. Without Klf5 acetylation, excess basal cells unnecessarily differentiate into luminal cells, which exhausts the pool of basal progenitor cells, causes excess luminal cells, interferes with proper formation of lumens, and retards postnatal development of prostates.

Roles of Klf5 and its acetylation in basal progenitors and their luminal differentiation could also occur in other organs with epithelia, because the expression of p63 is ubiquitous in the basal cells of various organs such as the lungs, mammary glands, digestive tracts, liver, epidermis etc., and Klf5 has been reported to play a role in basal cell proliferation in the epidermis[43] and in embryonic development and the development of organs such as the lung and mammary gland[44–48]. Such roles could be responsible for the decrease in body weight of mice by Klf5 deletion in p63-expressing cells (Supplementary Fig. 2d).

Among multiple signaling pathways implicated in prostate development, Notch signaling clearly participates in the function of Klf5 acetylation in prostatic epithelial homeostasis, as

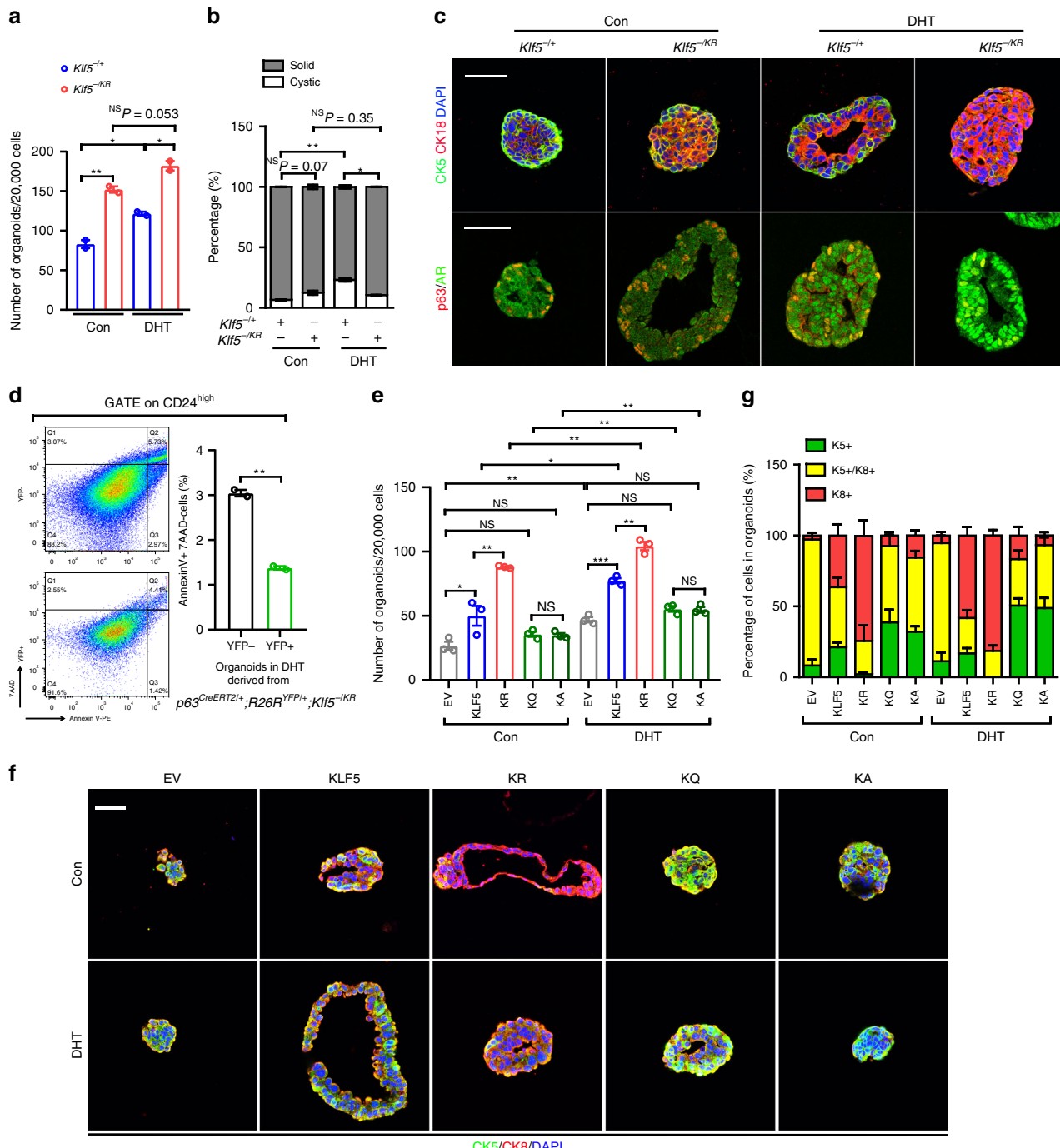

**Fig. 5 Acetylation of Klf5 maintains proper basal to luminal differentiation.** Cells used for organoid formation were isolated from $p63^{CreERT2/+};R26R^{YFP}/+$ mice with indicated *Klf5* genotypes immediately after 5 days of tamoxifen treatment at day 18 after birth. **a** Knockin of $Klf5^{KR}$ increased YFP+ organoids, regardless of DHT conditions. **b** Knockin of $Klf5^{KR}$ eliminated the induction of cystic organoids by DHT (1 nM). In **a**, **b**, $n$ = two biologically independent samples. **c** Knockin of $Klf5^{KR}$ gave rise to organoids with more luminal cells, as indicated by IF staining of basal markers p63 or CK5 and luminal markers CK18 or AR. **d** $Klf5^{KR}$ mutant decreased apoptotic cells in organoids under DHT treatment, as indicated by flow cytometry. $n$ = 2 biologically independent samples. The $Klf5^{KR}$ mutant was knocked in to YFP+ cells. **e**–**g** Wild-type *KLF5*, *KLF5^{KR}* (KR), *KLF5^{KQ}* (KQ) and *KLF5^{KA}* (KA) mutants were ectopically expressed in prostate cells isolated from $p63^{CreERT2/+};R26R^{YFP}/+; Klf5^{-/-}$ mice. EV, empty vector. **e** The number of organoids on day 6. DHT, 1 nM. $n$ = 3 biologically independent samples. **f** Detection of basal marker CK5 and luminal marker CK8 in organoids by IF staining. **g** Statistical analysis of the composition of basal and luminal cells in organoids with different forms of KLF5 in **f**. $n$ = 5–8 organoids. Representative images are selected based on the statistical analysis. Scale bar, 50 μm. Data are shown in mean ± S.E.M. NS not significant; *$P < 0.05$; **$P < 0.01$; ***$P < 0.001$ (two-tailed Student's $t$-test). Source data are provided as a Source Data file.

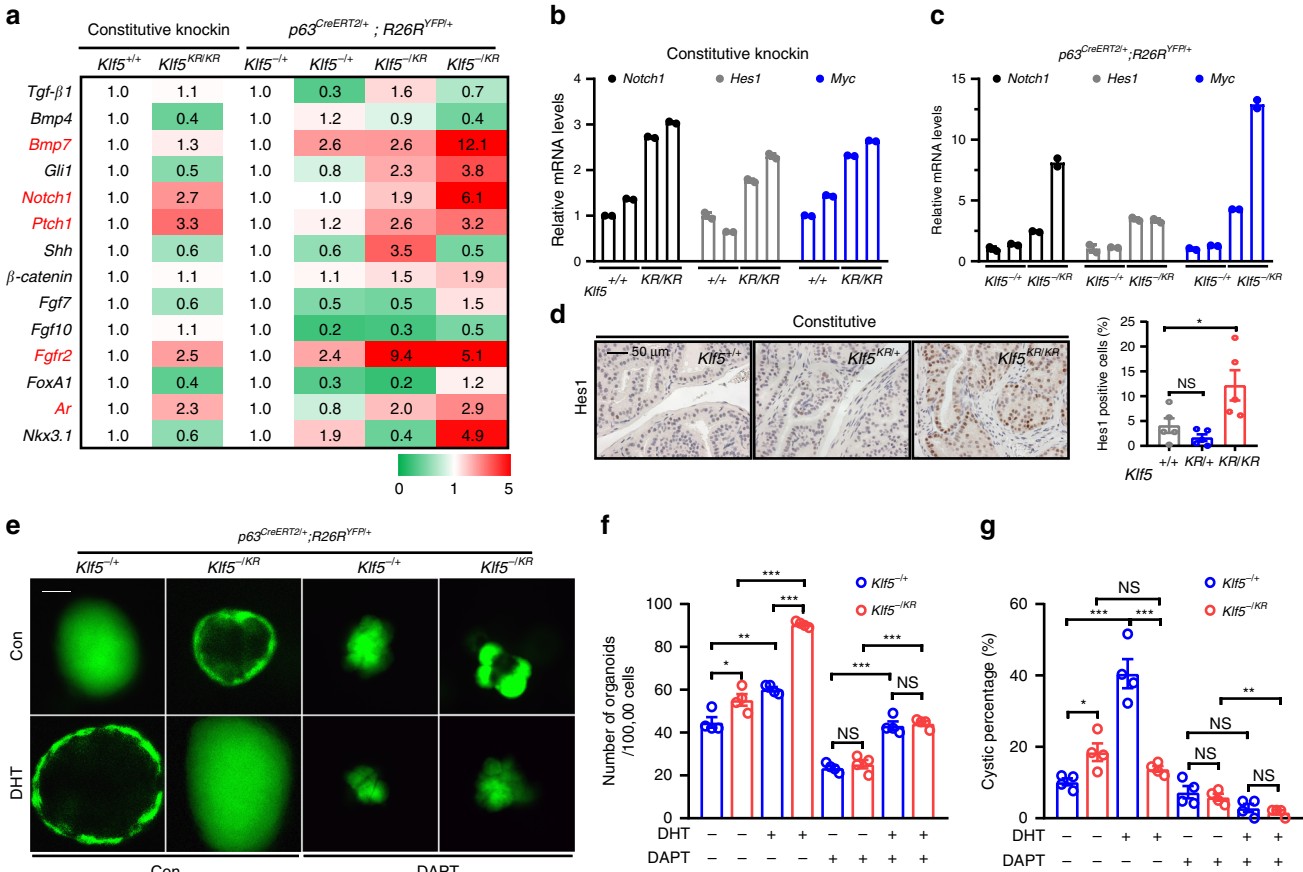

**Fig. 6 Deacetylation of KLF5 increases Notch1 signaling activity. a** Knockin of *Klf5^KR* altered the expression of multiple genes underlying prostate development, as detected by a real-time qPCR based screening. Genes highlighted in red showed consistent expression changes between constitutive and induced knockins. Each of the genotypes *p63^CreERT2/+*;*R26R^YFP/+*;*Klf5^−/+* and *p63^CreERT2/+*;*R26R^YFP/+*;*Klf5^−/KR* had 2 samples from different mouse prostates. **b–d** Knockin of *Klf5^KR* induced the expression of *Notch1* and its downstream target genes *Hes1* and *Myc*, as detected by real-time qPCR in 8-week-old prostates with constitutive (**b**) and induced (**c**) knockins. Experiments were performed in duplicate technically. IHC staining (**d**) was also used to confirm increased Hes1 expression in prostates with constitutive *Klf5^KR* knockin. Five different fields were analyzed for each genotype. **e–g** Inhibition of Notch signaling by the DAPT inhibitor blocked *Klf5^KR*-promoted organoid formation and eliminated cystic formation. Panel **e** shows cystic morphology, and panels **f** and **g** show statistical analyses. YFP labeled p63+ basal cells from 4-week-old prostates with conditional knockin were cultured with or without DHT. In **f–g**, n = 4 biologically independent samples. Representative images are selected based on the statistical analysis. Scale bar, 50 μm. Data are shown in mean ± S.E.M. NS not significant; *P < 0.05; **P < 0.01; ***P < 0.001 (two-tailed Student's t-test). Source data are provided as a Source Data file.

multiple members of the Notch signaling pathway were upregulated by *Klf5^KR* knockin (Fig. 6 and Supplementary Fig. 7), consistent with the previously defined function of Notch signaling in mouse prostates[38,39]. In addition, blocking Notch signaling eliminated the promoting effect of deacetylated Klf5 on organoid formation (Fig. 6f). We noticed that mechanisms other than Notch signaling could also mediate the function of Klf5 acetylation in prostate homeostasis (Fig. 6a), which remain to be determined.

Importantly, acetylation of Klf5 is essential for BPDLCs to survive after castration and regenerate after androgen restoration. Firstly, *Klf5^KR* knockin in basal progenitors sharply decreased BPDLCs that survived after castration (Fig. 7e). Secondly, in the subsequent regeneration by restoring androgens, the reappearance of BPDLCs was again attenuated by *Klf5^KR* knockin (Fig. 8b; Supplementary Fig. 8h). While more than 90% of prostate luminal cells die after castration in mice[16,17], some luminal cells have stem/progenitor features and are castration-resistant (e.g., CARNs[10]). It remains to be addressed whether castration-resistant BPDLCs overlap with CARNs. Necessity of Klf5 acetylation for castration resistance in BPDLCs also implicates KLF5 acetylation in the development of castration-resistant prostate cancer.

Although *Klf5^KR* knockin in basal progenitors makes more basal cells differentiate into BPDLCs than wild-type Klf5 during postnatal prostate development (Figs. 4g and 8f and Supplementary Figs. 6i and 8h), it contribute much less to the regeneration of luminal cells after castration and androgen restoration (Supplementary Fig. 8h). Taken together with the observations that basal cells are essentially not affected by castration, regeneration, or the deacetylation of Klf5 (Figs. 4d, 7f, and 8c); both luminal and basal progenitors give rise to luminal cells during the turnover of prostate epithelia; and luminal cells with deAc-Klf5 were less resistant to castration (Fig. 7e), we propose that Klf5 acetylation is necessary for the maintenance of luminal stem/progenitor cells, and such cells are primarily responsible for luminal regeneration after a castration-regeneration cycle. The *Klf5^KR* knockin mouse model should provide a unique tool for testing this hypothesis.

Findings in this study demonstrate that Klf5 and its acetylation are essential for the maintenance of basal progenitors and proper basal to luminal differentiation in prostate epithelial homeostasis. Acetylation of Klf5 is also indispensable for the survival and regeneration of BPDLCs to castration and androgen restoration (Fig. 9). They also demonstrate that PTM of a transcription

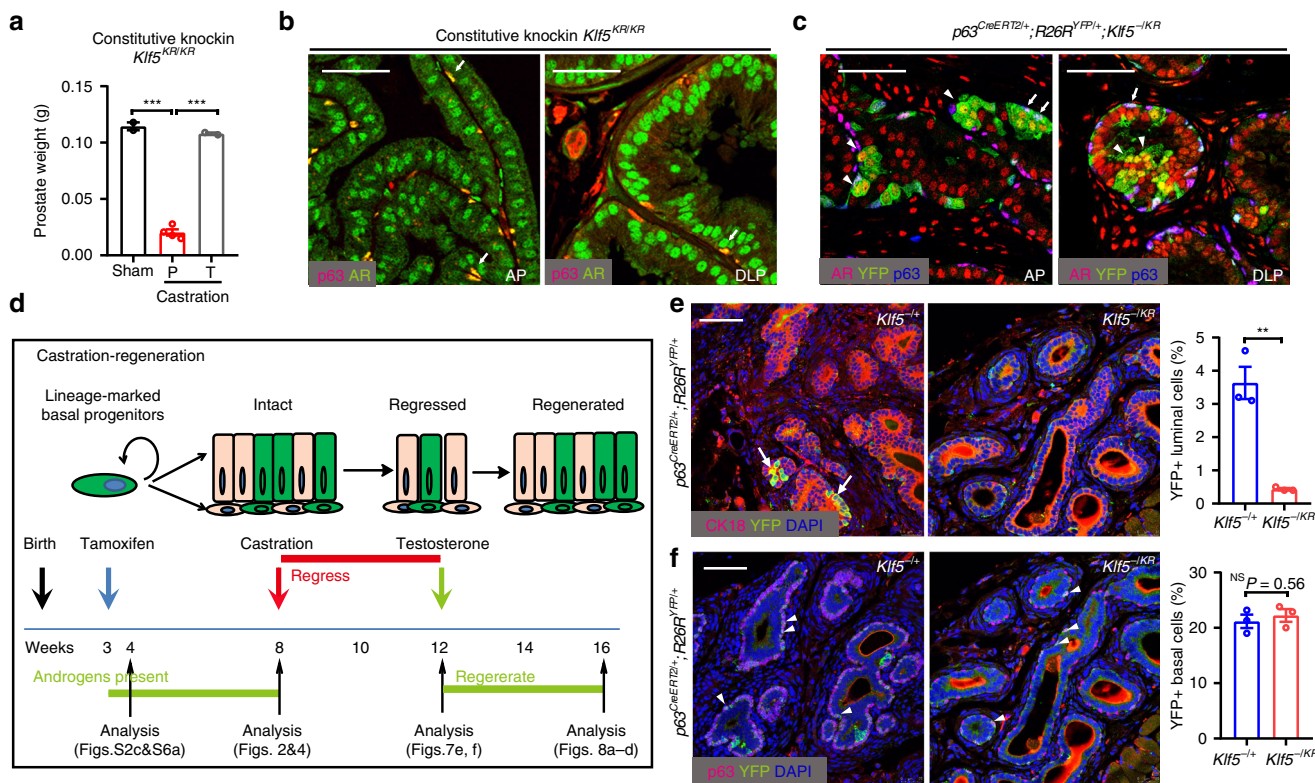

**Fig. 7 Deacetylation of Klf5 attenuates the survival of basal progenitor-derived luminal cells following castration. a** Prostates with constitutive $Klf5^{KR}$ knockin still responded to androgen deprivation and restoration, as indicated by prostate weights after castration and androgen restoration. P, placebo; T, testosterone. The numbers (n) of prostates: $n = 2$ for sham; $n = 4$ for castration+ P; $n = 2$ for castration+ T. **b, c** Expression of androgen receptor (Ar) was maintained in both luminal (indicated by arrowheads) and basal cells (indicated by arrows) in anterior prostates (AP) and dorsal lateral prostates (DLP) with both constitutive (**b**) and induced (**c**) knockins of $Klf5^{KR}$ (YFP+), as detected by IF staining. **d** Schedule schematic for knockin induction by tamoxifen at postnatal day 18, castration at postnatal week (PW) 8, administration of testosterone at PW 12, and regeneration at PW 16. **e** Castration depleted luminal cells with the $Klf5^{KR}$ knockin (YFP+), as revealed by IF staining of the CK18 luminal cell marker and YFP. **f** Castration did not change the percentage of basal cells with $Klf5^{KR}$ knockin (YFP+), as detected by IF staining of YFP and p63. Representative images are selected based on the statistical analysis. Scale bar, 50 μm. Three mice for each genotype in **e** and **f**. Data are shown in mean ± S.E.M. NS not significant; **$P < 0.01$ (two-tailed Student's t-test). Source data are provided as a Source Data file.

factor, i.e., acetylation of Klf5, can profoundly impact the development of a glandular organ.

## Methods

**Mouse strains.** The $Klf5^{LSL-K358R}$ allele was generated by gene targeting using standard procedures at the Transgenic Mouse and Gene Targeting Core of Emory University. Briefly, the same homologous recombination strategy used to target Klf5 in our previous study[23], including the 5′ and 3′ homology arms, was used to introduce the $Klf5^{LSL-K358R}$ allele (hereafter referred to as $Klf5^{LSL-KR}$), which contained two loxP sites, wild-type exons 2-4, stop codon, selection marker neo, and the fragment with the K358R mutation (Fig. 3a). The selection marker neo was removed by FLP to generate a mouse strain with the $Klf5^{LSL-K358R}$ allele, which should express wild-type Klf5 because the coding sequence including the K358R mutation was after a stop sequence. The two loxP sites enabled the excision of wild-type exons 2-4 and the stop sequence upon Cre expression, activating the $Klf5^{K358R}$ mutant allele. Long-range PCR (Supplementary Fig. 4a) and Southern blotting (Supplementary Fig. 4b) were performed to select embryonic stem (ES) cell clones that had the correct insertion of the $Klf5^{LSL-K358R}$ allele. The targeted ES clone 1A4 was microinjected into blastocysts and implanted into recipient albino C57BL/6 female mice. After confirming the $Klf5^{LSL-K358R}$ allele in chimeric mice (Supplementary Fig. 4c), two such mice were selected for breeding. Backcrossing was not required because both ES cells and mice used were in the C57BL/6 genetic background.

Mice homozygous for the $Klf5^{LSL-K358R}$ allele were not viable, which suggests a failure in expressing the wild-type allele of Klf5 from the targeted allele, because lack of Klf5 leads to the failure of ES cells in their derivation from the inner cell mass and thus causes implantation defects and early embryonic lethality[44]. To make this mouse strain useful for this project, we crossed heterozygous $Klf5^{LSL-K358R}$ mice to Ella-Cre mice, which express the Cre recombinase in the early mouse embryo and can thus activate the $Klf5^{KR}$ knockin in a wide range of tissues, including germ cells that transmit the transgene to progenies. By selecting

progenies with the $Klf5^{KR}$ allele but without the Cre transgene, we obtained mice that were homozygous for the $Klf5^{KR}$ knockin allele. These mice were viable and fertile, and did not display noticeable physical or behavioral abnormalities.

Because mice with homozygous $Klf5^{LSL-K358R}$ alleles are non-viable, we crossed the $Klf5^{flox/flox}$ knockout mice[23] to $Klf5^{LSL-K358R}$ mice to produce mice with one $Klf5^{LSL-K358R}$ allele and one $Klf5^{flox}$ allele. In these mice, Cre expression deletes Klf5 in one allele while simultaneously knocking in the $Klf5^{K358R}$ mutation in the other allele to produce $Klf5^{-/K358R}$ mice. $Klf5^{-/+}$ mice were used as controls. This strategy enables the conditional knock in of the $Klf5^{KR}$ allele.

Klf5 knockout ($Klf5^{flox}$, floxed Klf5 allele) mutant mice were established in our previous study[23]. $R26R^{YFP/+}$ (#0061480) mice were purchased from the Jackson Laboratory. PB-Cre4 transgenic mice were obtained from the NCI Mouse Models of Human Cancers Consortium (MMHCC, Frederick, MD, Cat#: 01XF5). Inducible $p63^{CreERT2}$ mice were generated by Dr. Jianming Xu's lab previously[4]. All mice were bred, housed and handled at an Emory University Division of Animal Resources facility according to guidelines and with the approval of the Institutional Animal Care and Use Committee at Emory. $Klf5^{K358R}$ knockin mice are available upon reasonable request and standard MTA procedures.

**Castration and regeneration.** All surgical procedures were conducted under aseptic conditions. The surgical area was decontaminated by shaving and applying sterilizing solutions. Male 8-week old mice were anesthetized with 3.5% isoflurane prior to surgery. To provide a balance between anesthesia and pain control, buprenorphine (0.05 mg/kg) was administered during the castration procedure. After skin preparation, a ventral midline incision was made in the scrotum. The skin was then retracted to expose the tunica, which was pierced, and the opening was stretched with blunt forceps. The testis was pushed out with gentle pressure on the pelvic area; the testicular artery was clamped and ligated, and then the testes were removed. The body wall and skin were closed with poliglecaprone suture (size 5-0, Monocryl. Ethicon, Somerville, NJ). Mice were carefully observed post-surgery and returned to cages after full recovery from anesthesia. Four weeks after castration, mice were implanted with testosterone (T) pellets to replace androgen

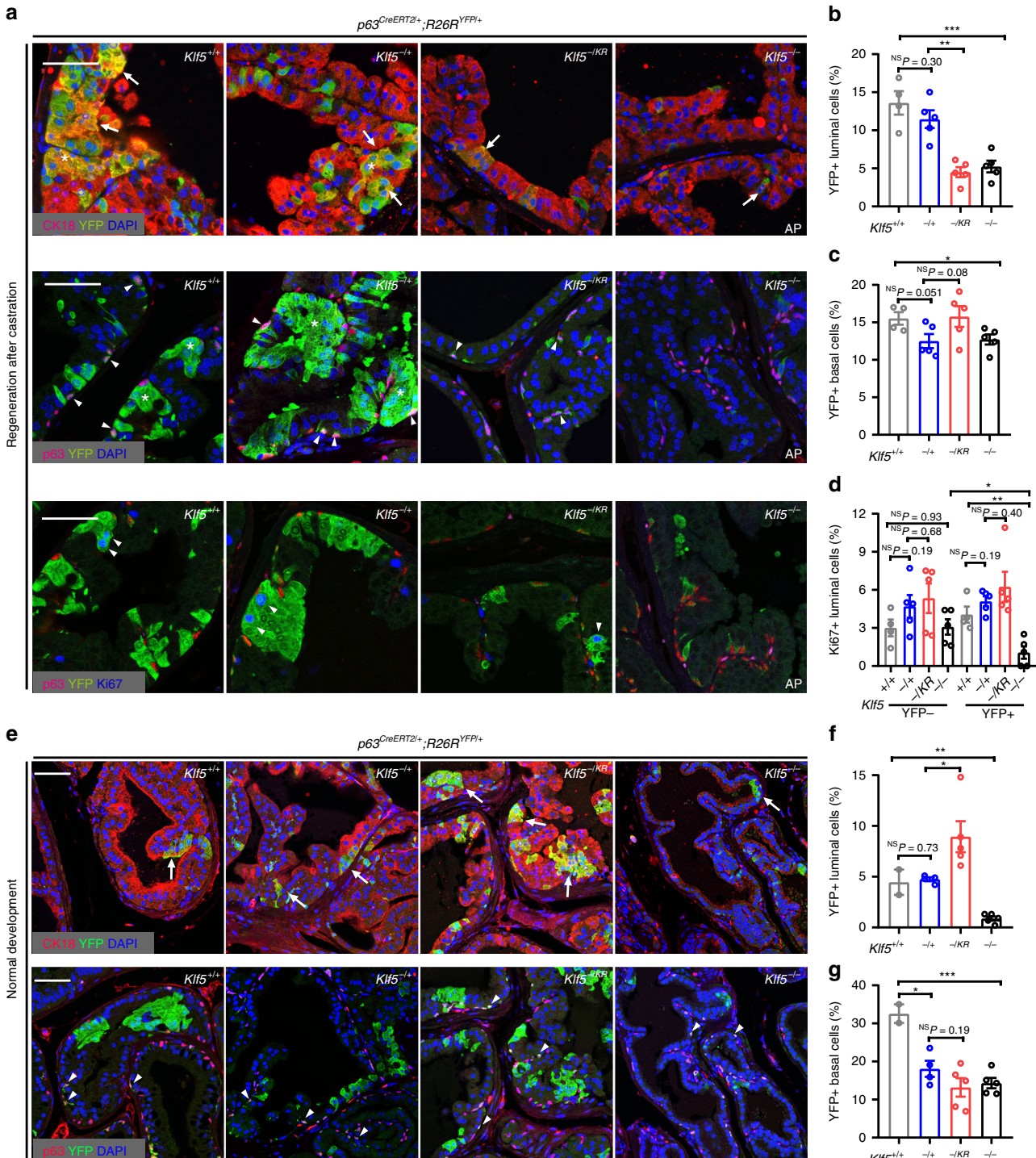

**Fig. 8 Deacetylation of Klf5 compromises androgen-induced regeneration of basal progenitor-derived luminal cells.** Knockin of *Klf5*$^{KR}$ was induced by tamoxifen at postnatal day 18 for 5 consecutive days, and YFP+ basal or luminal cells were analyzed by IF staining after one castration-regeneration cycle (**a–d**) or 16-week normal development (**e–g**). **a–c** Knockin of *Klf5*$^{KR}$ prevented the re-emergence of luminal cells (**b**) but did not have a detectable effect on basal cells (**c**) in regenerated prostates, as detected by IF staining of luminal (YFP+/CK18+) and basal (YFP+/p63+) cells. Arrows point to YFP+/CK18+ cells, and arrowheads to YFP+/p63+ cells. **d** Knockin of *Klf5*$^{KR}$ maintained, while homozygous deletion of *Klf5* significantly reduced, the proliferation rate in luminal cells. IF staining of p63 was used to mark basal cells (**a**, lower). YFP− cells were used as internal controls. In **b–d**, the numbers of mice used were as follows: *Klf5*$^{+/+}$, $n = 4$; *Klf5*$^{-/+}$, $n = 5$; *Klf5*$^{-/KR}$, $n = 5$; *Klf5*$^{-/-}$, $n = 5$. **e–g** Knockin of *Klf5*$^{KR}$ still increased luminal cells (**f**), but did not significantly affect basal cells (**g**), as detected by IF staining (**e**) of luminal cells (YFP+/CK18+) and basal cells (YFP+/p63+). Arrows point to YFP+/CK18+ luminal cells, and arrowheads to YFP+/p63+ basal cells. In **f**, **g** the numbers of mice are as follows: *Klf5*$^{+/+}$, $n = 2$; *Klf5*$^{-/+}$, $n = 4$; *Klf5*$^{-/KR}$, $n = 5$; *Klf5*$^{-/-}$, $n = 5$. Representative images are selected based on the statistical analysis. Scale bars, 50 μm. Data are shown in mean ± S.E.M. NS not significant; *$P < 0.05$; **$P < 0.01$, ***$P < 0.001$ (two-tailed Student's *t*-test). Source data are provided as a Source Data file.

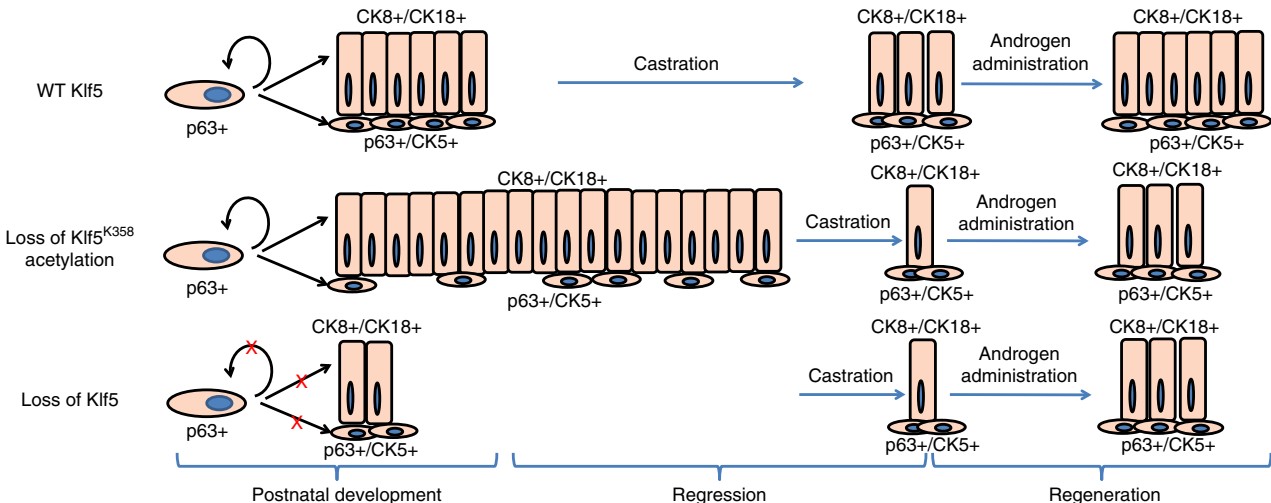

**Fig. 9 KLF5 and its acetylation regulate luminal differentiation of basal progenitors.** An illustrated model to depict the roles of KLF5 and its acetylation in the maintenance and proper differentiation of p63+ basal progenitor cells during postnatal development and the castration-regeneration process.

for experimental groups (15 mg/pellet/mouse; Innovative Research of America, Sarasota, FL) or with placebo pellets for control groups.

**Genotyping**. All mice were toe-clipped at the age of 8-12 days for labeling and genotyping. Toe tissues were incubated overnight at 56 °C with tissue lysis buffer (100 mM Tris-HCl, 5 mM EDTA, 200 mM NaCl, 0.2% SDS, 500 mg/ml proteinase K, pH 8.0). PCR-based genotyping was conducted with the primers listed in Supplementary Table 2. The original data are provided in the Source Data file.

**Histology**. Mice were euthanized at indicated time points according to the requirements of different experimental designs. A mixture of prostate, urethra, bladder and surrounding connective tissues was isolated, and then the prostate dissected from the mixture in PBS. Wet weights of prostates were measured immediately. Tissues for histopathological analysis were fixed in 10% neutral buffered formalin overnight, transferred to 70% ethanol, embedded in paraffin, sectioned at 5 µm, and stained with hematoxylin and eosin (H&E) at the Research Pathology Core Lab at the Winship Cancer Institute.

**Patient samples**. One prostate tissue microarray contains normal prostate tissues from 24 patients, obtained from US Biomax (BNS19011). Some samples were torn and/or had a dark nonspecific background, and had to be excluded from the final statistical analyses. Additional normal prostate specimens were obtained from the Department of Pathology, Emory University School of Medicine. All tissues are collected under the highest ethical standards with the donor being informed completely and with their consent.

**Establishment of isogenic cell lines**. The CRISPR-Cas9 system was used to eliminate KLF5 protein according to the protocol from the Feng Zhang laboratory[49]. Briefly, sgRNA-encoding DNA was designed and synthesized as DNA oligos specific for the KLF5 gene (i.e., 5′-CACCGACGGTCTCTGGGATTTGTA G-3′ and 5′-AAACCTACAAATCCCAGAGACCGTC-3′), annealed, cloned into the CRISPR-Cas9 lentivirus backbone lentiCRISPRv1 vector (Addgene, #49535), and lentiviruses were prepared following standard procedures. RWPE-1 cells (ATCC, Manassas, VA) were infected with lentiviruses and selected in puromycin-containing medium for 72–96 h. Single clones without KLF5 expression (KLF5−/− cells) were identified by Western blotting and confirmed by DNA sequencing (Supplementary Table 1). Retroviruses expressing wild-type KLF5, the acetylation-deficient mutant KLF5 (KLF5K369R), and the acetylation-mimicking mutant KLF5 (KLF5K369Q) were packaged with the envelope vector VSV-G and the gal/pol expression vector Ecopac in 293T cells and then applied to infect KLF5-null single clone, followed by selection in media containing Hygromycin B at 50 mg/ml for 8 days before use.

**Sphere formation and organoid culture assays**. For sphere formation assay (3D culture), RWPE-1 cells were plated at 3,000 cells/well into 8-well chamber slides (BD, #354118) precoated with 50 µl Matrigel (BD, #356231) in serum-free prostate epithelial basal medium (PrEBM) supplemented with 4 µg/ml insulin, B27 (Invitrogen), and 20 ng/ml EGF and bFGF. Sphere morphologies were examined by microscopy after 1–2 weeks of culture, and the total number of spheres was counted and analyzed.

For 3D prostate organoid culture assay, established procedures[9] were followed. Briefly, mouse prostates were dissected and digested by 5 mg/ml collagenase II

(Thermo Fisher, # 17101015) for 1.5 h at 37 °C, further digested to single cells by TrypLE (Life Technologies, # 12605-010), and then filtered with a 40-µm cell strainer. Ten to twenty thousand cells in 40 µl Matrigel (BD, #356231) were seeded onto prewarmed 8-well chamber slides. The culture medium contained 50x diluted B27, 1.25 mM N-acetylcysteine, 50 ng/ml EGF, 100 ng/ml Noggin, 500 ng/ml R-Spondin, 200 nM A83-01, 1 nM DHT and 10 µM Y27632. Organoid morphologies were recorded using phase contrast microscopy after 3–6 days of culture, and the total number of organoids was counted and analyzed.

Intact organoids were released from Matrigel using Dispase (1mg/ml) according to the established procedures[2], fixed in 10% neutral buffered formalin, transferred into 70% ethanol, suspended into Histogel (Thermo Scientific, HG-4000-012), and then embedded and sectioned according to standard histological procedures, followed by detection of basal and luminal markers using IF staining.

**Western blotting**. Briefly, the protein of indicated cells were collected using RIPA buffer (Sigma), and then loaded to SDS-PAGE gel (Bio-rad) for Western blotting following general protocol provided in the website of Cell Signaling Technology (https://www.cellsignal.com/contents/resources-protocols/western-blotting-protocol/western). The KLF5 antibody was generated in our previous study[50]. Other antibodies were purchased from commercial vendors, including ΔNp63 (Biolegend, 619002), CK5 (Biolegend, 905501), CK18 (Abcam, ab668), and β-actin (Sigma, A2066). The original data of Western blotting are provided in the Source Data file.

**Real-time qPCR**. After treatment, cells were lysed in the Trizol reagent (Invitrogen), and RNA was isolated according to the manufacturer's instructions. The cDNAs for mRNA expression analysis were synthesized from total RNA using RT-PCR kits from Promega (Madison, WI). Real-time qPCR primers are listed in Supplementary Table 3. Some primers used for testing genes related to prostate development were based on a previous report[51].

**Immunofluorescence staining and immunohistochemistry**. Formalin-fixed paraffin-embedded tissues were sectioned at 5 µm, deparaffinized in xylene, rehydrated in graded ethanol, subjected to antigen retrieval by boiling the slides in a pressure cooker for 3 min in a citrate buffer (10 mM trisodium citrate, pH 6.0), permeabilized with 0.5% (vol/vol) Triton X-100, and incubated with 10% goat serum and then with primary antibodies overnight at 4 °C. The antibodies included p63 (1:250, Biocare), CK18 (1:100, Abcam), YFP (1:1000, Abcam), CK5 (1:200, Biolegend), CK8 (1:200, Biolegend) and Ki67 (1:300, Thermo Fisher). Tissue sections were then incubated with the secondary antibodies (Alexa Fluor Dyes, Invitrogen, Carlsbad, CA) at 37 °C for 1 h, and DAPI staining was then performed. Fluorescent images were taken with a Leica SP8 confocal microscope at the Integrated Cellular Imaging Core Facility of Emory University. The immunofluorescent images were analyzed using the Fiji program. For staining of Ac-KLF5 and KLF5, the antibody was reported in our previous study[28,50] and used at a dilution of 1:250. For the unmasking step, the staining of Ac-KLF5 was performed in TE buffer (10 mM Tris and 1 mM EDTA, pH 9.0).

For immunohistochemistry staining, most steps were similar to those for immunofluorescence staining. After 10 min treatment with 3% H2O2, tissue sections were blocked with 5% normal goat serum, incubated first with primary antibodies at 4 °C overnight and then with EnVision Polymer-HRP secondary antibodies (Dako, Glostrup, Denmark) at room temperature for 1 h. After the application of DAB-chromogen, tissue sections were stained with hematoxylin,

dehydrated, and mounted. The primary antibodies included Probasin (1:200, Santa Cruz), Spink3 (1:200, Cell signaling), and Hes1 (1:500, Cell signaling). Antibodies specific for KLF5 and Ac-KLF5 were established in our previous study[23]. For the immunohistochemical staining of Ac-KLF5, a Tris-EDTA buffer (10 mM Tris/1 mM EDTA, pH 9.0) was used for antigen retrieval in a pressure cooker.

**Flow cytometry**. Prostate tissues were dissected and dissociated using collagenase II (5 mg/ml), and single cells were isolated by TrypLE (Life Technologies, # 12605-010) digestion and filtration with a 40-μm cell strainer. Single cell suspensions were stained with PE-conjugated anti-CD49f and Alexa Fluor® 647-conjugated anti-CD24 antibodies to indicate basal and luminal cells, or stained with a biotin-conjugated anti-lineage cocktail, PE-conjugated anti-CD49f, and APC-conjugated anti-Sca-1 antibodies, and PerCP-Cyanine5.5-conjugated streptavidin at 4 °C for 30 min to detect prostate basal stem cells[2]. Stained cells were analyzed using a BD (Franklin Lakes, NJ) FACSCantoII flow cytometer, and the data plotted using Flowjo software.

**SRB assay**. The sulforhodamine B (SRB) assay was used to measure cell growth rates[52]. RWPE-1 cells expressing different forms of KLF5 were seeded at 100 cells/ml to test colony formation or seeded at 2500 cells/ml to test cell growth rate using 24-well plates. Cells were collected at different time points after seeding. The SRB assay was performed according to established procedures.

**Statistical analysis**. Readings in all experiments were expressed as means ± standard errors. The statistical significance of differences between two groups was determined by unpaired Student $t$ test with normality tested; and two-sided p-values of 0.05 or smaller, 0.01 or smaller, and 0.001 or smaller are indicated by *, **, and ***, respectively. Two-way ANOVA tests were used for the analysis of cell growth curves in Figure S3c.

**Reporting summary**. Further information on research design is available in the Nature Research Reporting Summary linked to this article.

## Data availability

The data generated or analyzed during the current study are available within the article, supplementary information, attached source data file, and from the corresponding author upon reasonable request. The source data underlying Figs. 1–8 and Supplementary Figs. 1–8 are provided as a Source Data file.

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

## Acknowledgements

We thank Dr. Anthea Hammond of Emory University for editing the manuscript; Dr. Li Xin of University of Washington and Dr. Jing Chen of Emory University for their suggestions for the study. This work was supported by grants R01CA171189 and R01CA193455 from the National Cancer Institute, National Institutes of Health. Research reported in this publication was supported in part by the Integrated Cellular Imaging Core Facility, Winship Research Pathology Core, and Transgenic Mouse and Gene Targeting Core of Emory University Winship Cancer Institute and NIH/NCI under award number P30CA138292.

## Author contributions

B.Z. designed and performed most experiments, analyzed the data, wrote the paper; X.C. established the Klf5^K358R knockin mouse line; R.T., S.X., and Y.L. performed some of the animal experiments; J.J.N. and X.X. performed genotyping and tissue collection; D.-K.L. and J.X. generated the p63-Cre mouse strain; H.F.F. examined histology of mouse prostates; J.L.K. worked on the manuscript; A.O.O. provided some of the patient samples; and J.T.D. designed and supervised the study, provided overall guidance, and revised and finalized the manuscript.

## Competing interests

The authors declare no competing interests.
