## [Peer Review File · Nature Communications]

Reviewers' comments:

Reviewer #1 (Remarks to the Author):

Reviewer: J. Michael Ruppert, West Virginia University

The manuscript by Zhang and colleagues, "Klf5 and its acetylation control the amplification and luminal differentiation of prostatic basal progenitor cells" investigates the roles of acetylated (Ac) vs de-acetylated (deAc) Klf5 in prostate gland epithelial homeostasis. Using both in vitro (spheroid) and in vivo (genetically modified, constitutive or conditional mice) the paper first demonstrates a critical role of Klf5 for proliferation in the prostatic basal epithelial compartment, and then goes on to analyze deAc-Klf5 (Klf5-KR) for effects on basal and luminal proliferation, differentiation and/or recovery from castration +/- androgen therapy. Overall excellent work and very interesting. This is a highly informative and compelling paper but with three major problems as described below.

Major problems:

1) Lack of any evidence that lys-arg substitution (KR) actually mimics unAc-Klf5, or that lys-gln substitution (KQ) really mimics Ac-Klf5. It seems unreasonable to assume that these substitutions are necessarily informative about acetylation of the relevant lysine. For example, they could instead represent loss- or gain-of-function alleles with no relevance to acetylation at all. For example, not all S-D or T-E substitutions necessarily mimic S/T phosphorylation. Additional evidence could include subcellular localization studies, ChIP-seq analysis (differential localization on chromatin), protein stability or transcriptional activity. Analysis of other substitutions such as KA might help with arguments that KQ is specifically mimicking Ac-Klf5 or that KR is specifically mimicking deAc-Klf5. Or a HAT inhibitor might be used to phenocopy deAc-Klf5 in the spheroid assay.

2) Lack of evidence that anti Ac/unAc-Klf5 differentially stains basal and luminal layers in the prostate. Examination of the PLOS One reference provided (see Figs. 3D and 3E) revealed that the study was somewhat preliminary and lacking in several controls. Thus it is unclear whether or how acetylation of Klf5 is changing during differentiation in the prostate, a major gap for the paper.

3) Recurrent, selective interpretation and over-interpretation of the data, often excluding reasonable alternative interpretations. Effects on luminal cell biology are indirectly inferred following knockout in basal cells. The data is otherwise interesting and compelling.

Other Specific Comments:

1. The paper shows a clear role of Klf5 in basal layer of prostate, but conclusions about the role of Klf5 in luminal cells are all based upon indirect evidence, ignoring the possibility of epiphenomena related to alteration of the basal compartment. This is especially worrisome because effects in spheroids were different from effects in the mouse prostate (eg, data on the KR allele). There were no experiments that directly targeted luminal cells and no data indicating differential expression or activity of Ac/unAc-Klf5 in luminal cells. Therefore, conclusion of a role for Klf5 in luminal cells is not warranted, despite repeated over-interpretations in this area. See for example line 29 (Abstract).

2. Line 31 of the Abstract: "the function of Klf5 in progenitor cell amplification was restricted to deacetylated Klf5 (deAc-Klf5)". This statement does not seem to accurately reflect most of the data in the paper, which show that deAc-Klf5 (the KR allele) confers slow maturation of the prostate and also promotes basal to luminal differentiation.

3. Line 98: "whether Klf5 plays a role in lineage determination of epithelial cells has not been studied". This broad statement seems potentially problematic and should be double checked for

accuracy. Klf5 has clear roles in epithelial contexts such as corneal epithelium, mammary gland and perhaps in the gut.

4. Line 104: Ensure gene and protein nomenclature compliant with journal guidelines eg p15 vs CDKN2B.

5. Fig 1A: need to repeat the panel on right side. The Klf5 immunoblot didn't work very well as indicated by weak or no specific signal in control cells (comparing K2 to C2 and C4, all appear to have some positive signal); not clear why leftmost panel is necessary if the blot is successfully repeated (i.e., delete lanes 1-5).

6. Fig. 1A: Lack of any effect on CK18 in the RWPE-1 model is perhaps inconsistent with a role for Klf5 in luminal differentiation, as interpreted below from Fig. 1D and Fig. 1E data in the p63-Cre-ERT2 model. Although in vivo data should trump negative results in vitro, this apparent inconsistency is never addressed.

7. It is shown clearly in Fig. 1 that loss of Klf5 reduces both basal and luminal cell populations. However, to make the case that Klf5 is necessary for the proliferation of basal cells, subsequently their differentiation to luminal cells, and finally proliferation of these luminal Klf5 deficient cells, it would seem critical to complement the p63-Cre-ERT2 data with a luminal promoter-Cre driver. The current analysis suffers from overinterpretation and ignores possible indirect effects on luminal populations when the basal compartment is compromised by Klf5 deletion. A simple solution here might be to simply re-work the conclusion type statements in the paper, with modification of the title and abstract.

8. Data in Figs. 1D and 1E appear consistent with the notion that Klf5-deficient basal cells never make it to the luminal compartment at all. This might be an interesting result to pursue. Note that not every YFP+ cell is necessarily Klf5-deficient, so an occasional YFP+ luminal cell is still consistent with failure of Klf5-deficient cells to differentiate.

9. Fig. 1: for this figure DP and AP are apparently never defined as dorsal and anterior prostate, respectively. AP and DLP are however, defined in Fig. 6 legend.

10. Line 184 and Fig. 1H and 1I: Groups of YFP+ luminal cells, apparently used here to measure proliferation of luminal cells, could also derive from cell division of YFP+ basal cells, followed by their differentiation. Because basal proliferation is already clearly suppressed by deletion of Klf5, the conclusion that Klf5 is also impacting cell division of luminal cells is not well supported (no direct evidence). The counting of YFP+ luminal cell units does not appear to add much to the story and could be deleted.

11. Fig. 1L, 1M and S2F showing selective loss of Ki67 in Klf5-deficient luminal cells: If Ki67 is truly unaffected in the YFP negative cells in the same section (these are control Klf5 wild type cells), then make this critical case within Figure 1 and not in the supplement Fig. 2F. Lack of statistical significance of Ki67 "without YFP tracing" is not a relevant comparison, and is not a substitute for directly comparing Ki67 in the YFP+ and YFP negative luminal cells of the exact same tissue section.

12. Legend vs Results for Fig. 2C: unlike in the Results section describing Fig. 2C, in the legend it is stated that Klf5-KQ (presumably mimicking acetylation) "prevented" sphere formation. The Results section is worded more correctly.

13. Figs 2E and 2F could be switched as they are discussed out of order.

14. Paragraphs beginning on lines 254 vs 265: in these two paragraphs the FK358R allele is described firstly as failing to express wt Klf5 during development, and then later as expressing wt

Klf5 in adult tissues. This is confusing to this reader and is not well explained.

15. Line 293: "the absence of Klf5 acetylation causes precocious differentiation of basal cells to luminal cells". This conclusion seems at odds with spheroid data in Fig 2E. In spheroid model the KR allele promotes a basal phenotype, basal cell proliferation and possibly a block to luminal differentiation, whereas in the mouse it promoted precocious differentiation into luminal cells. These contextual differences should be discussed and perhaps attributed to differences such as the expression level (RT-PCR or IHC data).

16. If not already shown, staining the KR+ mouse tissues for total Klf5 would be important for ascertaining whether it is expressed at wt levels.

17. Fig. 3 data is mostly describing disordered development by the constitutive KR allele and is not very revealing. This data could be moved to the supplement.

18. As prostate function was apparently normal (?) in the KR mice, Figure 3 argues against any essential role of deAc-Klf5 in prostate development.

19. Line 310: Conclusion regarding "necessity of Klf5 acetylation for proper differentiation of basal cells to luminal cells during prostate development". The data in Fig. 3 doesn't necessarily implicate Klf5 function in luminal differentiation. Is it possible that forcing proliferation of basal cells leads to a compensatory increase in luminal differentiation?

20. Line 312: without further development this paragraph could be deleted.

21. Line 351 (re Fig. 4G) , this conclusion is not supported by the data, it could just as easily be that KR knockin promotes basal proliferation which then leads to enhanced luminal YFP+ groups as observed.

22. Line 359: "as indicated by more Ki67+ cells in YFP+ luminal cells but not in YFP- luminal cells (Figure 4I and 4J)." Examination of these figure panels reveals there is in fact no analysis of YFP- luminal cells illustrated, despite this indication in the Results text. This YFP- data is a critical missing control.

23. Line 362: "Collectively, these findings suggest that conditional knockin of KR in p63+ progenitors boosted their luminal differentiation program (Figure S5E), activated luminal cell proliferation (Figure 4G, 4I and 4J), and gave rise to more luminal cells during prostate development." The conclusion is not well supported by the data. Alternatively, it seems equally possible that luminal cells with the KR allele have a hybrid phenotype with aspects of both basal and luminal cells. This dedifferentiated state might help to explain the delayed development and smaller size of the prostate, which otherwise seems inconsistent with the data.

24. Line 367, this paragraph is confusing. Change "KR knockin" to "conditional KR knockin". And maybe change "germline knockin" to "constitutive knockin".

25. Line 448. "The decrease by KR in castration-resistant luminal cells but not in basal cells indicates that acetylation of Klf5 is required for the generation and/or maintenance of most castration-resistant luminal cells in the prostate." The data has other possible conclusions. This data, once again, suffers from an indirect approach, wherein conditional genetic changes are induced in one cell type (basal cells) and the phenotype is concluded to indicate a direct role of Klf5 acetylation in another cell type (luminal cells). The approach suffers from the possible contribution of indirect effects of an altered basal compartment on luminal cells.

26. The paper relies upon powerful genetic methods to study the role of Klf5 in prostate basal cells and their differentiation into luminal cells. However the interpretation of function and causality repeatedly attempts to extend to luminal cell biology, even though genetic studies are entirely restricted to the basal layer (PB-Cre, p63-Cre-ERT2). Consequently it is difficult to discern direct vs indirect effects and much of the data can be interpreted in more than one way.

27. In particular, there is the possibility of direct effects of deAc-Klf5 in basal cells leading to indirect effects on luminal cells that have little to do with Klf5 function. This is especially true as expression of deAc-Klf5 (Klf5-KR) in spheroids, the simpler model, indicates a select role in promotion of basal cell proliferation and perhaps a block to luminal differentiation. In contrast in the mouse model the KR allele (potentially mimicking deAc-Klf5) promotes luminal differentiation in context of a small gland that is slower to develop.

28. Opportunities to utilize YFP-negative cells in the same section as a relevant control were repeatedly missed, further raising concerns about indirect effects.

29. Studies analyzing the role of deAc-Klf5 (whether KR represents deAc-Klf5 is not clear) in castration and regeneration seem descriptive and phenomenological.

30. Line 558 (Discussion): Although it is concluded based upon indirect evidence that Ac-Klf5 is indispensable for proper luminal cell formation, there were few experiments using the KQ allele (putative acetylation mimic).

31. The Discussion section seems very very long.

32. It should be possible to rewrite the paper without overstating conclusions re Ac/deAc-Klf5 function in luminal cells.

--

Reviewer #2 (Remarks to the Author):

Using multiple mouse models as well as human prostate cell lines the author aimed in the present manuscript to investigate the role of Kruppel-like factor 5 (Klf5)- a basic transcription factor known to regulate multiple cellular processes including stemness/differentiation- in the development, differentiation and homeostasis of the prostate. The transcriptional activity of Klf5 is regulated by acetylation of lysine 358, thus the authors have generated several mouse strains including prostate-specific (conditional knockout), germline and p63+ basal cell-limited strains lacking Klf5 expression (knockout) or with acetylation-deficient Klf5 (by mutating lysine 358 to arginine) knockin. Most results have been generated by prostate in situ immunofluorescence stainings and ex vivo organoid culture of prostate-derived cells. The major findings of this manuscript are a requirement of Klf5 acetylation in luminal cells for proper prostate differentiation, while non-acetylated Klf5 is

essential for basal progenitor characteristics. Furthermore, acetylated Klf5 could be important for castration resistance of luminal cells, a finding that could be of value for prostate cancer researchers. Mechanistically, a contribution of Notch signaling on basal-to-luminal differentiation was investigated in more detail.

The study was well designed, experiments were conducted well-controlled, and results are convincing and statistically significant. Work presented is novel and of high interest for researchers in the field of prostate development/differentiation, but also for cancer research. All in all, the study is worth being published, although a major concern and few recommendations need to be addressed:

Major concern:

Mouse and human prostate differ substantially in organ morphology and tissue architecture,

although many but not all molecular features are similar. To increase the impact of this study it would therefore be worth to confirm whether the findings on mouse Klf5 acetylation are of relevance in the human prostate, too. This has been done only insufficiently in few experiments in the beginning of the study with two HPV-immortalized human prostate cell line/organoid models. The HPV oncoproteins E6/E7 are known for their ability to interfere with differentiation, thus interference with KLF5 signaling cannot be excluded. Relevance for KLF5 for human prostate differentiation could be addressed by using non-immortalized cells cultures obtained from normal areas of prostatectomy specimen to reproduce key cell line experiments. Expression analysis of acetylated KLF5 (along with unacetylated KLF5, basal/luminal markers, NKX3.1, etc...) in human prostate (cancer) specimen by (fluorescent) IHC would also greatly improve the impact of this study.

Recommendations:

- 1) A scheme/cartoon on the different cell types/lineages in the prostate along with markers would be helpful.
- 2) Although well-described in the text, the understanding of the generation of mouse strains would be eased by small schemes integrated into the main figures.
- 3) Figure 5A: what is the difference between rows 3&4 and 5&6 (p63-CreERT2 model)?
- 4) Please revise legends to the figures to contain all relevant information and abbreviations for a "stand-alone" reading and understanding of the figures.
- 5) The discussion is way too long and is more a repetition of the results section, rather than a true interpretation of the findings in the context of published literature.

--

Reviewer #3 (Remarks to the Author):

This manuscript investigates the function of Klf5 in prostate organogenesis using a series of cell line studies, mouse models, and organoid studies. The authors use mutant forms of Klf5 to conclude that deacetylated Klf5 is required for maintenance of basal progenitor cells, whereas acetylated Klf5 is necessary for luminal differentiation and regeneration.

While this topic is interesting and the results are of potential significance, this study has important flaws in its rationale and experimental methodology. Notably, the authors seem to propagate misconceptions about the prior literature on prostate basal cells and its interpretation. The authors imply that postnatal prostate development is largely mediated by multipotent basal progenitors (e.g., lines 509-510), but ignore the contribution of unipotent luminal progenitors. However, at the time point used for tamoxifen induction (3 weeks) in the mouse studies, most luminal cells arise from luminal progenitors, not basal progenitors; the authors never examine a different time point in their studies. Moreover, the authors often use the term "basal progenitor" when simply referring to basal cells, and use the term "progenitor" when discussing experiments with cell lines, which seems inappropriate. Furthermore, the authors employ both sphere formation and organoid formation assays, which is confusing, and do not discuss the differences between these assays with respect to basal and luminal differentiation under the conditions used.

In general, it is extremely difficult to understand the design of the experiments performed in this study. One major issue is that the authors never state the exact genotypes of the mice being analyzed, and instead employ a non-standard genetic nomenclature (e.g., what do W/W, F/W, F/F, KR/W, and F/FK358R mean?). These notations also often imply that Cre-mediated recombination is complete within the cells being analyzed, which should not be assumed. Furthermore, the manuscript is densely written and hard to follow; for example, the relationship between experiments shown in the supplemental figures and those in the main figures is often unclear. Finally, the authors have made little attempt to convey the broader significance of their findings.

Another major issue is that the authors never state the number of mice/samples analyzed, rarely

provide raw data for their quantitation and statistical analyses, and do not provide actual p-values. This is of particular concern since several key results are not particularly convincing in the absence of quantitation (e.g., Fig. 3G), whereas seemingly large differences are deemed as not significant (e.g., Fig. S5H).

Specific comments:

1. Fig. 1D, E: This lineage-tracing experiment is missing basic controls and information. The authors should examine a time point shortly after tamoxifen administration to show and quantitate the basal cell type specificity of marking. What is the efficiency of recombination? How many mice were analyzed?

2. Fig. 1L-N: The reported percentages of Ki67-positive luminal and basal cells in what should be phenotypically wild-type (W/W) mice at 8 weeks of age seem extremely high (20 and 15%, respectively). These images are also inconsistent with the W/W image shown in Fig. 3E.

3. Fig. S1J,K: The authors should show histology and immunostaining of these organoids to show any alterations in basal and/or luminal differentiation.

3. Lines 187-188: There seems to be little basis for the statement about "grouped YFP+ luminal cells" that "Each such group/unit should have been primarily derived from a single basal progenitor cell." Why couldn't grouped YFP+ luminal cells have been derived from distinct basal progenitors? Is there a difference between groups and units? The explanation for how analysis of YFP+ luminal cells and luminal units allows quantitative analyses of the ability of basal progenitors to form luminal cells is unclear.

4. Lines 231-232: The authors suggest that "these findings also suggest that acetylation of KLF5 could shift KLF5 function from self-renewal of self/progenitor cells to luminal differentiation", but this seems overly speculative given that the sphere formation assay does not assess luminal differentiation. Notably, AR expression is uniform and cytoplasmic.

5. Fig. 3A: The histology of the KR/KR prostate is not noticeably different from the controls. However, the quality of this image is poor, as the section shown at higher-power is torn.

6. Fig. 3H,I: The reported increase in secretory proteins in the KR/KR prostate seems to differ from the histology shown in Fig. 3A, and quantitation of staining intensity is not an appropriate assay to support this conclusion. The authors should perform Western blotting of prostate secretions to address this issue. If overall luminal areas are decreased, there might be an increased density of secretory proteins, as opposed to increased secretions.

7: Fig. 3L: The KR/KR organoid image shown is quite unusual, as it seems to have basal cells in the interior and luminal cells on the exterior. The authors should examine additional markers to support their conclusion that luminal differentiation is enhanced in the mutant.

8: Fig. 4E: The CK18 immunostaining is quite heterogeneous in these images. Could this reflect defective luminal differentiation? The authors should examine other luminal markers to address this possibility.

9. Fig. 4G-J: If luminal proliferation is increased in the F/FK358R prostates, how can the authors conclude that there is increased basal differentiation into luminal cells? How can the authors be confident that "luminal units" arise from single basal progenitors?

10. Fig. 4M: Why is the AR staining in these organoids cytoplasmic?

11. Fig. 5E: These images are difficult to understand. Since the YFP should be expressed uniformly in all cells, why are some organoids much brighter than others? The authors claim that the morphological differences between organoids are due to alterations in luminal apoptosis, yet they do not show histology of the organoids or examine apoptosis directly.

12. Lines 449-450: The authors conclude that "acetylation of Klf5 is required for the generation and/or maintenance of most castration-resistant luminal cells in the prostate", but have only examined the luminal cells that derive from basal progenitors during organogenesis, which is a minority of the luminal population. The authors should use other Cre drivers to investigate this issue.

13. Fig. 7A-D: The authors show data for YFP+ cells in the regeneration experiment shown, but should also show data for YFP- cells as an internal control.

Point-by-point responses to reviewers' comments

Manuscript NCOMMS-18-35469-T, by Zhang et al., entitled “*Klf5 and its acetylation control the amplification and luminal differentiation of prostatic basal progenitor cells*”

In order to fully address each of the comments from reviewers, we have performed a substantial number of new experiments. Below we address the reviewers' comments one by one.

Reviewer #1 (Remarks to the Author): J. Michael Ruppert, West Virginia University

The manuscript by Zhang and colleagues, “Klf5 and its acetylation control the amplification and luminal differentiation of prostatic basal progenitor cells” investigates the roles of acetylated (Ac) vs de-acetylated (deAc) Klf5 in prostate gland epithelial homeostasis. Using both in vitro (spheroid) and in vivo (genetically modified, constitutive or conditional mice) the paper first demonstrates a critical role of Klf5 for proliferation in the prostatic basal epithelial compartment, and then goes on to analyze deAc-Klf5 (Klf5-KR) for effects on basal and luminal proliferation, differentiation and/or recovery from castration +/- androgen therapy. Overall excellent work and very interesting. This is a highly informative and compelling paper but with three major problems as described below.

Major problems:

1) Lack of any evidence that lys-arg substitution (KR) actually mimics unAc-Klf5, or that lys-gln substitution (KQ) really mimics Ac-Klf5. It seems unreasonable to assume that these substitutions are necessarily informative about acetylation of the relevant lysine. For example, they could instead represent loss- or gain-of-function alleles with no relevance to acetylation at all. For example, not all S-D or T-E substitutions necessarily mimic S/T phosphorylation. Additional evidence could include subcellular localization studies, ChIP-seq analysis (differential localization on chromatin), protein stability or transcriptional activity. Analysis of other substitutions such as KA might help with arguments that KQ is specifically mimicking Ac-Klf5 or that KR is specifically mimicking deAc-Klf5. Or a HAT inhibitor might be used to phenocopy deAc-Klf5 in the spheroid assay.

Response: Previous studies from many groups including our own have established the lys-arg substitution (KR) as a valid mutation eliminating posttranslational modifications of lysines in KLF5, and that lys-gln substitution (KQ) mimics acetylated lysine (Ac-K), including the acetylation of KLF5 (Ac-KLF5). For example, Dr. Ryozo Nagai's group reported that the coactivator/acetylase p300 acetylated KLF5 at K369 (which is homologous to K358 in mouse Klf5), and the lys-arg mutation (K369R) decreased KLF5's transcriptional activities in PDGFA expression and attenuated its ability to promote cell proliferation¹. In our previous study, while wildtype KLF5 interacted with the p300 acetylase, the K369R mutant prevented this interaction but induced the interaction of KLF5 with other factors including MYC²; and the K369R mutation affected the binding of KLF5 to *CDKN2B* and *MYC* promoters and reversed the regulatory function of KLF5 in cell proliferation and tumor growth²⁻⁵, which we have cited in the fourth paragraph of the Introduction. Similarly, we also reported that the lys-gln (K369Q) mutant was

less susceptible to protein degradation than the K369R mutant⁶. More directly, we have found that antibody against acetylated KLF5 at K369 did not react to the K369R mutant (new Fig. S4E),

[Redacted]

In our unpublished data from another manuscript in preparation in which the acetylation of KLF5 was examined for its role in bone metastasis of prostate cancer, we isolated protein complexes for both KR and KQ, and performed mass spec analysis. We found that KR and KQ mutants formed distinct transcriptional complexes in the DU 145 prostate cancer cells (Response Fig. 1); and RNA-Seq and ChIP-Seq analyses of the same cells further indicate that KR and KQ mutants regulate different sets of genes (Response Fig. 2A) and have different bindings at least to some gene promoters (Response Fig. 2B).

Nevertheless, we took the reviewer's suggestion and constructed a KA mutant, and tested whether KQ and KA mutants have similar effects on organoid formation in mouse prostate basal cells (new Fig. 5E-5F). The new findings demonstrated similar promoting effects of KQ and KA mutants on the maintenance of basal cells in organoids, while in contrast the KR mutant gave rise to more luminal cells, providing additional evidence for the notion that KQ mimics Ac-Klf5 and KR mimics deAc-Klf5 in basal to luminal differentiation. These new findings are described in the Results section of the revised manuscript (Line 374 on page 11 in the revised manuscript) as follows: "To further test the role of KLF5 acetylation in the progenitor capacity of basal cells, we expressed wildtype $KLF5$ ($KLF5^{WT}$), acetylation-deficient mutant $KLF5^{K369R}$ ($KLF5^{KR}$), and acetylation-mimicking mutants $KLF5^{K369Q}$ ($KLF5^{KQ}$) and $KLF5^{K369A}$ ($KLF5^{KA}$) in YFP+ and Klf5-null basal cells isolated from $p63^{CreERT2/+};R26R^{YFP/+};Klf5^{-/-}$ mouse prostates immediately after tamoxifen administration. $KLF5^{WT}$ and $KLF5^{KR}$ gave rise to more YFP+ organoids regardless of DHT conditions, while $KLF5^{KQ}$ and $KLF5^{KA}$ did not have such an effect (Fig. 5E), suggesting that deacetylation of KLF5 promotes organoid formation of basal cells. Furthermore, expression of $KLF5^{WT}$ restored normal differentiation of basal cells, as suggested by cystic organoids where basal cells (CK5+) resided around the outer layers and luminal cells (CK8+) orientated towards the lumen (Fig. 5F). DHT increased luminal cells (CK8+) in the organoids with $KLF5^{WT}$ (Fig. 5F, 5G). The $KLF5^{KR}$ mutant led to organoids with many more luminal cells (CK8+); while in contrast, $KLF5^{KQ}$ and $KLF5^{KA}$ mutants led to organoids with more basal cells (CK5+) (Fig. 5F, 5G). These results further support roles of KLF5 and its acetylation in basal progenitor maintenance and basal to luminal differentiation, with acetylated KLF5 retaining basal features but deacetylated KLF5 leading to the differentiation of excess basal cells to luminal cells."

2) Lack of evidence that anti Ac/unAc-Klf5 differentially stains basal and luminal layers in the prostate. Examination of the PLOS One reference provided (see Figs. 3D and 3E) revealed that the study was somewhat preliminary and lacking in several controls. Thus it is unclear whether or how acetylation of Klf5 is changing during differentiation in the prostate, a major gap for the paper.

Response: We agree with the reviewer that the previous study published in *PLoS One* should have included more controls. To address this concern, we have stained both human and mouse prostate specimens using the Ac-KLF5 specific antibody and the antibody that detects total KLF5.

We first validated the antibodies against Ac-KLF5 and total KLF5 in two pairs of control samples (Response Fig. 3). One pair included HaCaT cells before and after TGF- β treatment, since it was established in a previous study that TGF- β treated HaCaT cells express a much higher level of

Ac-KLF5³ and thus can be used as a positive control for staining Ac-KLF5. The other pair included DU 145 cells in which *KLF5* has been deleted by CRISPR-Cas9 system, which can thus serve as a perfect negative control for the KLF5 sample. KLF5-expressing DU 145 cells then served as a positive control for KLF5 expression. Rabbit IgG was used as an isotype control. Considering that IF staining of Ac-KLF5 must be performed in Tris-EDTA buffer for antigen unmasking, and the staining of total KLF5 on the other hand needs to be in a citrate buffer, IgG controls were performed in both buffers.

As expected, total KLF5 was expressed in HaCaT cells, TGF- β

treated HaCaT cells, and KLF5-expressing DU 145 cells but not in KLF5-null DU 145 cells. The expression of Ac-KLF5 was detected in TGF- β treated HaCaT cells and KLF5-expressing DU 145 cells, but not in parental HaCaT cells or KLF5-null DU 145 cells (Response Fig. 3). Consistent with our studies in the manuscript, these expression patterns validate the use of these antibodies in the detection of total KLF5 and Ac-KLF5 in prostate specimens.

We also used a previously developed antibody to detect deAc-KLF5 by IF staining, but the detection was not successful as the signal was weak and inconsistent.

Therefore, detection of total KLF5 and Ac-KLF5 was performed in normal prostate of both human and mouse to assess their expression patterns in basal and luminal layers (new Fig. 1, see *next page*). These results are now described in the Results section of the revised manuscript (Line 139 on page 6 in the revised manuscript), as below: “To understand the roles of KLF5 and its acetylation in prostate development, we evaluated their expression patterns in prostates by costaining total KLF5 and acetylated KLF5 (Ac-KLF5) with basal cell marker p63 and luminal cell marker CK8 in human (Fig. 1A) and mouse (Fig. 1C) prostate tissues. In human prostates, total KLF5 was detected in about 62% and Ac-KLF5 in 38% of basal cells. In contrast, total KLF5 was expressed in about 50% and Ac-KLF5 only in about 15% of luminal cells (Fig. 1B). In mouse prostates, while 68% and 53% of basal cells expressed total KLF5 and Ac-KLF5, respectively (Fig. 1D), 70% and 27% of luminal cells expressed total KLF5 and Ac-KLF5, respectively (Fig. 1D). Therefore, whereas total KLF5 is expressed in both basal and luminal cells, Ac-KLF5 is more commonly expressed in basal cells than in luminal cells (one-fold difference).”

3) Recurrent, selective interpretation and over-interpretation of the data, often excluding reasonable alternative interpretations. Effects on luminal cell biology are indirectly inferred following knockout in basal cells. The data is otherwise interesting and compelling.

Response: We appreciate Dr. Ruppert's comment. Our study was intended to focus on the roles of Klf5 and its acetylation in basal progenitor cells, including their amplification and luminal differentiation. Accordingly, for most of the key experiments we used the p63-Cre mouse strain, in which Cre expression is activated in basal progenitor cells. The reviewer is thus correct that the knockin was introduced into basal cells. We have extensively revised the manuscript to describe and interpret our data more accurately, especially emphasizing the fact that both *Klf5* deletion and *Klf5*^{KR} knockin were introduced into basal progenitors.

Regarding whether the effects on luminal cell biology are indirectly inferred, we must emphasize that the current study focuses on the differentiation of a basal progenitor cell into a luminal cell, as defined by mutating *Klf5* in basal progenitor cells and investigating the subsequent alterations in luminal cells that have arisen from such basal progenitors. Although both basal and luminal cells proliferate, differentiation is indispensable for a basal cell to become a luminal cell, which is well established in the field. Therefore, for each cluster of YFP+ luminal cells, at least one of the cells must have originated from a basal progenitor cell via differentiation. As we describe below, we used YFP+ luminal units to exclude the effects of

basal amplification and luminal proliferation on the number of differentiated YFP+ luminal cells, and the definition of a YFP+ luminal unit has been clarified in the revised manuscript (Line 326 on page 10 in the revised manuscript; see more specific explanations in the response to specific comment 1 as below). We have provided more responses to several specific concerns in this regard, including Specific comments 1, 7, 10, 19, 21, 25, 26 and 27 from the reviewer.

We agree that we certainly need to clarify some points, and we have revised the manuscript accordingly, but the conclusion that Klf5 deacetylation affects luminal cell differentiation from basal cells is well supported. We have clarified and corrected points as needed following this reviewer's specific comments below. We also appreciate the reviewer's comment that "*the data is otherwise interesting and compelling*".

Other Specific Comments:

1. The papers shows a clear role of Klf5 in basal layer of prostate, but conclusions about the role of Klf5 in luminal cells are all based upon indirect evidence, ignoring the possibility of epiphenomena related to alteration of the basal compartment. This is especially worrisome because effects in spheroids were different from effects in the mouse prostate (eg, data on the KR allele). There were no experiments that directly targeted luminal cells and no data indicating differential expression or activity of Ac/unAc-Klf5 in luminal cells. Therefore, conclusion of a role for Klf5 in luminal cells is not warranted, despite repeated over-interpretations in this area. See for example line 29 (Abstract).

Response: It is true that we did not directly engineer *Klf5* in luminal cells and luminal cells can divide to produce more luminal cells, and therefore we cannot directly conclude a role for Klf5 in luminal cells. The premise of this study is the role of Klf5 and its acetylation in basal progenitor cells and those luminal cells that have originated from such basal progenitor cells, i.e., basal progenitor-derived luminal cells, because *Klf5* was engineered in p63+ basal cells, which give rise to both basal and luminal cells. We realized that we were not specific enough when using the term "luminal cells" in the original version, which, when applied to our findings and conclusions, really refers to the specific subpopulation of luminal cells now named "basal progenitor-derived luminal cells". Nevertheless, there are two aspects that we'd like to clarify regarding to this comment.

The first aspect relates to the differentiation of basal cells into luminal cells. We demonstrated that loss of Klf5 in basal progenitors decreased the number of luminal cells (new Fig. 2G) and deacetylation at Klf5 had the opposite effect (new Fig. 4G). The key concern of this comment is whether we can conclude that Klf5 plays a role in the differentiation of basal progenitor cells to luminal cells based on changes in the number of luminal cells. Below, we outline how this conclusion was drawn:

The number of YFP+ luminal cells can be influenced by abnormalities in 3 processes: (I) proliferation (or amplification) of basal progenitors; (II) differentiation of basal progenitors to luminal cells; and (III) proliferation of luminal cells.

For processes I and III, we introduce several terms: “YFP+ luminal units”, “YFP+ luminal cluster”, and “single YFP+ luminal cell not in a cluster”. A YFP+ luminal cluster is a cluster of 2 or more adjacent YFP+ luminal cells. Each YFP+ luminal cluster should have originated from either the differentiation of a YFP+ basal progenitor cluster (which originated from a single originally YFP-labeled basal progenitor via amplification), the proliferation of a differentiated YFP+ luminal cell, or the combination of these two processes. Each single YFP+ luminal cell not adjacent to any other YFP+ cells should have arisen from a single basal progenitor cell via basal to luminal differentiation, since luminal cells normally do not migrate horizontally due to the tight junctions between them⁷. Therefore, the use of YFP+ luminal units, which include both YFP+ luminal clusters and YFP+ luminal single cells, can exclude the effect of basal cell amplification (process I) and luminal cell proliferation (process III) on the analysis when investigating the role of Klf5 and its acetylation in the differentiation of YFP-labeled basal progenitors to YFP+ luminal cells. Notably, we did not observe any two or more adjacent basal cells that were labeled with YFP immediately after tamoxifen administration (new Fig. S2C, S6A), which indicates that each of the YFP+ basal clusters should have originated from a distinct basal progenitor that is not adjacent to other such cells. Therefore, use of YFP+ luminal units also excludes the impact of YFP+ basal clusters on the analysis. Consequently, the decrease and increase in YFP+ luminal units by loss of Klf5 and deacetylation at Klf5, respectively, should indicate a role of Klf5 and its acetylation in basal to luminal differentiation.

For processes I and III, we have analyzed the proliferation rates of both basal and luminal cells by staining the Ki67 marker (new Fig. 2K, 4M, S6D), which directly indicates changes in cell proliferation caused by the loss or deacetylation of Klf5. Deacetylation of Klf5 did not increase the number of basal cells (new Fig. 4D) or the Ki67 index in basal cells (new Fig. S6D), which excludes any contribution of basal cell proliferation to increased YFP+ luminal cells and units after the knockin of Klf5^{KR}.

In the revised manuscript, we have removed the differentiation data from RWPE-1 cells, since sphere formation of RWPE-1 cells cannot faithfully recapitulate the process of basal to luminal differentiation (This question has also been addressed in our response to Specific comments 6 and 15 as below). As a substitute, we employed an organoid formation assay, in which basal progenitor cells can faithfully differentiate to luminal cells. In this assay, KLF5^{KR} mutation led to more luminal cells in the organoids whereas KLF5^{KQ} and KLF5^{KA} mutants maintained basal marker expression (new Fig. 5F, 5G). These findings are consistent with the *in vivo* data.

In the revised manuscript, we have provided another piece of data to further support a role of Klf5 acetylation in basal to luminal differentiation, i.e., deacetylation of Klf5 disrupted the typical cystic structures in androgen-induced organoids.

The second aspect of this set of comments relates to “basal progenitor-derived luminal cells”, which refer to those YFP+ luminal cells in our study. In our system, Cre expression was induced only in p63+ basal cells, so any YFP+ luminal cells must have originated from basal cells. Considering that luminal cells can also arise from luminal stem progenitor cells, it is necessary

to clarify that YFP+ luminal cells in our study refer only to “basal progenitor-derived luminal cells”. We have made this clarification throughout the manuscript.

Therefore, alterations in both castration response and androgen-mediated regeneration were all limited to this specific population of luminal cells – “basal progenitor-derived luminal cells”. Our findings of different responses to androgen for these cells thus also indicate a role of Klf5 acetylation in basal to luminal differentiation.

Accordingly, we have revised the manuscript extensively to clarify the definition of “YFP+ luminal units”, explain how we draw a conclusion regarding the basal to luminal differentiation, emphasizing that our findings in luminal cells apply only to “basal progenitor-derived luminal cells”, and clarify that our study applies specifically to basal to luminal differentiation but not to luminal to luminal proliferation or differentiation. The manuscript has been extensively revised in this regard.

Taken together, our conclusion of Klf5 and its acetylation modulating basal to luminal differentiation is supported by multiple lines of direct evidence. The possibility of the alteration in the basal compartment has been well considered.

With regard to knocking in *Klf5*^{K358R} in luminal cells, we argue that such a design falls to a different premise, i.e., the role of Klf5 and its acetylation in luminal cells (proliferation and differentiation). This premise is totally different from that in the current study, i.e., the role of Klf5 and its acetylation in basal progenitor cells (proliferation and luminal differentiation). Such a design would not help to address our questions in this study, because luminal cells, including those from basal cells, are no longer basal cells. Nevertheless, the knockin of *Klf5*^{K358R} in luminal cells would address a different set of interesting questions, and thus should be worth pursuing in a different study.

2. Line 31 of the Abstract: “the function of Klf5 in progenitor cell amplification was restricted to deacetylated Klf5 (deAc-Klf5)”. This statement does not seem to accurately reflect most of the data in the paper, which show that deAc-Klf5 (the KR allele) confers slow maturation of the prostate and also promotes basal to luminal differentiation.

Response: We have changed this statement to “Additionally, acetylation of Klf5 was essential for the maintenance and proper luminal differentiation of basal progenitors, as deacetylation of Klf5 caused excess basal-to-luminal differentiation; attenuated androgen-mediated organoid organization; and retarded postnatal development of the prostate.”(Line 36 on page 2 in the revised manuscript)

3. Line 98: “whether Klf5 plays a role in lineage determination of epithelial cells has not been studied”. This broad statement seems potentially problematic and should be double checked for accuracy. Klf5 has clear roles in epithelial contexts such as corneal epithelium, mammary gland and perhaps in the gut.

Response: We agree with the reviewer in the sense that the role of Klf5 in epithelial proliferation has been reported in corneal epithelium, the mammary gland, and the gut. However, studying the role of Klf5 in lineage specification from progenitors to luminal cells by using a lineage tracing system is still lacking. Nevertheless, we have changed the sentence to “Whether and how Klf5 plays a role in lineage determination of prostate epithelial cells has not been studied.”(Line 111 on page 5 in the revised manuscript)

4. Line 104: *Ensure gene and protein nomenclature compliant with journal guidelines eg p15 vs CDKN2B.*

Response: We have corrected these accordingly. “Biochemically, acetylation alters the transcriptional complex of KLF5 and the expression of its downstream target genes including CDKN2B, MYC, PDGFA, CDKN1A, etc.” (Line 116 on page 6 in the revised manuscript)

5. Fig 1A: *need to repeat the panel on right side. The Klf5 immunoblot didn't work very well as indicated by weak or no specific signal in control cells (comparing K2 to C2 and C4, all appear to have some positive signal); not clear why leftmost panel is necessary if the blot is successfully repeated (i.e., delete lanes 1-5).*

Response: We have repeated the experiment for the KLF5 immunoblot on the right side (new Fig. 2A). The leftmost panel is necessary because it contains different clones.

6. Fig. 1A: *Lack of any effect on CK18 in the RWPE-1 model is perhaps inconsistent with a role for Klf5 in luminal differentiation, as interpreted below from Fig. 1D and Fig. 1E data in the p63-Cre-ERT2 model. Although in vivo data should trump negative results in vitro, this apparent inconsistency is never addressed.*

Response: Lack of an effect on CK18 in the RWPE1 model could be attributed to the fact that cells used for Western blotting were from 2-D culture, in which RWPE1 cells did not undergo differentiation. Moreover, expression of the luminal marker CK18 in RWPE-1 cells indicates that this cell line is not a faithful *in vitro* model for basal cells and for mimicking basal to luminal differentiation. More importantly, RWPE-1 cells were unable to differentiate to luminal cells completely in the sphere formation assay, as suggested by Reviewer #3 and indicated by cytoplasmic location of AR even in DHT-containing medium. Collectively, this cell line is not a reliable model for conclusions regarding basal to luminal differentiation. The sphere formation efficiency should only suggest progenitor capability.

7. *It is shown clearly in Fig. 1 that loss of Klf5 reduces both basal and luminal cell populations. However, to make the case that Klf5 is necessary for the proliferation of basal cells, subsequently their differentiation to luminal cells, and finally proliferation of these luminal Klf5 deficient cells, it would seem critical to complement the p63-Cre-ERT2 data with a luminal promoter-Cre driver. The current analysis suffers from overinterpretation and ignores possible indirect effects on luminal populations when the basal compartment is compromised by Klf5*

deletion. A simple solution here might be to simply re-work the conclusion type statements in the paper, with modification of the title and abstract.

Response: We appreciate the suggestions. As described in the specific comment 1 above, the premise of our study is the roles of Klf5 and its acetylation in basal progenitor cells, including their proliferation/amplification and differentiation of luminal cells. It has been established that multipotent basal cells can proliferate and differentiate into 3 lineages of cells: basal, luminal and neuroendocrine cells. While both basal and luminal cells can proliferate to produce more basal and luminal cells respectively, differentiation is indispensable for a basal cell to become a luminal cell. Based on this principle and the fact that loss of Klf5 only occurred in basal cells due to the basal cell-specific p63 promoter, alterations in luminal cells caused by Klf5 knockout must have been involved in the basal to luminal differentiation (as we explained in the response to specific concern 1 above, *page 6* in this file). We also explain below how we reached our conclusion (Line 195 on page 7 in the revised manuscript):

“A basal progenitor can divide to increase the pool of basal cells, a basal cell can differentiate to become a luminal cell, and a luminal cell can divide to produce more luminal cells^{8,9}. Therefore, the decrease in YFP+ luminal cells by Klf5 deletion in basal progenitors could be attributed to reduced amplification of basal progenitors, interruption of basal to luminal differentiation, and/or compromised luminal cell division. To clarify which of these processes is affected by *Klf5* loss, we analyzed YFP+ luminal clusters, each of which contains at least 2 adjacent YFP+/CK18+ cells. Each cluster should have been primarily derived from a single YFP-labeled p63+ basal progenitor, by the amplification and subsequent differentiation of a basal progenitor and/or division of a luminal cell that had differentiated from a basal progenitor. In either case, basal to luminal differentiation must occur, and a cluster of YFP+ luminal cells can then be considered as a unit to better indicate the differentiation, minimizing the effect of basal or luminal cell proliferation on the analysis⁸. In addition, a single YFP+ luminal cell not in a cluster, which should have also arisen from the differentiation of a basal progenitor cell because luminal cells normally do not migrate horizontally due to their tight junctions⁸, was also considered as a unit in the analysis. Interestingly, the absence of Klf5 significantly decreased the number of YFP+ luminal units (Fig. 2H), supporting a role of Klf5 in the differentiation of basal to luminal cells.” On the other hand, use of a luminal cell-specific Cre would not address the question of Klf5 and its acetylation in basal progenitors, because luminal cells are already beyond the basal to luminal differentiation. A luminal cell-specific Cre can address a different set of questions and has its own merit though.

8. Data in Figs. 1D and 1E appear consistent with the notion that Klf5-deficient basal cells never make it to the luminal compartment at all. This might be an interesting result to pursue. Note that not every YFP+ cell is necessarily Klf5-deficient, so an occasional YFP+ luminal cell is still consistent with failure of Klf5-deficient cells to differentiate.

Response: We appreciate this positive comment.

9. Fig. 1: for this figure DP and AP are apparently never defined as dorsal and anterior prostate, respectively. AP and DLP are however, defined in Fig. 6 legend.

Response: We have corrected the figure legends according to the comment (Line 994 on page 28 in the revised manuscript).

10. Line 184 and Fig. 1H and 1I: Groups of YFP+ luminal cells, apparently used here to measure proliferation of luminal cells, could also derive from cell division of YFP+ basal cells, followed by their differentiation. Because basal proliferation is already clearly suppressed by deletion of *Klf5*, the conclusion that *Klf5* is also impacting cell division of luminal cells is not well supported (no direct evidence). The counting of YFP+ luminal cell units does not appear to add much to the story and could be deleted.

Response: We have deleted old Fig. 1H, the “YFP+ group” data, and its description from the Results section, considering that the YFP+ group could not accurately measure the proliferation of luminal cells. However, we would like to keep old Fig. 1I (new Fig. 2H), the “YFP+ luminal unit” data. As we describe and discuss above in our responses to specific comment 1, each “YFP+ luminal unit” is defined as a cluster of 2 or more adjacent YFP+/CK18+ luminal cells or a single YFP+/CK18+ luminal cell. The purpose of defining and using “YFP+ luminal unit” is to exclude the contribution of basal amplification and luminal proliferation to the number of YFP+ luminal cells. At least one YFP+ luminal cell in a YFP+ luminal cluster has come from the original YFP-labeled basal progenitor cell via differentiation. The rest of YFP+ luminal cells within a unit most likely came from both the proliferation of existing YFP+ luminal cells via division and the proliferation of basal progenitor cells. Use of “YFP+ luminal unit” excludes those both “luminal cell-derived luminal cells” and “basal cell-derived basal cells” for their potential effects on the number of differentiated luminal cells from basal cells.

11. Fig. 1L, 1M and S2F showing selective loss of Ki67 in *Klf5*-deficient luminal cells: If *Ki67* is truly unaffected in the YFP negative cells in the same section (these are control *Klf5* wild type cells), then make this critical case within Figure 1 and not in the supplement Fig. 2F. Lack of statistical significance of *Ki67* “without YFP tracing” is not a relevant comparison, and is not a substitute for directly comparing *Ki67* in the YFP+ and YFP negative luminal cells of the exact same tissue section.

Response: We appreciate this suggestion. We have moved old Figure S2F to the main figure (currently new Fig. 2K). We have some interesting new findings that, in the wildtype group (*Klf5*^{+/+}), the percentages of Ki67+ cells in YFP+ basal cells and YFP+ luminal cells were higher than in YFP- cells with marginal significance, which suggests that the progenies of p63+ progenitors proliferated faster. Loss of *Klf5* suppressed the proliferative advantage of basal and luminal progenies of p63+ basal progenitors.

12. Legend vs Results for Fig. 2C: unlike in the Results section describing Fig. 2C, in the legend it is stated that *Klf5*-KQ (presumably

mimicking acetylation) "prevented" sphere formation. The Results section is worded more correctly.

Response: We thank the reviewer for the comments. We have corrected the figure legend to make it more accurate, as shown in the revised manuscript, "Sphere-forming capabilities of RWPE-1 cells were restored by KLF5, enhanced by KR (deAc-KLF5), but not improved by KQ (Ac-KLF5), as indicated by images (D) and numbers (E) of spheres." (Line 65 on page 5 in the *Supplementary information*)

13. Figs 2E and 2F could be switched as they are discussed out of order.

Response: We appreciate this comment. Currently, the old figure 2 has been moved to the supplementary data as Supplementary Figure 3. This change is also based on Reviewer #3's suggestion that "the sphere formation assay does not assess luminal differentiation", because the differentiation of luminal cells cannot be fully completed in the sphere assay by using RWPE-1 cells. In this regards, checking basal and luminal markers in RWPE-1 spheres is not meaningful. As a substitute, we overexpressed KLF5, KLF5^{KR}, KLF5^{KQ} and KLF5^{KA} in YFP+/Klf5-null basal cells isolated from mouse prostates of $p63^{CreERT2/+};R26R^{YFP/+};Klf5^{-/-}$ immediately after tamoxifen administration, and then performed organoid formation assays following the procedures established by Jarno Drost et al. from Dr. Yu Chen and Dr. Hans Clevers Labs. The differentiation of luminal cells was much better in organoids and these data are now presented as new Fig. 5E-5F (See *page 2* in this file).

14. Paragraphs beginning on lines 254 vs 265: in these two paragraphs the FK358R allele is described firstly as failing to express wt Klf5 during development, and then later as expressing wt Klf5 in adult tissues. This is confusing to this reader and is not well explained.

Response: We appreciate this comment and apologize for the confusion. Mice homozygous for the $Klf5^{LSL-K358R}$ (FK358R in the old manuscript) allele were not viable, which is likely due to the lack of Klf5 protein (Line 254 in the Old manuscript). However, we found that the $Klf5^{LSL-K358R}$ allele can still be successfully transcribed into wild type Klf5 mRNA (new Fig. S4D, middle). We have revised these two paragraphs by removing unnecessary information to avoid confusion, as shown in the revised manuscript (Line 239 on page 8). "Heterozygous $Klf5^{LSL-KR/+}$ mice were crossed to *Ella-Cre* mice, which express the Cre recombinase in the early mouse embryo and can thus activate the $Klf5^{K358R}$ (hereafter referred to as $Klf5^{KR}$) allele by removing the wild-type allele between the two loxP sites in a wide range of tissues including germ cells that transmit the transgene to progenies.

Expression of the $Klf5^{KR}$ allele was confirmed at both the mRNA and protein levels by sequencing and immunohistochemical (IHC) staining. The $Klf5^{KR}$ allele indeed expressed $Klf5^{KR}$ mutant mRNA, while the $Klf5^{LSL-K358R}$ allele expressed wild-type $Klf5$ mRNA at a comparable level (Fig. S4D). As expected, the $Klf5^{KR}$ allele also expressed Klf5 protein with a staining intensity comparable to wild-type Klf5, but the expression level of Ac-Klf5 in $Klf5^{KR/KR}$ mice was apparently decreased, as detected by IHC staining (Fig. S4E)."

15. Line 293: “the absence of *Klf5* acetylation causes precocious differentiation of basal cells to luminal cells”. This conclusion seems at odds with spheroid data in Fig 2E. In spheroid model the KR allele promotes a basal phenotype, basal cell proliferation and possibly a block to luminal differentiation, whereas in the mouse it promoted precocious differentiation into luminal cells. These contextual differences should be discussed and perhaps attributed to differences such as the expression level (RT-PCR or IHC data).

Response: We thank the reviewer for this important comment. We certainly need to clarify the inconsistency. The lineage tracing with p63-Cre (new Fig. 4C-I) provides a well-established system for basal progenitor cells to proliferate and complete their differentiation to luminal cells^{10, 11}. In this *in vivo* system, *Klf5*^{KR} mutant promoted basal progenitor differentiation to luminal cells. The *in vitro* spheroid assay using human RWPE-1 prostate cell line cannot be used for assessing luminal differentiation from basal cells, because RWPE-1 cells originally expressed both basal marker p63 and CK5 and luminal marker CK18, and the cells inside the prostate spheres were unable to completely differentiate into luminal cells, as commented by Reviewer #3. In our study, we also found that even the cells inside spheres with wild-type KLF5 did not have AR nuclear localization (old Fig. 2E), which suggests that such spheres have not yet completed their differentiation under the culture conditions. We cannot make conclusions about their relationship to luminal differentiation. Therefore, we have removed this IF staining data of this cell line and the related description in the manuscript. Thanks again for these critical comments.

As a substitute, we overexpressed KLF5, KLF5^{KR}, KLF5^{KQ} and KLF5^{KA} in YFP+/Klf5-null basal cells isolated from *p63*^{CreERT2/+}; *R26R*^{YFP/+}; *Klf5*^{-/-} mouse prostates immediately after tamoxifen administration and then performed an organoid formation assay, which provides a much more reliable model for mimicking the process of basal to luminal differentiation (new Fig. 5E-5G, See page 2 in this file). The results were actually consistent with that from RWPE-1 cells, as the number of organoids was increased by KLF5 and KLF5^{KR}, but not impacted by KLF5^{KQ} and KLF5^{KA} mutants. More interestingly, KLF5 gave rise to organoids with normal morphology where basal cells (CK5+) reside around the outer layers and luminal cells (CK8+) orientated toward the lumen. In addition, the KLF5^{KR} mutant led to organoids with many more luminal cells (CK8+); while in contrast, KLF5^{KQ} and KLF5^{KA} mutants led to organoids with more basal cells (CK5+) (new Fig. 5F, 5G). These results match the *in vivo* data quite well, which confirms that knockin of *Klf5*^{KR} produced more luminal cells. This line of data further supports a role of the KR mutant in the promotion of organoid formation by enhancing the basal to luminal differentiation.

16. If not already shown, staining the KR+ mouse tissues for total *Klf5* would be important for ascertaining whether it is expressed at wt levels.

Response: We have shown the data in old Fig. S3F (new Fig. S4E, See page 2 in this file).

17. Fig. 3 data is mostly describing disordered development by the constitutive KR allele and is not very revealing. This data could be moved to the supplement.

Response: We agree with this suggestion and have moved most of these results into new Fig. S5.

18. As prostate function was apparently normal (?) in the KR mice, Figure 3 argues against any essential role of deAc-Klf5 in prostate development.

Response: We appreciate the comment. These mice had constitutive knockin of KLF5^{K358R} throughout their bodies, and the “apparently normal” prostates were more likely due to a compensatory mechanism during embryonic development. Most likely they are not related to roles of Klf5 and its acetylation in postnatal development of the prostate, so we have moved most of these data to the supplementary data section (new Fig. S5). Nevertheless, we noticed that prostates from these mice were significantly smaller, had less p63+ cells but formed larger spheres, which still indicates an impact of KLF5^{K358R} on tissue homeostasis.

19. Line 310: Conclusion regarding "necessity of Klf5 acetylation for proper differentiation of basal cells to luminal cells during prostate development". The data in Fig. 3 doesn't necessarily implicate Klf5 function in luminal differentiation. Is it possible that forcing proliferation of basal cells leads to a compensatory increase in luminal differentiation?

Response: Basal cells can proliferate to produce more basal cells; and they can also differentiate into luminal cells. Luminal cells can also proliferate to produce more luminal cells. First of all, we demonstrated that the number and proliferation rate of basal cells were not significantly promoted by the KLF5^{K358R} knockin (new Fig. 4D and S6D). Secondly, we used YFP+ luminal units, which include both a cluster of 2 or more adjacent YFP+/CK18+ luminal cells and a single YFP+/CK18+ luminal cell, to exclude the effect of basal and luminal cell proliferation on our analysis (see our response to specific comment 1 above for more details on *page 6* in this file). More luminal units in the KLF5^{K358R} group indicates enhanced basal to luminal differentiation. Therefore, we can still draw the conclusion that K358R knockin promoted luminal differentiation from basal cells. Also, as described in the revised manuscript and our response above (specific comment 1), we now have the organoid data that further indicates the function of Klf5 acetylation in basal to luminal differentiation.

20. Line 312: without further development this paragraph could be deleted.

Response: This is also a piece of evidence for the role of KLF5^{K358R} knockin in mouse prostates, although it is not directly associated with postnatal development of the prostate. We have deleted this paragraph and the related results according to the reviewer’s suggestion.

21. Line 351 (re Fig. 4G) , this conclusion is not supported by the data, it could just as easily be that KR knockin promotes basal proliferation which then leads to enhanced luminal YFP+ groups as observed.

Response: This issue has also been addressed above in our responses to specific comment 1 (*Page 6* in this file). We showed that the *Klf5*^{KR} knockin did not promote the number

and proliferation rate of basal cells (new Fig. 4D and S6D), which excludes the possible contribution of basal cell proliferation to enhanced luminal YFP+ groups/clusters. Nevertheless, we have thoroughly revised the manuscript to address the definition and use of grouped YFP+ luminal cells (now called YFP+ luminal clusters) as below (Line 320 on page 10 in the revised manuscript): “To determine whether basal to luminal differentiation is accelerated by *Klf5*^{KR} knockin in prostates, we measured the number of YFP+ luminal clusters (Fig. 4H), which were defined as a cluster of 2 or more adjacent YFP+ luminal cells. Each of such clusters should have originated from either the differentiation of a YFP+ basal progenitor cluster, the proliferation of a single YFP+ luminal cell, or the combination of these two processes. A single YFP+ luminal cell that is not in a cluster should have originated from the differentiation of a single basal progenitor⁸. Considering each single YFP+ luminal cell or YFP+ luminal cluster as a YFP+ luminal unit, we used the number of YFP+ luminal units rather than YFP+ luminal cells to indicate basal to luminal differentiation, which minimizes the effect of cell proliferation on our analyses. By counting the number of YFP+ luminal units, we found that *Klf5*^{KR} knockin increased YFP+ luminal units by more than one-fold (Fig. 4I), indicating that interruption of *Klf5* acetylation promotes basal to luminal differentiation.”

22. Line 359: *“as indicated by more Ki67+ cells in YFP+ luminal cells but not in YFP- luminal cells (Figure 4I and 4J).” Examination of these figure panels reveals there is in fact no analysis of YFP- luminal cells illustrated, despite this indication in the Results text. This YFP- data is a critical missing control.*

Response: We appreciate this comment. We have added the data with YFP- cells as a control. Previously, the total group included both YFP+ and YFP- cells (new Fig. 4M).

23. Line 362: *“Collectively, these findings suggest that conditional knockin of KR in p63+ progenitors boosted their luminal differentiation program (Figure S5E), activated luminal cell proliferation (Figure 4G, 4I and 4J), and gave rise to more luminal cells during prostate development.” The conclusion is not well supported by the data. Alternatively, it seems equally possible that luminal cells with the KR allele have a hybrid phenotype with aspects of both basal and luminal cells. This dedifferentiated state might help to explain the delayed development and smaller size of the prostate, which otherwise seems inconsistent with the data.*

Response: As we discuss in our responses to other comments (See details in specific comment 1 on page 6 in this file), the conclusion is supported by our data, including new data from the organoid formation assay (new Fig. 5E-G, see page 2 in this file).

We also appreciate the thoughtful comment on the hybrid phenotype. We have tested this hypothesis by co-staining the prostates with both basal marker CK5 and luminal marker CK8 to focus on the hybrid phenotype (new Fig. 4J, 4K). Knockin of *Klf5*^{KR} did not affect the percentage

of YFP+ cells positive for both the CK5 basal marker and the CK8 luminal marker. Current data supports that acetylated KLF5 and deacetylated KLF5 play distinct roles in prostate development. Although the *Klf5^{KR}* knockin promoted luminal differentiation (new Fig. 5E-G, See page 2 in this file), it is likely that absence of Klf5 acetylation results in less basal progenitors that are maintained at the basal stage and thus cannot support the full process of prostate development.

24. Line 367, this paragraph is confusing. Change "KR knockin" to "conditional KR knockin". And maybe change "germline knockin" to "constitutive knockin".

Response: We appreciate the suggestion and have revised the manuscript accordingly as "After the *Klf5^{KR}* mutant was conditionally knocked in, the number of organoids was significantly increased regardless of DHT treatment (Fig. 5A)" (Line 360 on page 11 in the revised manuscript). In addition, all uses of "germline knockin" have been changed to "constitutive knockin".

25. Line 448. "The decrease by KR in castration-resistant luminal cells but not in basal cells indicates that acetylation of Klf5 is required for the generation and/or maintenance of most castration-resistant luminal cells in the prostate." The data has other possible conclusions. This data, once again, suffers from an indirect approach, wherein conditional genetic changes are induced in one cell type (basal cells) and the phenotype is concluded to indicate a direct role of Klf5 acetylation in another cell type (luminal cells). The approach suffers from the possible contribution of indirect effects of an altered basal compartment on luminal cells.

Response: This concern was raised several times in different contexts. In our study, we have monitored the number of basal cells before and after castration (new Fig. 4D, 7F), and the numbers of YFP+ basal cells were similar between the wildtype and the *Klf5^{KR}* knockin groups, which excludes the possible contribution of indirect effects of an altered basal compartment on luminal cells. In addition, the phrase "luminal cells" in this context only referred to "basal progenitor-derived luminal cells", which was less clear in the original version of the manuscript. Even for basal cells, their contribution was limited during castration, because castration mainly caused apoptosis of luminal cells. Considering that there were more YFP+ luminal cells in the *Klf5^{KR}* knockin group before castration (new Fig. 4G), we can conclude that acetylation of Klf5 is required for the generation and/or maintenance of most castration-resistant progenitor-derived luminal cells. We have revised the manuscript to make it more concise (Line 455 on page 13 in the revised manuscript): "acetylation of Klf5 is likely essential for basal progenitor-derived luminal cells to survive castration."

26. The paper relies upon powerful genetic methods to study the role of Klf5 in prostate basal cells and their differentiation into luminal cells. However the interpretation of function and causality repeatedly attempts to extend to luminal cell biology, even though genetic studies are entirely restricted to the basal layer (PB-Cre, p63-Cre-ERT2). Consequently it is difficult to discern direct vs indirect effects and much of the data can be interpreted in more than one way.

Response: The same concern has been raised earlier by this reviewer. As we have discussed in our response to Specific comment 1 above and as recognized by the reviewer, the premise of the study is the proliferation and differentiation of basal progenitor cells and the proliferation and castration response of basal progenitor-derived luminal cells. In particular, when implying our findings to luminal cells, our findings are limited to basal progenitor-derived luminal cells, which was not clearly defined and well clarified in our original manuscript.

Differentiation is an indispensable process for a basal cell to become a luminal cell. The number of basal progenitor-derived luminal cells is attributed to three processes: the amplification of basal cells, the differentiation of basal cells to luminal cells, and the proliferation of basal progenitor-derived luminal cells. Regarding the alterations in basal amplification and luminal proliferation, we have analyzed the proliferation marker Ki67 in both YFP+ basal and YFP+ luminal cells directly.

Regarding the basal to luminal differentiation, we have taken into account the basal amplification and luminal proliferation by counting “YFP+ luminal units”, which helps to exclude the contribution of basal and luminal proliferation, because each cluster of YFP+ luminal cells (considered as one unit) derives from the amplification of basal progenitors and the proliferation of one basal progenitor-derived luminal cell. Therefore, we can use change in YFP+ luminal units to conclude the effect of *Klf5*^{KR} knockin on basal to luminal differentiation.

27. In particular, there is the possibility of direct effects of deAc-Klf5 in basal cells leading to indirect effects on luminal cells that have little to do with Klf5 function. This is especially true as expression of deAc-Klf5 (Klf5-KR) in spheroids, the simpler model, indicates a select role in promotion of basal cell proliferation and perhaps a block to luminal differentiation. In contrast in the mouse model the KR allele (potentially mimicking deAc-Klf5) promotes luminal differentiation in context of a small gland that is slower to develop.

Response: First of all, our response to Specific comment 1 also applies to this comment. In addition, as discussed in our response to Specific comment 15, the sphere formation of RWPE-1 cells does not represent a good model for luminal differentiation, considering that even the wildtype spheres (old Fig. 2E) did not have nuclear localization of AR. Reviewer #3 also pointed out that the RWPE-1 model is not so helpful for addressing whether acetylation of Klf5 plays a role in the basal to luminal differentiation. Therefore, we have removed this IF staining data and the related description in the manuscript. We appreciate these critical comments.

As a substitute, we ectopically expressed KLF5^{WT}, KLF5^{KR}, KLF5^{KQ} and KLF5^{KA} in YFP+/Klf5-null basal cells isolated from mouse prostates of the *p63*^{CreERT2/+}; *R26R*^{YFP/+}; *Klf5*^{-/-} genotype immediately after tamoxifen administration, and then performed an organoid formation assay, which represents a much more reliable model for mimicking the process of basal to luminal differentiation (new Fig. 5E-5G, See page 2 in this file). Consistent with findings from RWPE-1 cells, the number of organoids was enhanced by KLF5 and KLF5^{KR}, but not improved by KLF5^{KQ} and KLF5^{KA}. More interestingly, KLF5 gave rise to organoids with normal morphologies, where basal cells (CK5+) reside around the outer layers and luminal cells (CK8+) orientate toward the

lumen. On the other hand, the KLF5^{KR} mutant gave rise to organoids with many more luminal cells (CK8+); and in contrast, the KLF5^{KQ} and KLF5^{KA} mutants generated organoids with more basal cells (CK5+) (new Fig. 5G, See *page 2* in this file). These results are consistent with the *in vivo* data, further indicating that knockin of the *Klf5*^{KR} allele produced more luminal cells. Therefore, it is likely that the KLF5^{KR} mutant promotes organoid formation by promoting basal to luminal differentiation. As mentioned in our response to Specific comment 15 above, our current data suggests that acetylated KLF5 and deacetylated KLF5 play distinct roles in prostate development. DeAc-Klf5 promotes basal to luminal differentiation, while Ac-Klf5 maintains basal progenitor cells at the basal stage. Although *Klf5*^{KR} knockin promotes luminal differentiation, it is likely that absence of acetylation at Klf5 results in fewer basal progenitors which are maintained at the basal stage and thus cannot support the full process of prostate development.

Another point raised again by the reviewer is that the effects of deAc-KLF5 in luminal cells could be disassociated with Klf5 acetylation, but could be a consequence of KR's effects on basal cells. As we have addressed above, we monitored the changes in basal cells (new Fig. 4D, 7F and 8C) at different time points, but found no significant differences in the number and proliferation rate of basal cells, which excludes the possibility raised by the reviewer.

28. Opportunities to utilize YFP-negative cells in the same section as a relevant control were repeatedly missed, further raising concerns about indirect effects.

Response: We thank the reviewer for this suggestion. In the original manuscript, we used total cells (including both YFP+ and YFP- cells) instead of YFP- cells as a negative control in our analyses (old Fig. S2F, S2G and S5H). We agree with the reviewer that YFP- cells represent a better negative control. Accordingly, we have added the numbers of YFP- cells in our analyses (new Fig. 4M, S6D, 8D). As shown in new Fig. 4M (See *page 15* in this file), the Ki67+ cells were significantly higher in YFP+ luminal cells with *Klf5*^{KR} knockin when compared to the YFP- cells in the same sections or YFP+ cells with wildtype *Klf5*. These results strengthen our conclusion that *Klf5*^{KR} knockin in basal progenitors results in more proliferative luminal cells.

29. Studies analyzing the role of deAc-Klf5 (whether KR represents deAc-Klf5 is not clear) in castration and regeneration seem descriptive and phenomenological.

Response: We appreciate this comment. These studies could be descriptive by nature, but they indicate the fate of Klf5^{KR}-expressing basal cells during castration and regeneration. The basal progenitor-derived luminal cells expressing Klf5^{KR} were almost completely eliminated during castration, while many more of those with wildtype *Klf5* survived castration. This finding clearly indicates a role of acetylation of Klf5 in the maintenance of castration-resistant luminal cells that originated from basal progenitors. This role could be involved in the development of castration-resistant prostate cancer, because TGF- β induces acetylation of KLF5 and drives castration resistant prostate cancer¹².

We wonder whether the surviving basal progenitor-derived luminal cells during castration could be, or at least overlap with, CARNs. We tested this speculation by detecting the expression of Nkx3.1 via IF staining, and tried three different antibodies against Nkx3.1. One was from Santa Cruz (#sc-393190), one from Sigma (#WH0004824M2), and the third one was from Dr. Michael M. Shen whose group established CARNs as luminal progenitors¹³. We also tried to enhance the signals using the tyramide amplification system¹⁴. Currently, we have successfully detected Nkx3.1 signals in intact prostates and regenerated prostates by using the antibody from Dr. Shen. The staining signals in castrated prostates were rare, and, without greater contrast, it is more challenging to obtain reproducible data (Response Fig. 4, See *next page*). We cannot make a conclusion at this stage. Moreover, we only have a limited amount of the antibodies from Dr. Shen and we cannot further address this question at this time. Considering that this question would be addressed by knocking in the *Klf5*^{KR} mutant directly into luminal cells with CK18-Cre, we would plan to address this question in a future study that will focus on the role of Klf5 and its acetylation entirely in luminal cells.

Nevertheless, we discussed our data in depth in the Discussion section in our revised manuscript (Line 601 on page 17 in the revised manuscript) as below: “Acetylation of Klf5 is essential for basal progenitor-derived luminal cells to survive castration, because interruption of Klf5 acetylation in basal progenitor cells sharply decreased the survival of their luminal progenies after mice were castrated (Fig. 7E). In the subsequent regeneration by restoring androgens, the reappearance of basal progenitor-derived luminal cells was again attenuated by the interruption of Klf5 acetylation (Fig. 8B), resulting in a drop in luminal cells from 13.6% before castration to 4.5% after androgen-mediated regeneration (Fig. S8F). While more than 90% of prostate luminal cells die after castration in mice^{15, 16}, some luminal cells are castration-resistant, including those defined as CARNs, which are able to give rise to three different lineages of epithelial cells and are thus considered luminal stem/progenitor cells¹³. It remains to be addressed whether the basal progenitor-derived luminal cells surviving after castration could overlap with CARNs. These findings also strongly suggest that Ac-Klf5 functions in castration resistance in basal progenitor-derived luminal cells. KLF5 acetylation may thus be required for the development of castration-resistant prostate cancer. Castration resistance is a common problem in the treatment of prostate cancer, and whether acetylation of Klf5 contributes to castration resistance in prostate cancer is thus a clinically relevant question worth addressing. In this regard, the mouse strain with the *Klf5*^{LSL-K358R} allele should prove helpful.

In the regenerated prostates with Klf5 deacetylation, three unexpected changes were detected. First, the ratio of YFP+ luminal cells in prostates with wild-type Klf5 was increased to about 11.5%, which was about 3-fold greater than that in prostates before regeneration (with or without castration) (Fig. S8F). Second, the ratio of YFP+ luminal cells in regenerated prostates with wild-type Klf5 was significantly higher than that with deAc-Klf5 (11.5% vs. 4.5%) (Fig. S8F), opposing the respective ratios in postnatal prostates. Third, interruption of Klf5 acetylation was much less effective in promoting basal to luminal differentiation in regenerated prostates than in postnatal prostates before castration (4.5% vs. 13.6% for YFP+ luminal cells). Taken together with the observations that basal cells are essentially not affected by castration, regeneration, or the deacetylation of Klf5 (Fig. 4D, 7F, 8C); both luminal and basal progenitors give rise to

luminal cells during the turnover of prostate epithelia; deacetylation of Klf5 promoted the differentiation of basal progenitors to luminal cells (Fig. 4G, S6H, 8F); and luminal cells with deAc-Klf5 were less resistant to castration (Fig. 7E), we propose that basal progenitors with deacetylation at Klf5 have different capabilities in the formation of luminal cells during postnatal development and the regeneration of prostates. The *Klf5*^{K358R} knockin mouse model should thus provide a unique tool for further studying prostate homeostasis and the role of Klf5 acetylation in diseases, including castration-resistant prostate cancer.”

30. Line 558 (Discussion):
 Although it is concluded based upon indirect evidence that Ac-Klf5 is indispensable for proper luminal cell formation, there were few experiments using the KQ allele (putative acetylation mimic).

Response: We appreciate this suggestion, which also relates to the reviewer’s concern that K>R mutation might not represent the deacetylation of Klf5 in Major concern 1. As we discussed earlier in our response to Major concern 1, there are a large number of published studies that indicate the acetylation of human KLF5 at K369 (K358R for mouse Klf5), and show that K>R mutation indeed abolishes the acetylation of Klf5. In the revised manuscript, we have added a study using both K>Q and K>A mutants (new Fig. 5E-5G, See page 2 in this file). Both K>Q and K>A mutants gave rise to more basal cells in organoids; but in contrast, the K>R mutant resulted in organoids with more luminal cells (new Fig. 5F, 5G). These data clearly indicate opposing functions of K>R and K>Q (and K>A) mutants in basal to luminal differentiation, addressing the issue raised by the reviewer.

31. The Discussion section seems very very long.

Response: We have thoroughly revised and re-written the Discussion. While removing undesired paragraphs and reducing the Discussion to about half of the original version, we also re-ordered the logical flow to strengthen our conclusions.

32. It should be possible to rewrite the paper without overstating conclusions re Ac/deAc-Klf5 function in luminal cells.

Response: We appreciate the comments. We have thoroughly revised/re-written the paper to avoid overstatements, especially paying extra attention to making sure that all of the conclusions are restricted to basal progenitors and their progenies.

Reviewer #2 (Remarks to the Author):

Using multiple mouse models as well as human prostate cell lines the author aimed in the present manuscript to investigate the role of Kruppel-like factor 5 (Klf5)- a basic transcription factor known to regulate multiple cellular processes including stemness/differentiation- in the development, differentiation and homeostasis of the prostate. The transcriptional activity of Klf5 is regulated by acetylation of lysine 358, thus the authors have generated several mouse strains including prostate-specific (conditional knockout), germline and p63+ basal cell-limited strains lacking Klf5 expression (knockout) or with acetylation-deficient Klf5 (by mutating lysine 358 to arginine) knockin. Most results have been generated by prostate in situ immunofluorescence stainings and ex vivo organoid culture of prostate-derived cells. The major findings of this manuscript are a requirement of Klf5 acetylation in luminal cells for proper prostate differentiation, while non-acetylated Klf5 is essential for basal progenitor characteristics. Furthermore, acetylated Klf5 could be important for castration resistance of luminal cells, a finding that could be of value for prostate cancer researchers. Mechanistically, a contribution of Notch signaling on basal-to-luminal differentiation was investigated in more detail.

The study was well designed, experiments were conducted well-controlled, and results are convincing and statistically significant. Work presented is novel and of high interest for researchers in the field of prostate development/differentiation, but also for cancer research. All in all, the study is worth being published, although a major concern and few recommendations need to be addressed:

Major concern:

Mouse and human prostate differ substantially in organ morphology and tissue architecture, although many but not all molecular features are similar. To increase the impact of this study it would therefore be worth to confirm whether the findings on mouse Klf5 acetylation are of relevance in the human prostate, too. This has been done only insufficiently in few experiments in the beginning of the study with two HPV-immortalized human prostate cell line/organoid models. The HPV oncoproteins E6/E7 are known for their ability to interfere with differentiation, thus interference with KLF5 signaling cannot be excluded. Relevance for KLF5 for human prostate differentiation could be addressed by using non-immortalized cells cultures obtained from normal areas of prostatectomy specimen to reproduce key cell line experiments. Expression analysis of acetylated KLF5 (along with unacetylated KLF5, basal/luminal markers, NKX3.1, etc...) in human prostate (cancer) specimen by (fluorescent) IHC would also greatly improve the impact of this study.

Response: We appreciate these comments. We have co-stained Ac-Klf5 and total KLF5 with basal marker CK5 or luminal marker CK8 in normal prostate samples of both human and mouse to assess the relevance of our findings in human prostate (new Fig. 1, See page 5 in this file). As we described in the first paragraph of the Results section in the revised manuscript (Line 142 on page 6), "In human prostates, total KLF5 was detected in about 62% and Ac-KLF5 in 38% of basal cells. In contrast, total KLF5 was expressed in about 50% and Ac-KLF5 only in about 15% of luminal cells (Fig. 1B). In mouse prostates, while 68% and 53% of basal cells expressed total

KLF5 and Ac-KLF5, respectively (Fig. 1D), 70% and 27% of luminal cells expressed total KLF5 and Ac-KLF5, respectively (Fig. 1D). Therefore, whereas total KLF5 is expressed in both basal and luminal cells, Ac-KLF5 is more commonly expressed in basal cells than in luminal cells (one-fold difference)." Ac-KLF5 is preferentially expressed in basal cells in both human and mouse, indicating a similar expression pattern of Ac-KLF5 between human and mouse prostates. Higher expression of Ac-KLF5 in basal cells is consistent with the role of Klf5 acetylation in basal progenitors.

The antibody against deacetylated KLF5 did not produce reliable data, and we thus cannot make conclusions regarding the expression of deAc-KLF5 between human and mouse prostates.

More directly addressing role of KLF5 and its acetylation in human prostate epithelial cells, we have knocked down the endogenous *KLF5*, ectopically expressed *KLF5*, *KLF5^{KR}* and *KLF5^{KQ}* in primary culture of human prostate epithelial cells, which were purchased from ATCC (TCC® PCS-440-010), and then performed organoid culture assays using these cells (Response Fig. 5, See *next page* in this file). The findings are consistent with those from both RWPE-1 human cells and mouse prostates, as knockdown of *KLF5* significantly decreased the number of organoids, and the decrease was rescued by ectopic expression of *KLF5* or *KLF5^{KR}* but not by that of *KLF5^{KQ}* (Response Fig. 5A, 5B, See *next page* in this file). These results support the role of deAc-KLF5 in maintaining organoid forming capacity.

However, we decided not to include these data in the main figures, because the differentiation of luminal cells was not complete even in wildtype group, as indicated by the expression of both basal marker CK5 and luminal marker CK8 (Response Fig. 5C, See *next page* in this file). Considering that it is best to employ a model that can recapitulate the process of basal to luminal differentiation, we included a similar organoid formation experiment using mouse prostate cells with the genotype of *p63^{CreERT2/+};R26R^{YFP/+};Klf5^{-/-}* (new Fig. 5E-G). As we have described in the revised manuscript (Line 378 on page 12 in the revised manuscript), "*KLF5^{WT}* and *KLF5^{KR}* gave rise to more YFP+ organoids regardless of DHT conditions, while *KLF5^{KQ}* and *KLF5^{KA}* did not have such an effect (Fig. 5E), suggesting that deacetylation of KLF5 promotes organoid formation of basal cells." This finding is consistent with the data from primary prostate epithelial cells (Response Fig. 5).

We have obtained additional data that further addresses the comment, as described in the revised manuscript (Line 381 on page 12): "Furthermore, expression of *KLF5^{WT}* restored normal differentiation of basal cells, as suggested by cystic organoids where basal cells (CK5+) resided around the outer layers and luminal cells (CK8+) orientated towards the lumen (Fig. 5F)." These findings validate the *in vitro* organoid model as a model of complete basal to luminal differentiation, which provides the opportunity to test whether Ac-KLF5 and deAc-KLF5 have distinct roles in the process of basal to luminal differentiation. The findings are indeed quite supportive, as described as below (Line 384 on page 12 in the revised manuscript): "The *KLF5^{KR}* mutant led to organoids with many more luminal cells (CK8+); while in contrast, *KLF5^{KQ}* and *KLF5^{KA}* mutants led to organoids with more basal cells (CK5+) (Fig. 5F, 5G). These results further support roles of KLF5 and its acetylation in basal progenitor maintenance and basal to luminal

differentiation, with acetylated KLF5 retaining basal features but deacetylated KLF5 leading to the differentiation of excess basal cells to luminal cells.”

Further supporting the relevance of KLF5 acetylation in the human prostate is a manuscript under preparation in which we found that acetylated KLF5 is upregulated in metastases of prostate cancer; and more importantly, the KLF5^{KQ} mutant promoted but the KLF5^{KR} mutant inhibited bone metastasis of prostate cancer in a tibial injection mouse model of prostate cancer metastasis. Although we have not made an intrinsic connection between bone metastasis and basal to luminal differentiation,

both of which are regulated by KLF5 acetylation, the metastasis data provide additional evidence that acetylation at KLF5 is involved in pathophysiology of the prostate.

Response Fig.5

Recommendations:

1) A scheme/cartoon on the different cell types/lineages in the prostate along with markers would be helpful.

Response: We thank the reviewer for this suggestion and have revised the model figure (new Fig. 8H) to include lineage markers.

2) Although well-described in the text, the understanding of the generation of mouse strains would be eased by small schemes integrated into the main figures.

Response: We have included the schematic for the generation of the KLF5^{KR} knockin mouse strain in new Fig. 3A.

3) Figure 5A: what is the difference between rows 3&4 and 5&6 (p63-CreERT2 model)?

Response: Rows 3 and 4 are two prostate samples with the $p63^{CreERT2/+};R26R^{YFP/+};Klf5^{-/+}$ genotype, while rows 5 and 6 are two prostate samples with the $p63^{CreERT2/+};R26R^{YFP/+};Klf5^{-/KR}$ genotype.

genotype. We have revised the legend to clearly describe these genotypes (Line 1061 on page 30 in the revised manuscript).

4) Please revise legends to the figures to contain all relevant information and abbreviations for a “stand-alone” reading and understanding of the figures.

Response: We appreciate the comments. We have revised all legends to figures with more precise information to make them clearer.

5) The discussion is way too long and is more a repetition of the results section, rather than a true interpretation of the findings in the context of published literature.

Response: We have thoroughly revised and re-written the Discussion to avoid repeating the description of Results and to focus on conclusions and how they were drawn. The Discussion section is now about half of the length of that in the original submission.

Reviewer #3 (Remarks to the Author):

This manuscript investigates the function of Klf5 in prostate organogenesis using a series of cell line studies, mouse models, and organoid studies. The authors use mutant forms of Klf5 to conclude that deacetylated Klf5 is required for maintenance of basal progenitor cells, whereas acetylated Klf5 is necessary for luminal differentiation and regeneration.

Major comment 1:

While this topic is interesting and the results are of potential significance, this study has important flaws in its rationale and experimental methodology. Notably, the authors seem to propagate misconceptions about the prior literature on prostate basal cells and its interpretation. The authors imply that postnatal prostate development is largely mediated by multipotent basal progenitors (e.g., lines 509-510), but ignore the contribution of unipotent luminal progenitors. However, at the time point used for tamoxifen induction (3 weeks) in the mouse studies, most luminal cells arise from luminal progenitors, not basal progenitors; the authors never examine a different time point in their studies. Moreover, the authors often use the term “basal progenitor” when simply referring to basal cells, and use the term “progenitor” when discussing experiments with cell lines, which seems inappropriate. Furthermore, the authors employ both sphere formation and organoid formation assays, which is confusing, and do not discuss the differences between these assays with respect to basal and luminal differentiation under the conditions used.

Response: We truly appreciate these thoughtful and constructive comments. While imprecise description and interpretation led to some of the concerns, new experiments were performed to address some other points. Below we describe how we have revised the manuscript to address these concerns.

First of all, the premise of this study is basal progenitor cells and their differentiation to luminal cells, because basal cell-specific p63-Cre mice were used to knock in the *Klf5*^{K358R} mutant and p63+ cells contain basal progenitors. As indicated by findings from those who established the p63-Cre mouse line^{10, 11} and our own data in this manuscript (new Fig. 2D, 2E), YFP+ basal cells labeled by p63-Cre give rise to both basal and luminal lineages and thus have the features of progenitor cells.

We realized that we often used the term “luminal cells” in the original manuscript when we really referred to “basal progenitor-derived luminal cells”, which appeared to have caused some confusion. In this regard, we have used “basal progenitor-derived luminal cells” when applicable to distinguish these cells from other luminal cells (e.g. luminal cells originated from luminal progenitors).

While our study does not involve luminal progenitors, we did not ignore the contribution of unipotent luminal progenitors to prostate postnatal development. In fact, we clearly stated in the Introduction section that multipotent basal progenitors and unipotent luminal progenitors contribute to the formation of luminal cells in the prostate in lines 90-93 in the revised

manuscript: “One is basal multipotent stem cells that divide and differentiate into basal, luminal and neuroendocrine cells; the other is unipotent basal and luminal progenitors that give rise to basal and luminal cells, respectively¹³.” In addition, the contribution of multipotent basal progenitors to luminal cells has been well accepted, as summarized in the text in lines 69-74 in the revised manuscript: “Accumulating evidence indicates the existence of a small population of self-renewing basal cells¹, basal cells can reconstitute xenografts in the renal capsules of immunodeficient mice when mixed with embryonic urogenital sinus mesenchymal (UGSM) cells, and the reconstituted tissues contain the referenced three cell lineages^{2,3}. Lineage tracing with basal cell-specific gene promoters of keratin 5 (K5) and p63 revealed that basal cells can give rise to other lineages including the luminal lineage⁴⁻⁶.” We indeed realized that some of the sentences (lines 509-510) in the original manuscript, as mentioned by the reviewer, could mislead readers, so we have removed these in the revised version.

Secondly, the reviewer raises a concern regarding the contribution of basal progenitors in the generation of luminal cells at the time point used for tamoxifen induction at day 18 after birth. Although our detection of YFP+ luminal cells five weeks after tamoxifen administration proves the successful differentiation of basal progenitors to luminal cells (new Fig. 4G), we agree with the reviewer that an earlier time for tamoxifen administration helps and could be better. Accordingly, we added postnatal day 7 as another time point for inducing the knockin of *Klf5^{KR}* by tamoxifen, at which point basal cells play a more significant role in the generation of luminal cells, as reported by Dr. Cedric Blanpain and Dr. Weiqiang Gao^{7,8}. At this time point, our findings were even stronger than those from the time point in the original manuscript, as indicated by the expression of YFP, basal marker p63, luminal marker CK8 and proliferation marker Ki67 (new Fig. S6E-S6L, See *next page* in this file). These new findings are now described in lines 340-346 on page 11 in the revised manuscript, as below: “We also treated mice with tamoxifen at postnatal day 7 to induce *Klf5^{KR}* knockin, at which time basal cells more actively differentiate into luminal cells for prostate development^{13,36}. Prostates were collected at postnatal week 6, before sexual maturation, for analysis. The findings were consistent with those from knockin at postnatal day 18, and *Klf5^{KR}* knockin clearly gave rise to more proliferative YFP+ luminal cells while not affecting the number or proliferation rate of basal cells (Fig. S6E-S6L, Table S2g).”

Thirdly, indeed it is not appropriate to use the term “basal progenitor” when simply referring to basal cells, or the term “progenitor” when discussing experiments with cell lines. This also occurred to “luminal cells” versus “basal progenitor-derived luminal cells”. We have revised the manuscript carefully to be precise and more specific with different terms, only keeping “basal progenitor cells” when describing the basal cells which have the capability of luminal differentiation. We also agree that the current results from human prostate cell lines serve more as clues rather than as evidence for our study. They are definitely not sufficient to indicate the role of KLF5 and its mutants in maintaining progenitor capability, because we neither performed serial passaging assay to indicate self-renewal nor observed a complete luminal differentiation in the sphere formation assay. Therefore, we have revised the results and conclusions from immortalized human prostate cell lines as below: “These findings prompted us to test a role of KLF5 in progenitor capacity using *in vivo* models” in the second

paragraph of the Result sections (Line 159 on page 6 in the revised manuscript); and “*KLF5*^{WT} and *KLF5*^{K369R} restored, but *KLF5*^{K369Q} did not affect, the expression of basal markers CK5, ΔNp63 and CK14 (Fig. S3A, S3B) and efficiencies of colony (Fig. S3C) and sphere formation (Fig. S3D, S3E), suggesting different roles of acetylated *KLF5* and deacetylated *KLF5* in the maintenance of progenitor capability of prostate epithelial cells” (Line 226 on page 8 in the revised manuscript).

Finally, regarding the sphere formation and organoid formation assays, we indeed had some misinterpretations that have been corrected. Luminal differentiation is not complete in the sphere formation assay, so this assay is not a reasonable model to test basal progenitor capacity. Moreover, RWPE-1 cells express both basal and luminal markers at the initial step of sphere formation, which raises a concern about how faithful this model could be in mimicking the process of basal to luminal differentiation. We therefore extensively revised the manuscript by moving most of the sphere formation data into the supplemental material, shortening the description of these results and only using them as a suggestion for the potential roles of *KLF5* and its acetylation in progenitor capacity. The data showing IF staining in the sphere assay (old figure 2E) were also removed for the same reasons.

As a substitute for the sphere formation assay using immortalized human prostate epithelial cells, we ectopically expressed *KLF5*, *KLF5*^{KR}, *KLF5*^{KQ} and *KLF5*^{KA} in YFP+/Klf5-null basal cells isolated from mouse prostates with the *p63*^{CreERT2/+}; *R26R*^{YFP/+}; *Klf5*^{-/-} genotype immediately after tamoxifen administration, and then performed an organoid formation assay, which provides a better model for mimicking the process of basal to luminal differentiation (new Fig. 5E-5G, See

page 2 in this file). Consistent with the effects in RWPE-1 cells, the number of organoids was increased by KLF5 and KLF5^{KR} but not by KLF5^{KQ} and KLF5^{KA}. More interestingly, organoids expressing KLF5 had normal phenotypes, where basal cells (CK5+) resided around the outer layers and luminal cells (CK8+) orientate toward the lumen in organoids; but organoids with KLF5^{KR} had many more luminal cells (CK8+) and organoids with KLF5^{KQ} and KLF5^{KA} had more basal cells (CK5+)(new Fig. 5G). These findings are consistent with those from *in vivo* analyses.

In the revised manuscript, we have now clearly described the procedures for how we performed the sphere formation assay and the organoid culture assay in the Methods section (Line 735 on page 21 in the revised manuscript).

Major concern 2:

In general, it is extremely difficult to understand the design of the experiments performed in this study. One major issue is that the authors never state the exact genotypes of the mice being analyzed, and instead employ a non-standard genetic nomenclature (e.g., what do W/W, F/W, F/F, KR/W, and F/FK358R mean?). These notations also often imply that Cre-mediated recombination is complete within the cells being analyzed, which should not be assumed. Furthermore, the manuscript is densely written and hard to follow; for example, the relationship between experiments shown in the supplemental figures and those in the main figures is often unclear. Finally, the authors have made little attempt to convey the broader significance of their findings.

Response: Many of the issues raised here appear to be caused by less-than-optimal writing and some imprecise and even improper description and interpretation of findings. In this regard, we have extensively revised both the figures and manuscript to improve the presentation and correct/clarify other issues, including the use of standard genetic nomenclatures. In the revised manuscript, we now use standard gene nomenclatures, with *Klf5*^{LSL-K358R} referring to the allele with loxP-stop-LoxP before the K358R mutation, and *Klf5*^{lox} referring to the *Klf5* allele floxed by loxP sites. Detailed information of the *Klf5*^{KR} allele is shown in the new Fig. 3A. In prostate samples analyzed, Cre-mediated recombination has removed the gene sequence between the LoxP sites, so we used the following notations: *Klf5*^{+/+}, both *Klf5* alleles are wildtype; *Klf5*^{-/+}, heterozygous deletion of *Klf5*; *Klf5*^{-/-}, homozygous deletion of *Klf5*; and *Klf5*^{-/KR}, one allele of *Klf5* is deleted while the *Klf5*^{KR} mutation is simultaneously introduced in the other allele.

Cre-mediated recombination is often incomplete within cells that have been analyzed as Cre can be inactive in some cells. However, tracing with YFP in all experimental groups should have kept this issue under control when interpreting findings.

We appreciate the comment that “the manuscript is densely written and hard to follow; for example, the relationship between experiments shown in the supplemental figures and those in the main figures is often unclear”, and have extensively revised and re-written the manuscript. Unnecessary methodology descriptions have been moved to the Methods section; redundant descriptions of results have been removed; the discussion section has been shortened to about half of the original length; and data presentation has been reorganized to place closely

associated data together (e.g., we merged the data of Ki67 in YFP- cells in old supplementary figures to the main figures; and reorganized the old Fig. 3 and old Fig. S4 by placing related data together). The logical flow has also been improved.

Regarding the comment that “the authors have made little attempt to convey the broader significance of their findings”, we have shortened the discussion and considered the implications of our findings in multiple aspects. Fate determination of basal progenitor cells is one of them, which is essential to the development of prostates and other organs. In addition, our findings also imply that acetylation of KLF5 plays a role in castration resistance in basal progenitor-derived luminal cells, which could be involved in the development of castration resistant prostate cancer.

Below are some text sections where we have indicated the importance/significance of our study:

Line 638 on page 18 in the revised manuscript: “These findings establish Klf5 and its acetylation as cell fate determinants of basal progenitor cells in prostate epithelial homeostasis. They also demonstrate that PTM of a transcription factor, i.e., acetylation of Klf5, can profoundly impact the development of a glandular organ.”

Line 571 on page 16 in the revised manuscript: “An essential role of Klf5 and its acetylation in the maintenance and luminal differentiation of basal progenitors could also occur in other organs with epithelia, because the expression of p63 is ubiquitous in the basal cells of various organs such as the lungs, mammary glands, digestive tracts, liver, epidermis etc.”

Line 612 on page 17 in the revised manuscript: “These findings also strongly suggest that Ac-Klf5 functions in castration resistance in basal progenitor-derived luminal cells. KLF5 acetylation may thus be required for the development of castration-resistant prostate cancer. Castration resistance is a common problem in the treatment of prostate cancer, and whether acetylation of Klf5 contributes to castration resistance in prostate cancer is thus a clinically relevant question worth addressing. In this regard, the mouse strain with the *Klf5*^{LSL-K358R} allele should prove helpful.”

Line 634 on page 18 in the revised manuscript: “The *Klf5*^{K358R} knockin mouse model should thus provide a unique tool for further studying prostate homeostasis and the role of Klf5 acetylation in diseases, including castration-resistant prostate cancer.”

Major concern 3:

Another major issue is that the authors never state the number of mice/samples analyzed, rarely provide raw data for their quantitation and statistical analyses, and do not provide actual p-values. This is of particular concern since several key results are not particularly convincing in the absence of quantitation (e.g., Fig. 3G), whereas seemingly large differences are deemed as not significant (e.g., Fig. S5H).

Response: We thank the reviewer for these comments. In addressing these issues, we have now revised Table S2, which contains all of the raw data, including all the information specified above. We have also specified the number of mice used for each analysis in the corresponding figure legends, and provided p-values for all comparisons that were not statistically significant (these were missing before). The significance statement is also included at the end of each figure legend. Inclusion of actual p-values clearly shows some comparisons with marginal significance, which is actually helpful for reading the data in a more accurate manner.

Specific comments:

1. Fig. 1D, E: This lineage-tracing experiment is missing basic controls and information. The authors should examine a time point shortly after tamoxifen administration to show and quantitate the basal cell type specificity of marking. What is the efficiency of recombination? How many mice were analyzed?

Response: We have examined another time point immediately after 5 consecutive days of tamoxifen treatment, and show the data in new Fig. S2C. By costaining YFP and p63, we found the efficiency of recombination to be about 9% in all of the genotypes (new Fig. S2C). Three or four mice were used for each group in this assay ($Klf5^{+/+}$, n=3; $Klf5^{+/-}$, n=4; $Klf5^{-/-}$, n=3), which was also stated in the corresponding figure legends. The raw data is now provided in Table S2f.

2. Fig. 1L-N: The reported percentages of Ki67-positive luminal and basal cells in what should be phenotypically wild-type (W/W) mice at 8 weeks of age seem extremely high (20 and 15%, respectively). These images are also inconsistent with the W/W image shown in Fig. 3E.

Response: We appreciate the reviewer pointing this out and agree that the percentage of Ki67-positive cells seems unexpectedly high. However, after double-checking our original data, we confirm that the data is correct as is (See raw data in Table S2a). One potential reason for this extremely high percentage of Ki67-positive cells in both basal and luminal cells could be that the basal and luminal cells were derived from p63+ basal progenitors (YFP-labeled), and such cells inherently have a higher rate of proliferation. As presented in the new Fig. 2K (See page 11 in this file), in prostates with wildtype $Klf5$ ($Klf5^{+/+}$) (n=5), the YFP+ basal cells (those originated from p63+ basal progenitor cells) had a higher Ki67+ rate (15.5%) compared to YFP- basal cells (8.4%), although the difference was marginally significant. In the same mice, the YFP+ luminal cells (i.e., those derived from p63+ basal progenitor cells) had a Ki67 index of about 20%, while the YFP- cells had a Ki67 index of 13.4%, which was also marginally significant (new Fig. 2K). These data suggest that both the basal and luminal progenies of p63+ basal progenitor cells have a higher proliferation rate. Loss of $Klf5$, on the other hand, decreased the cell proliferation

in both YFP+ basal and luminal cells, which suggests a necessary role of Klf5 in cell proliferation, i.e., Klf5 could have contributed to the higher proliferation rates in both the basal and luminal progenies of p63+ basal progenitors.

Another likely reason is that mouse prostates at 8 weeks are still premature, and epithelial cells at this stage are still quite active and varying in proliferation compared to those in prostates at maturation. The percentage of proliferating cells was lower in mouse prostates with constitutive KR knockin (new Fig. S5D), which could be attributed to the different genetic backgrounds between the two mouse lines (the conditional knockin mice have one quarter of the 129 background, whereas the constitute KR knockin mice are totally in the B6 background). Different genetic backgrounds could affect postnatal development of the prostate, giving rise to variable proliferation rates during development.

3. Fig. S1J,K: The authors should show histology and immunostaining of these organoids to show any alterations in basal and/or luminal differentiation.

Response: We appreciate this comment. We have detected the expression of basal marker CK5 and luminal marker CK8 using IF staining (new Fig. S1L). Loss of Klf5 disrupted the process of basal to luminal differentiation, as indicated by the lack of CK8+ cells oriented to the lumen.

3. Lines 187-188: There seems to be little basis for the statement about “grouped YFP+ luminal cells” that “Each such group/unit should have been primarily derived from a single basal progenitor cell.” Why couldn’t grouped YFP+ luminal cells have been derived from distinct basal progenitors? Is there a difference between groups and units? The explanation for how analysis of YFP+ luminal cells and luminal units allows quantitative analyses of the ability of basal progenitors to form luminal cells is unclear.

Response: We appreciate these comments. Indeed there were insufficient description and clarification regarding the definition and use of YFP+ luminal units in our analysis. This issue was also raised by Reviewer #1. In the revised manuscript, we have renamed “grouped YFP+ luminal cell” as “a YFP+ luminal cluster”, which refers to a cluster of 2 or more adjacent YFP+ luminal cells. On the other hand, a “YFP+ luminal unit” refers to either a YFP+ luminal cluster or a single YFP+ luminal cell that is not adjacent to any other YFP+ luminal cells. As we have discussed in our response to the specific comment 1 from Reviewer #1 (See page 6 in this file), we didn’t observe any 2 or more adjacent basal cells that were labeled with YFP originally (new Fig. S2C, S6A), and luminal cells do not migrate horizontally because of their tight junctions⁷. Therefore, each YFP+ luminal cluster should have originated from an originally YFP+ labelled basal

progenitor cell via amplification of the basal progenitor, basal to luminal differentiation, and/or subsequent proliferation of the differentiated YFP+ luminal cell in that cluster, because the initial YFP activation occurred only in basal cells. Similarly, each single YFP+ luminal cell should have also originated from a single YFP+ basal progenitor via basal to luminal differentiation. A “YFP+ luminal unit” thus refers to either a “YFP+ luminal cluster” or “YFP+ luminal single cell”. We have eliminated the use of YFP+ groups to avoid confusion.

The purpose of using the term “YFP+ luminal units” is to exclude the impact of basal cell amplification and luminal cell proliferation on how many YFP+ luminal cells originated from originally YFP-labeled basal progenitors via differentiation. In other words, a YFP+ luminal unit is a differentiation unit, focusing on the quantitative analyses of basal to luminal differentiation. The idea of using YFP+ luminal units to quantitatively analyze the ability of basal to luminal differentiation is similar to the idea of “clonal analysis” in Dr. Cédric Blanpain’s study⁸.

In addition, knockin of the *Klf5*^{KR} mutant did not increase basal cell proliferation, which excludes the contribution of basal cell proliferation to the increase in YFP+ luminal units.

Because we did not observe any 2 or more adjacent basal cells that were labeled with YFP originally (new Fig. S2C, See page 31 in this file), it is convincing that different clusters of YFP+ luminal cells should have been derived from distinct basal progenitors, the origin of all YFP+ cells in the same YFP+ luminal cluster should arise from the same originally YFP-labelled basal progenitor.

Considering these comments, we have revised the paper to clarify these terms, their use in our analyses, and the interpretation of our findings (Line 195 on page 7 in the revised manuscript): “A basal progenitor can divide to increase the pool of basal cells, a basal cell can differentiate to become a luminal cell, and a luminal cell can divide to produce more luminal cells^{6,13}. Therefore, the decrease in YFP+ luminal cells by Klf5 deletion in basal progenitors could be attributed to reduced amplification of basal progenitors, interruption of basal to luminal differentiation, and/or compromised luminal cell division. To clarify which of these processes is affected by Klf5 loss, we analyzed YFP+ luminal clusters, each of which contains at least 2 adjacent YFP+/CK18+ cells. Each cluster should have been primarily derived from a single YFP-labeled p63+ basal progenitor, by the amplification and subsequent differentiation of a basal progenitor and/or division of a luminal cell that had differentiated from a basal progenitor. In either case, basal to luminal differentiation must occur, and a cluster of YFP+ luminal cells can then be considered as a unit to better indicate the differentiation, minimizing the effect of basal or luminal cell proliferation on the analysis¹³. In addition, a single YFP+ luminal cell not in a cluster, which should have also arisen from the differentiation of a basal progenitor cell because luminal cells normally do not migrate horizontally due to their tight junctions¹³, was also considered as a unit in the analysis. Interestingly, the absence of Klf5 significantly decreased the number of YFP+ luminal units (Fig. 2H), supporting a role of Klf5 in the differentiation of basal to luminal cells.”

4. Lines 231-232: The authors suggest that “these findings also suggest that acetylation of KLF5 could shift KLF5 function from self-renewal of self/progenitor cells to luminal

differentiation”, but this seems overly speculative given that the sphere formation assay does not assess luminal differentiation . Notably, AR expression is uniform and cytoplasmic.

Response: We appreciate this comment as some statements indeed require clarification. Considering sphere formation assay of RWPE-1 cells is not a faithful model for basal progenitor capacity and basal to luminal differentiation; thus have removed this and made extensive revisions in the Discussion section, as we discuss above in our response to the last point of major concern 1 (See page 28 in this file).

As a substitute for the experiment related to this comment, we ectopically expressed KLF5, KLF5^{KR}, KLF5^{KQ} and KLF5^{KA} in YFP+/Klf5-null basal cells isolated from mouse prostates with the *p63^{CreERT2/+};R26R^{YFP/+};Klf5^{-/-}* genotype immediately after tamoxifen administration, and performed an organoid formation assay, which provides a better model for mimicking the process of basal to luminal differentiation *in vitro* (new Fig. 5E-5G, See page 2 in this file). Consistent with the effects in RWPE-1 cells, the number of organoids was increased by KLF5 and KLF5^{KR}, but not affected by KLF5^{KQ} and KLF5^{KA}. In addition, the KLF5^{KR} organoids had many more luminal cells (CK8+) but the KLF5^{KQ} and KLF5^{KA} organoids had more basal cells (CK5+)(new Fig. 5G, See page 2 in this file), which is consistent with the *in vivo* findings.

5. Fig. 3A: The histology of the KR/KR prostate is not noticeably different from the controls. However, the quality of this image is poor, as the section shown at higher-power is torn.

Response: The unapparent changes in the prostates of germline *Klf5^{KR}* mice could be attributed to a compensatory mechanism during embryo development. On the other hand, we noticed that prostates in these constitutive *Klf5^{KR}* mice were significantly smaller, had less p63+ cells, and formed larger spheres *in vitro*. These data were consistent with the findings in conditional *Klf5^{KR}* knockin mice. We have replaced the previous images with those of high quality (new Fig. 3B).

6. Fig. 3H,I: The reported increase in secretory proteins in the KR/KR prostate seems to differ from the histology shown in Fig. 3A, and quantitation of staining intensity is not an appropriate assay to support this conclusion. The authors should perform Western blotting of prostate secretions to address this issue. If overall luminal areas are decreased, there might be an increased density of secretory proteins, as opposed to increased secretions.

Response: We agree with the reviewer. Accordingly, we have removed the IHC data, and performed Western blotting to address the changes of prostate secretions (new Fig. S5F). Knockin of the *Klf5^{KR}* mutant did not affect the

overall secretions in the lumen, but it increased two secretory proteins, i.e., probasin and Spink3, in secrete prostate fluid.

7: Fig. 3L: The KR/KR organoid image shown is quite unusual, as it seems to have basal cells in the interior and luminal cells on the exterior. The authors should examine additional markers to support their conclusion that luminal differentiation is enhanced in the mutant.

Response: We appreciate the comment. We have repeated the staining of organoids with the *Klf5*^{KR} knockin by staining basal marker CK5 and luminal marker CK8 (new Fig. 3H). While the wild-type organoids had a typical basal-luminal organization with basal cells (CK5+) residing around the outer layers and luminal cells (CK8+) oriented toward the lumen, the *Klf5*^{KR} organoids had fewer and deranged basal cells and more luminal cells. We have replaced the original images.

8: Fig. 4E: The CK18 immunostaining is quite heterogeneous in these images. Could this reflect defective luminal differentiation? The authors should examine other luminal markers to address this possibility.

Response: In response to this excellent question, we have stained another luminal marker, CK8 (new Fig. 4F). CK8 expression was homogeneous, thus minimizing the possibility of defective luminal differentiation.

9. Fig. 4G-J: If luminal proliferation is increased in the F/FK358R prostates, how can the authors conclude that there is increased basal differentiation into luminal cells? How can the authors be confident that “luminal units” arise from single basal progenitors?

Response: This is one of the conclusions that required clarification and further experiments. As we have discussed in our response to this reviewer’s second specific comment 3 (two comments were numbered 3), we have clarified the definition of “YFP+ luminal units”, which include both “YFP+ luminal clusters” and “YFP+ luminal single cells” (i.e., a YFP+ luminal cell that is not adjacent to any other YFP+ luminal cells). As we have extensively discussed in our response to Reviewer 1’s specific comment 1 on page 6 in this file, each YFP+ luminal unit must have originated from a YFP-labeled basal progenitor via the process of basal to luminal differentiation. The logic for this claim is as follows: the increase in YFP+ luminal cells can be attributed to excess amplification of basal progenitors, differentiation of excess basal cells to luminal cells, and/or enhanced proliferation of luminal cells. Of these causes, excess amplification of basal progenitors is less likely to occur, because the *Klf5*^{KR} knockin did not increase the number of YFP+ basal cells (new Fig. 4D). Although the proliferation of basal progenitor-derived YFP+ luminal cells contributes to the increase in YFP+ luminal cells, such a

contribution can be excluded when YFP+ luminal units are analyzed, because the proliferation of basal and luminal cells does not increase the number of YFP+ luminal units and luminal cells do not migrate horizontally due to their tight junctions⁷.

Of course, clusters of basal progenitors can differentiate into clusters of luminal cells, and consequently, an increase in YFP+ basal clusters can contribute to an increase in YFP+ luminal clusters. However, initial Cre activation does not appear to occur in adjacent basal cells, as we did not observe any two or more YFP+ basal cells that were adjacent to each other immediately after tamoxifen treatment (new Fig. S2C, See page 31 in this file). Taken together with the use of YFP+ luminal units, we are confident that changes in YFP+ luminal units can thus indicate changes in the differentiation of basal progenitors to luminal cells. We also noticed that knockin of *Klf5*^{KR} increased YFP+ luminal units but did not increase YFP+ basal cells, providing another line of evidence supporting our conclusion.

With additional organoid experiments in which deacetylation of Klf5 caused abnormalities changes in both the distribution of basal and luminal cells and luminal organization, we now have additional and more direct evidence for the conclusion that loss or deacetylation of Klf5 impairs the basal to luminal differentiation (new Fig. 5E-G, see page 2 in this file).

10. Fig. 4M: Why is the AR staining in these organoids cytoplasmic?

Response: We appreciate this comment. Organoids in old Fig. 4K and old Fig. 4L were under the DHT treatment condition, but those in the old Fig. 4M were not. We overlooked changing IF staining figures when finalizing the manuscript, and we do apologize

for that. The cytoplasmic staining of AR in organoids of the old Fig. 4M was due to the lack of DHT, which binds to AR to cause AR's nuclear translocation. We have performed additional experiments and reorganized these data by including both the conditions with and without DHT treatment in new Fig. 5C (bottom panel). Addition of DHT rendered clear AR staining in the nucleus.

11. Fig. 5E: These images are difficult to understand. Since the YFP should be expressed uniformly in all cells, why are some organoids much brighter than others? The authors claim that the morphological differences between organoids are due to alterations in luminal apoptosis, yet they do not show histology of the organoids or examine apoptosis directly.

Response: The reviewer is correct. The photos of live organoids in Matrigel were taken directly with a phase-contrast fluorescent microscope. In fact the YFP intensities were similar at the first look, but it took time to focus on different organoids and take pictures, during which some signals were quenched to some extent. We have repeated these experiments with better

quality in new Fig. 6E, in which the brightness of the organoids was similar, as pictures were taken as organoids were identified and focused. Regarding the issue of apoptosis, we have detected apoptotic luminal cells in the organoids derived from mouse prostates with the $p63^{CreERT2/+}; Rosa^{YFP/+}; Klf5^{-/KR}$ genotype, and found that YFP+ luminal cells ($Klf5^{KR}$ knockin) had less apoptosis than those YFP- luminal cells (wildtype $Klf5$) (new Fig. 5D), suggesting a role of $Klf5^{KR}$ in suppressing apoptosis in luminal cells during organoid formation.

12. Lines 449-450: The authors conclude that “acetylation of $Klf5$ is required for the generation and/or maintenance of most castration-resistant luminal cells in the prostate”, but have only examined the luminal cells that derive from basal progenitors during organogenesis, which is a minority of the luminal population. The authors should use other Cre drivers to investigate this issue.

Response: We have now clarified our conclusion in the sense that it is specific for those basal progenitor-derived luminal cells rather than all luminal cells. We appreciate the reviewer bringing up this confusion. We have emphasized this subpopulation of luminal cells by concluding that acetylation of $Klf5$ is required for the generation and/or maintenance of castration-resistant basal progenitor-derived luminal cells in the prostate.

Our models are restricted to basal progenitors, their differentiation into luminal cells, and their luminal progenies. Whether and how $KLF5$ and its acetylation function in luminal progenitor cells and their differentiation into luminal cells represents another series of important questions that are worth addressing, but they are more appropriate for another project. For example, the $CK18^{CreERT2}$ mice can be used to specifically knock in the $Klf5^{KR}$ allele in luminal progenitor cells and examine the effects of $Klf5$ deacetylation on postnatal development of the prostate and its epithelial homeostasis.

13. Fig. 7A-D: The authors show data for YFP+ cells in the regeneration experiment shown, but should also show data for YFP- cells as an internal control.

Response: We appreciate this comment. We originally used total cells as an internal control (old Fig. S7E), but we agree with the reviewer that YFP- cells serve as better controls. In this regard, we have changed the control to YFP- cells and reorganized the data with those from YFP+ cells in the new Fig. 8D. The cell proliferation index was not increased in the $Klf5^{KR}$ knockin group but significantly decreased in luminal cells after a castration and regeneration cycle, strengthening our conclusions.

References

1. Miyamoto, S. *et al.* Positive and negative regulation of the cardiovascular transcription factor KLF5 by p300 and the oncogenic regulator SET through interaction and acetylation on the DNA-binding domain. *Mol Cell Biol* **23**, 8528-8541 (2003).
2. Guo, P., Zhao, K.W., Dong, X.Y., Sun, X. & Dong, J.T. Acetylation of KLF5 alters the assembly of p15 transcription factors in transforming growth factor-beta-mediated induction in epithelial cells. *J Biol Chem* **284**, 18184-18193 (2009).
3. Guo, P. *et al.* Pro-proliferative factor KLF5 becomes anti-proliferative in epithelial homeostasis upon signaling-mediated modification. *J Biol Chem* **284**, 6071-6078 (2009).
4. Guo, P. *et al.* Opposing effects of KLF5 on the transcription of MYC in epithelial proliferation in the context of transforming growth factor beta. *J Biol Chem* **284**, 28243-28252 (2009).
5. Li, X. *et al.* Interruption of KLF5 acetylation converts its function from tumor suppressor to tumor promoter in prostate cancer cells. *Int J Cancer* **136**, 536-546 (2015).
6. Tao, R. *et al.* HDAC-mediated deacetylation of KLF5 associates with its proteasomal degradation. *Biochem Biophys Res Commun* **500**, 777-782 (2018).
7. Wang, J. *et al.* Symmetrical and asymmetrical division analysis provides evidence for a hierarchy of prostate epithelial cell lineages. *Nature communications* **5**, 4758 (2014).
8. Ousset, M. *et al.* Multipotent and unipotent progenitors contribute to prostate postnatal development. *Nat Cell Biol* **14**, 1131-1138 (2012).
9. Wang, Z.A. *et al.* Lineage analysis of basal epithelial cells reveals their unexpected plasticity and supports a cell-of-origin model for prostate cancer heterogeneity. *Nat Cell Biol* **15**, 274-283 (2013).
10. Lee, D.K., Liu, Y., Liao, L., Wang, F. & Xu, J. The prostate basal cell (BC) heterogeneity and the p63-positive BC differentiation spectrum in mice. *Int J Biol Sci* **10**, 1007-1017 (2014).
11. Pignon, J.C. *et al.* p63-expressing cells are the stem cells of developing prostate, bladder, and colorectal epithelia. *Proc Natl Acad Sci U S A* **110**, 8105-8110 (2013).
12. Pu, H., Begemann, D.E. & Kyprianou, N. Aberrant TGF-beta Signaling Drives Castration-Resistant Prostate Cancer in a Male Mouse Model of Prostate Tumorigenesis. *Endocrinology* **158**, 1612-1622 (2017).
13. Wang, X. *et al.* A luminal epithelial stem cell that is a cell of origin for prostate cancer. *Nature* **461**, 495-500 (2009).
14. Yoo, Y.A. *et al.* Bmi1 marks distinct castration-resistant luminal progenitor cells competent for prostate regeneration and tumour initiation. *Nature communications* **7**, 12943 (2016).
15. Evans, G.S. & Chandler, J.A. Cell proliferation studies in the rat prostate: II. The effects of castration and androgen-induced regeneration upon basal and secretory cell proliferation. *Prostate* **11**, 339-351 (1987).
16. Evans, G.S. & Chandler, J.A. Cell proliferation studies in rat prostate. I. The proliferative role of basal and secretory epithelial cells during normal growth. *Prostate* **10**, 163-178 (1987).
17. Guo, C., Zhang, B. & Garraway, I.P. Isolation and characterization of human prostate stem/progenitor cells. *Methods Mol Biol* **879**, 315-326 (2012).

18. Lukacs, R.U., Goldstein, A.S., Lawson, D.A., Cheng, D. & Witte, O.N. Isolation, cultivation and characterization of adult murine prostate stem cells. *Nat Protoc* **5**, 702-713 (2010).
19. Leong, K.G., Wang, B.E., Johnson, L. & Gao, W.Q. Generation of a prostate from a single adult stem cell. *Nature* **456**, 804-808 (2008).

Reviewers' Comments:

Reviewer #1:

Remarks to the Author:

This revised manuscript by Zhang and colleagues reports the role of Klf5 and Klf5 acetylation in prostatic epithelial development. Overall the Reviewers' concerns have been thoroughly addressed. The conclusions are now well-presented, seem very clear and are largely consistent between the various models.

Minor comments (textual):

Line 110: there are two contexts provided for this sentence, including "in prostate tumors" followed by "in prostate epithelial cells", which makes this sentence difficult to understand.

Line 118: the use of "etc" is redundant and this text should be deleted.

Reviewer: John M. Ruppert

Line 265: "unnecessarily" could be changed to "prematurely".

Line 330-331: "more than 1-fold" could perhaps be changed to "by approximately x %" and showing a p value in the corresponding figure.

Reviewer #2:

Remarks to the Author:

The authors have addressed all concerns raised by the reviewer in the original submission. Additional experiments have been performed to corroborate the findings in the human prostate. Readability of the manuscript has been increased with inclusion of schemes/cartoons and a more concise discussion section. The manuscript is now acceptable in its present form.

Reviewer #3:

Remarks to the Author:

This resubmission has been extensively revised to add new experimental data, correct previous misstatements and over-interpretations, improve clarity of the data presentation, and shorten the discussion. Overall, these revisions have greatly improved the manuscript, but several issues of interpretation remain, as described below. In addition, the overall significance of this work is still somewhat elusive, since most of the analyses focus on a relatively small subset of prostate epithelial cells, namely luminal cells that are derived from basal progenitors.

One remaining concern is that several conclusions about proliferation rates are based solely on Ki67 immunostaining, and not on BrdU incorporation assays. The careful use of both assays could potentially help distinguish changes in luminal cell number due to basal-to-luminal differentiation versus basal or luminal proliferation, for example in the KR mutant analyses.

Another issue is the low Cre-mediated recombination efficiency in these studies (approximately 9%; Figure S2C), raising the possibility that a subset of basal cells that is not representative of the entire population has been analyzed. Consistent with this possibility, the percentage of YFP-positive luminal cells before and after castration is similar (Fig. 4G, 7E), which is unexpected since most luminal cells undergo apoptosis following castration.

Finally, although the authors have provided much more quantitation of their data in the revised

manuscript, there are still several gaps. For example, a short-term analysis of marking immediately after tamoxifen induction at P7 was not performed, and the ratio of labeled basal cells to luminal units is not provided for the experiment shown in Figure 8.

Specific comments:

1. Line 266: "Underdeveloped prostates with fewer basal cells but more luminal cells suggests that interruption of Klf5 acetylation may cause more basal cells to unnecessarily differentiate into luminal cells, although other mechanisms could also be involved." What are these other mechanisms?
2. Line 278: "Decrease in basal cells, increase in luminal cells, decrease in cystic organoids, and of organoid organization by the interruption of Klf5 acetylation further support a role of Klf5 acetylation in the proper differentiation of basal to luminal cells." However, basal-to-luminal differentiation could be unaffected while luminal cell proliferation is increased in mutants.
3. Line 560: "The roles of Klf5 and its acetylation in the maintenance of basal progenitors and their luminal differentiation greatly impact postnatal development of the prostate". The authors show that prostate weight is affected at 8, 12, and 16 weeks, but not 24 weeks (Figure 3C), and other analyses are only shown at 8 weeks. These results suggest that the effects of Klf5 deletion/mutation are compensated for by 24 weeks of age.
4. Line 638: To state that Klf5 and its acetylation are "cell fate determinants of basal progenitor cells" represents an overstatement. Instead, it seems more appropriate to say that Klf5 and its acetylation is essential for basal-to-luminal differentiation.
5. Figure 2I: This plot should be labeled as percentage of YFP-positive basal and luminal cells, not percentage of survival, as cell death was not examined.
6. Figure 4: This figure should contain a plot similar to that in Figure 2I.
- 7: Figure 5: The DHT-negative conditions are confusing and do not really provide additional insights. These data could be removed or at least separated into a supplemental figure.

Point-by-point responses to reviewers' comments

Manuscript NCOMMS-18-35469-A-Z, by Zhang et al., entitled “*Klf5 and its acetylation maintain basal progenitors and proper basal to luminal differentiation in the prostate*”

Reviewer #1 (Remarks to the Author):

Overall comment: *This revise manuscript by Zhang and colleagues reports the role of Klf5 and Klf5 acetylation in prostatic epithelial development. Overall the Reviewers' concerns have been thoroughly addressed. The conclusions are now well-presented, seem very clear and are largely consistent between the various models.*

Response: We appreciate these favorable comments.

Minor comment 1: *Line 110: there are two contexts provided for this sentence, including “in prostate tumors” followed by “in prostate epithelial cells”, which makes this sentence difficult to understand.*

Response: We have removed the second context “in the prostate” and changed the sentence to the following: “In prostate tumors induced by Pten deletion¹⁻³, loss of Klf5 appears to alter the constitution of basal and luminal cells⁴.” (Line 112 in the revised manuscript)

Minor comment 2: *Line 118: the use of “etc” is redundant and this text should be deleted.*

Response: Deleted as suggested (Line 121 In the revised manuscript).

Minor comment 3: *Reviewer: John M. Ruppert*

Response: We have updated the name of the reviewer in the acknowledgement of the revised manuscript.

Minor comment 4: *Line 265: “unnecessarily” could be changed to “prematurely”.*

Response: We have made the change as suggested (Line 268 in the revised manuscript).

Minor comment 5: *Line 330-331: “more than 1-fold” could perhaps be changed to “by approximately x %” and showing a p value in the corresponding figure.*

Response: We have made changes as suggested in the manuscript: “By counting the number of YFP+ luminal units, we found that *Klf5^{KR}* knockin increased YFP+ luminal units by approximately 1.32 folds (Fig. 4i, $p = 0.0098$)” (Line 334 in the revised manuscript). The p-value is 0.0098, which is indicated in the figure and its legend as “**”, $p < 0.01$. We use this style throughout the manuscript.

Reviewer #2 (Remarks to the Author):

Overall comment: *The authors have addressed all concerns raised by the reviewer in the original submission. Additional experiments have been performed to corroborate the findings in the human prostate. Readability of the manuscript has been increased with inclusion of schemes/cartoons and a more concise discussion section. The manuscript is now acceptable in its present form.*

Response: We appreciate the favorable comments from the second reviewer.

Reviewer #3 (Remarks to the Author):

Overall comment: *This resubmission has been extensively revised to add new experimental data, correct previous misstatements and over-interpretations, improve clarity of the data presentation, and shorten the discussion. Overall, these revisions have greatly improved the manuscript, but several issues of interpretation remain, as described below. In addition, the overall significance of this work is still somewhat elusive, since most of the analyses focus on a relatively small subset of prostate epithelial cells, namely luminal cells that are derived from basal progenitors.*

Response: We appreciate the overall favorable comments from the reviewer. Regarding the comment on the significance of basal derived luminal cells in prostate epithelial cells, we must emphasize again that the focus of this study is the differentiation of basal progenitors.

As the reviewer also recognizes, some luminal cells are derived from basal progenitors, and this statement is well established and accepted based on a series of studies, as introduced in the revised manuscript: “One is basal multipotent stem cells that divide and differentiate into basal, luminal and neuroendocrine cells; the other is unipotent basal and luminal progenitors that give rise to basal and luminal cells, respectively⁵.” (Line 93); and “Lineage tracing with basal cell-specific gene promoters of keratin 5 (K5) and p63 revealed that basal cells can give rise to other lineages including the luminal lineage⁶⁻⁸.” (Line 76). Findings in this manuscript also support this statement (new Fig. 2d, 2e), as YFP+ basal cells labeled by p63-Cre gave rise to both basal and luminal lineages. The findings also indicate the progenitor capacity of basal cells. It is under this context that we describe the roles of Klf5 and its acetylation in basal progenitors and their differentiation, which clearly has its significance.

Regarding how many luminal cells are derived from basal progenitors and how great or small the percentage of such cells is in prostate epithelial cells, there are two facts that suggest an under estimation by our findings. One is that the percentage of basal progenitor-derived luminal cells was significantly higher when Cre was activated (to mark p63+ cells and knock in K358R) at postnatal week (PNW) 1 and cells analyzed at PNW 3 (i.e., 25.4%), comparing to 13.6% when Cre was activated at PNW 3 and cells analyzed at PNW 8 (new Fig. 4g, S6i). Therefore, at an earlier stage postnatal development, more basal progenitors appear to develop into luminal cells. The other fact is that the efficiency of p63-Cre-mediated GFP expression does not occur in 100% basal cells.

Nevertheless, we agree with the reviewer that whether Klf5 and its acetylation contribute to the generation of luminal cells from luminal stem/progenitor cells is an important question to address. This question has its own merit for studying, so the experiments addressing this question belongs to a different study.

Comment 1: *One remaining concern is that several conclusions about proliferation rates are based solely on Ki67 immunostaining, and not on BrdU incorporation assays. The careful use of both assays could potentially help distinguish changes in luminal cell number due to basal-to-luminal differentiation versus basal or luminal proliferation, for example in the KR mutant analyses.*

Response 1: We appreciate the reviewer for this comment. We have performed BrdU incorporation assay (new Fig. S6n) to confirm our conclusions from Ki67 immunostaining. BrdU was administered into a $p63^{Cre^{ERT2/+}};R26R^{YFP/+};Klf5^{-/KR}$ mouse at 16 hours before tissue collection, and tissue sections were then costained for BrdU, YFP and basal and luminal markers. As shown in new Fig. S6n, knockin of $Klf5^{KR}$ significantly increased BrdU incorporation rate in YFP+ luminal cells relative to YFP- luminal cells, supporting our conclusion from Ki67 staining that knockin of $Klf5^{KR}$ promoted luminal cell proliferation. In basal cells, BrdU incorporation rate in YFP+ cells was similar to that in YFP- cells, indicating that knockin of $Klf5^{KR}$ did not affect basal cell proliferation, which is also consistent with the finding from Ki67 staining.

We also detected PCNA (Response Fig. 1, see *next page*), another proliferating marker, to further confirm the proliferation status of basal and luminal cells. Similarly, more PCNA positive cells were observed in YFP+ luminal cells from the $Klf5^{KR}$ knockin group than in those in YFP- luminal cells or control groups (Response Fig. 1A, 1C). Once again, $Klf5^{KR}$ knockin did not affect PCNA positive rate in basal cells (Response Fig. 1A, 1B). Therefore, use of PCNA as another cell proliferation marker would also supports our conclusions about proliferation rates.

Response Figure 1

Comment 2: Another issue is the low Cre-mediated recombination efficiency in these studies (approximately 9%; Figure S2C), raising the possibility that a subset of basal cells that is not representative of the entire population has been analyzed. Consistent with this possibility, the percentage of YFP-positive luminal cells before and after castration is similar (Fig. 4G, 7E), which is unexpected since most luminal cells undergo apoptosis following castration.

Response 2: We agree with the reviewer that the efficiency of Cre-mediated recombination was not as high (new Fig. S2c), considering that p63-Cre-mediated recombination was close to 28% in basal cells when tamoxifen was administered at 2 weeks after birth⁶. The reasons are unknown but genetic backgrounds of mouse strains could be one of them. Nevertheless, a lower labeling efficiency should not impact the representativeness of labeled cells. In addition, a lower labeling efficiency could have prevented the formation of basal cell clusters (two or more YFP+ adjacent basal cells) after tamoxifen administration, which in fact allowed us to use YFP+ luminal units over basal cells to draw the conclusion for the roles of *Klf5* and its acetylation in basal-to-luminal differentiation. We would also like to emphasize that the distribution pattern of YFP+ basal cells looked random, which at least partially excluded the preference in labeling basal progenitors by YFP.

The percentage of YFP+ luminal cells before and after castration was not different in the *Klf5*^{-/+} group (approximately 3.3% before castration and approximately 3.6% after castration); but for the *Klf5*^{-/KR} knockin group, the percentage of YFP+ luminal cells was dramatically decreased by castration from 13.6% before castration to 0.4% after castration. Because both YFP+ and YFP- luminal cells undergo apoptosis after castration, it makes sense that the percentages of both YFP+ and YFP- luminal cells in the *Klf5*^{-/+} group (one normal *Klf5* allele) were similar before and after castration. Findings in the *Klf5*^{-/+} group also indicate that luminal

cells derived from basal progenitors (YFP+) have the same response to castration as do those not from basal progenitors (YFP-). This background makes the data in the *Klf5*^{-KR} group more profound in indicating that acetylation of Klf5 is required for the survival of basal progenitor-derived luminal cells following castration.

Comment 3: Finally, although the authors have provided much more quantitation of their data in the revised manuscript, there are still several gaps. For example, a short-term analysis of marking immediately after tamoxifen induction at P7 was not performed, and the ratio of labeled basal cells to luminal units is not provided for the experiment shown in Figure 8.

Response 3: We appreciate the reviewer for this comment. In our study, knockin of *Klf5*^{KR} by p63-Cre was induced by tamoxifen at postnatal day 7 for 5 consecutive days. We collected the mice immediately after tamoxifen treatment and then stained basal marker p63 and YFP (new Fig. S6e). Knockin of *Klf5*^{KR} did not affect the efficiency of YFP labeling in basal cells under this condition.

According to the reviewer’s suggestion, we also provided the ratio of YFP labeled luminal units to basal cells after one castration-regeneration cycle (new Fig. S8f) and after 16-weeks of normal development (new Fig. S8g). As expected, the ratio of YFP+ luminal units to basal cells was increased after normal development for 16 weeks, although the increase did not reach statistical significance. Notably, *Klf5*^{KR} knockin significantly increased the ratio of YFP+ luminal units to basal cells at 8 weeks (Response Fig. 2, corresponding to new Fig. 4), and the trend is similar to that seen for 16 weeks of normal development. After one castration-regeneration cycle, however, the ratio of YFP+ luminal units to basal cells was sharply decreased, suggesting that the basal to luminal differentiation is attenuated by the interruption of Klf5 acetylation during the regeneration of prostates.

Specific comments 1:

Line 266:
“Underdeveloped prostates with fewer basal cells but more luminal cells suggests that interruption of Klf5 acetylation may cause more basal cells to unnecessarily differentiate into luminal cells, although other mechanisms could also be involved.” What are these other mechanisms?

Response 1: We did not have any specifics in this regard. Accordingly, we have deleted this sentence to avoid misunderstanding (Line 269 in the revised manuscript).

Specific comments 2: *Line 278: “Decrease in basal cells, increase in luminal cells, decrease in cystic organoids, and of organoid organization by the interruption of Klf5 acetylation further support a role of Klf5 acetylation in the proper differentiation of basal to luminal cells.” However, basal-to-luminal differentiation could be unaffected while luminal cell proliferation is increased in mutants.*

Response 2: This sentence was limited to the organoid assay of prostate cells with constitutive Klf5^{KR} knockin, so it was not conclusive for a role of Klf5 acetylation in basal-to-luminal differentiation. We therefore revised the sentence to the following: “Decrease in basal cells, increase in luminal cells, decrease in cystic organoids, and disruption of organoid organization by the interruption of Klf5 acetylation suggests a role of Klf5 acetylation in the proper development of prostates.” (Line 281 in the revised manuscript). Role of Klf5 acetylation in the basal-to-luminal differentiation is more conclusive in the induced Klf5^{KR} knockin mouse model.

Specific comments 3: *Line 560: “The roles of Klf5 and its acetylation in the maintenance of basal progenitors and their luminal differentiation greatly impact postnatal development of the prostate”. The authors show that prostate weight is affected at 8, 12, and 16 weeks, but not 24 weeks (Figure 3C), and other analyses are only shown at 8 weeks. These results suggest that the effects of Klf5 deletion/mutation are compensated for by 24 weeks of age.*

Response 3: We agree with the reviewer on this possibility. Accordingly, we have added this point in the Discussion: “Interestingly, the decrease in prostate weight was observed at 8, 12 and 16 weeks but not at 24 weeks (Fig. 3c), suggesting that the effect of Klf5 deletion or deacetylation are compensated for by 24 weeks of age.”(Line 572 in the revised manuscript).

Specific comments 4: *Line 638: To state that Klf5 and its acetylation are “cell fate determinants of basal progenitor cells” represents an overstatement. Instead, it seems more appropriate to say that Klf5 and its acetylation is essential for basal-to-luminal differentiation.*

Response 4: We have revised the manuscript according to this comment: “These findings establish that Klf5 and its acetylation maintain basal progenitors and proper basal to luminal differentiation in prostate epithelial homeostasis.” (Line 647 in the revised manuscript).

Specific comments 5: *Figure 2i: This plot should be labeled as percentage of YFP-positive basal and luminal cells, not percentage of survival, as cell death was not examined.*

Response 5: We have revised the figure as suggested (new Fig. 2i).

Specific comments 6: *Figure 4: This figure should contain a plot similar to that in Figure 2i.*

Response 6: We had this panel in Fig. S6c, which has now been revised in its style according to the suggestion (new Fig. S6c).

Specific comments 7: *Figure 5: The DHT-negative conditions are confusing and do not really provide additional insights. These data could be removed or at least separated into a supplemental figure.*

Response 7: The changes in organoids caused by DHT treatment in wildtype group validated the organoid system, so the DHT-negative groups served as controls for the effect of DHT. As it is necessary to compare the $Klf5^{KR}$ group to the wildtype group for changes before and after DHT treatment for whether $Klf5^{KR}$ knockin attenuates androgen-induced formation of cystic organoids and luminal differentiation from basal cells, separation of the DHT-negative groups to the supplementary data would interfere with the interpretation and understanding of the data.

Reference:

1. Wang, S. *et al.* Prostate-specific deletion of the murine Pten tumor suppressor gene leads to metastatic prostate cancer. *Cancer Cell* **4**, 209-221 (2003).
2. Wang, S. *et al.* Pten deletion leads to the expansion of a prostatic stem/progenitor cell subpopulation and tumor initiation. *Proc Natl Acad Sci U S A* **103**, 1480-1485 (2006).
3. Lu, T.L. *et al.* Conditionally ablated Pten in prostate basal cells promotes basal-to-luminal differentiation and causes invasive prostate cancer in mice. *Am J Pathol* **182**, 975-991 (2013).
4. Xing, C. *et al.* Klf5 deletion promotes Pten deletion-initiated luminal-type mouse prostate tumors through multiple oncogenic signaling pathways. *Neoplasia* **16**, 883-899 (2014).
5. Ousset, M. *et al.* Multipotent and unipotent progenitors contribute to prostate postnatal development. *Nat Cell Biol* **14**, 1131-1138 (2012).
6. Lee, D.K., Liu, Y., Liao, L., Wang, F. & Xu, J. The prostate basal cell (BC) heterogeneity and the p63-positive BC differentiation spectrum in mice. *Int J Biol Sci* **10**, 1007-1017 (2014).
7. Pignon, J.C. *et al.* p63-expressing cells are the stem cells of developing prostate, bladder, and colorectal epithelia. *Proc Natl Acad Sci U S A* **110**, 8105-8110 (2013).
8. Wang, Z.A. *et al.* Lineage analysis of basal epithelial cells reveals their unexpected plasticity and supports a cell-of-origin model for prostate cancer heterogeneity. *Nat Cell Biol* **15**, 274-283 (2013).

Reviewers' Comments:

Reviewer #3:

Remarks to the Author:

The authors have satisfactorily addressed the remaining critiques from the previous submission.